# From Link Prediction to Forecasting: Addressing Challenges in Batch-based Temporal Graph Learning

**Moritz Lampert** ⬤                                       *moritz.lampert@uni-wuerzburg.de*
*Chair of Machine Learning for Complex Networks*
*Center for Artificial Intelligence and Data Science (CAIDAS)*
*Julius-Maximilians-Universität Würzburg, DE*

**Christopher Blöcker** ⬤                              *christopher.bloecker@uni-wuerzburg.de*
*Chair of Machine Learning for Complex Networks*
*Center for Artificial Intelligence and Data Science (CAIDAS)*
*Julius-Maximilians-Universität Würzburg, DE*

**Ingo Scholtes** ⬤                                        *ingo.scholtes@uni-wuerzburg.de*
*Chair of Machine Learning for Complex Networks*
*Center for Artificial Intelligence and Data Science (CAIDAS)*
*Julius-Maximilians-Universität Würzburg, DE*

**Reviewed on OpenReview:** *https://openreview.net/forum?id=iZPAykLE3l*

## Abstract

Dynamic link prediction is an important problem considered in many recent works that propose approaches for learning temporal edge patterns. To assess their efficacy, models are evaluated on continuous-time and discrete-time temporal graph datasets, typically using a traditional batch-oriented evaluation setup. However, as we show in this work, a batch-oriented evaluation is often unsuitable and can cause several issues. Grouping edges into fixed-sized batches regardless of their occurrence time leads to information loss or leakage, depending on the temporal granularity of the data. Furthermore, fixed-size batches create time windows with different durations, resulting in an inconsistent dynamic link prediction task. In this work, we empirically show how traditional batch-based evaluation leads to skewed model performance and hinders the fair comparison of methods. We mitigate this problem by reformulating dynamic link prediction as a *link forecasting* task that better accounts for temporal information present in the data.

## 1 Introduction

Many scientific fields study data that can be modelled as graphs, where nodes represent entities connected by edges, such as social (Lazer et al., 2009), financial (Bardoscia et al., 2021), or molecular networks (David et al., 2020). Apart from the mere topology of interactions, that is, who is connected to whom, network data increasingly include information on *when* interactions occurred. The resolution of such *temporal graphs* varies from snapshots with, for example, yearly political collaboration networks (Fowler, 2006) – called *discrete-time* temporal graphs (Kazemi et al., 2020) – to high-resolution online interactions (Kumar et al., 2019) or contacts between individuals via proximity (Vanhems et al., 2013), named *continuous-time* temporal graphs.

Building on the growing importance of temporal data and the success of graph neural networks (GNNs) for static graphs (Corso et al., 2024), deep graph learning has recently been extended to temporal (or dynamic) graphs (Feng et al., 2024). To this end, several temporal graph neural network (TGNN) architectures have

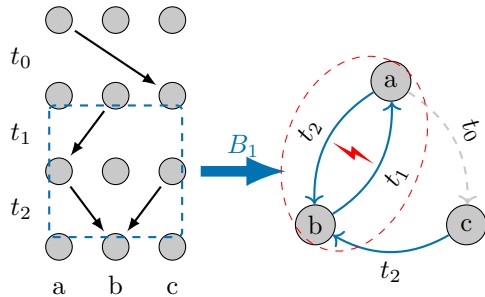

(a) **Information loss:** Three edges with different time-stamps are grouped into the same batch with batch size $b = 3$. The batched (blue) edges are processed in parallel and can only utilise information from previous batches (grey dashed edges). This implies that edge $(b, a, t_1)$ cannot be used for the prediction of edges $(a, b, t_2)$ (highlighted with red bolt) or $(c, b, t_2)$.

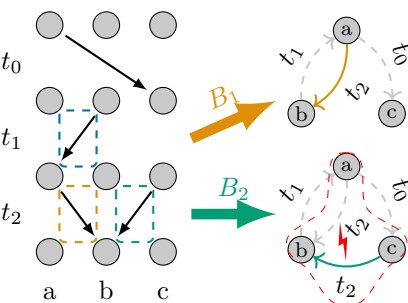

(b) **Information leakage:** Edges $(a, b, t_2)$ and $(c, b, t_2)$ with the same timestamp are grouped into different batches with batch size $b = 1$. The two edges are processed consecutively due to their batch assignment. Thus, information about edge $(a, b, t_2)$ (grey dashed edge in $B_2$ because it was previously processed in $B_1$) is leaked during the prediction of edge $(c, b, t_2)$ in batch $B_2$ (green).

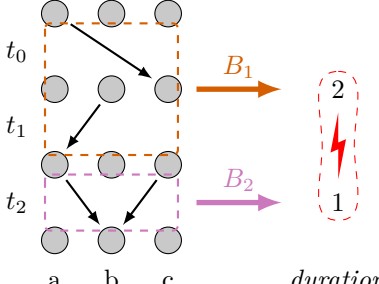

(c) **Varying batch durations:** The durations of batches $B_1$ and $B_2$ differ for batch size $b = 2$. This changes the difficulty of the link prediction task for each batch because this difficulty can strongly depend on the prediction horizon. For example, it is often easier to predict interactions in the near than in the distant future.

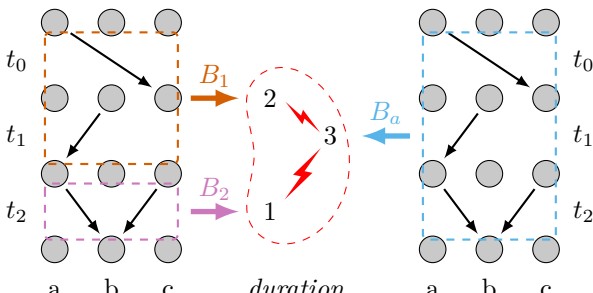

(d) **Tunable batch size:** Treating the batch size as a hyperparameter aggravates the problem shown in (c). Since batch duration depends on batch size, using different batch sizes results in different batch durations, such as for $b = 2$ vs. $b' = 4$. This effectively makes it possible to tune the task difficulty to the model.

Figure 1: Illustration of the issues in batch-based evaluation of dynamic link prediction.

been proposed that simultaneously learn temporal and topological patterns. These architectures are often evaluated in a *dynamic link prediction* setting, where the task is to predict the existence of edges during some future time window, for example, to provide recommendations to users (Kumar et al., 2019).

TGNNs utilise *temporal batches* to speed up training for dynamic link prediction. To construct these batches, the temporally ordered edges are divided into a sequence of chunks, all containing the same number of edges, which causes several problems illustrated in Figure 1: (a) Some TGNNs process edges within a batch *as if* they occurred simultaneously, essentially discarding the temporal information between edges of the same batch, which is problematic in high-resolution data. (b) In low-resolution data, snapshots typically contain many edges with identical timestamps; such edges are often assigned to different batches. This introduces an artificial temporal order amongst those edges that is not present in the data, leaking information that should not be available to the model. (c) Fixed-size batches often result in varying batch durations, effectively leading to an incoherent task with varying difficulty for each batch. For example, it is easier to predict the weather 24 times tomorrow (at each hour) than to predict it for the next 24 days at midday. (d) Related to the previous issue, treating the batch size as a tunable parameter affects the resulting batch durations and changes the task difficulty, thus making model performance between different models incomparable when they choose different batch sizes.

The issues (a)–(d) arise mainly from applying standard procedures from *static* link prediction to temporal graphs, and can be problematic for real-world applications where the timing of edge occurrences is crucial. For example, consider routing in opportunistic networks, where the goal is to find the fastest routes between nodes in a temporal network, such as for cellular traffic offloading (Trifunovic et al., 2017). Using temporal link prediction to find edges that will occur in the future can improve routing efficiency (Fang et al., 2021; Gou & Wu, 2022; Huang et al., 2015; Shu et al., 2022). However, incorrect predictions harm routing efficiency, especially during periods of low activity, when the routing problem is harder due to the graph's sparsity (Garg et al., 2020). Therefore, an accurate performance estimate of the prediction model *for each time window* is crucial. Yet, current batch-based evaluation approaches group edges into equally-sized batches, regardless of their occurrence time. Because this approach weighs each edge equally, it essentially ignores the detrimental effect of incorrect predictions on routing efficiency during periods of low activity.

To address these issues in the evaluation of link prediction in temporal graphs, we propose *link forecasting*— a task definition based on equally long and, therefore, equally weighted time windows instead of batches of varying duration. Our link forecasting definition mitigates the aforementioned issues (a)–(d) of link prediction by adequately considering the available temporal information. Link forecasting provides more realistic performance estimates for applications such as routing in opportunistic networks. In addition to introducing link forecasting, we make the following contributions:

- We quantify information loss and leakage resulting from the aggregation of edges into batches on 14 empirical temporal graphs.

- We experimentally evaluate state-of-the-art TGNNs for *link forecasting*. Our results identify scenarios where batch-based evaluation leads to skewed model performance, for example, in datasets with inhomogeneously distributed temporal activity for evaluation. Further, we show that the batch-based approach overestimates the performance of memory-based models in discrete-time datasets.

- We provide implementations of the link forecasting task for commonly used graph learning frameworks such as DyGLib (Yu et al., 2023) and PyTorch Geometric (Fey & Lenssen, 2019).

Although parallel processing of edges is a technical necessity for efficient model training, we demonstrate how ignoring the occurrence time of edges when splitting them into batches leads to various problems. With dynamic link forecasting, we propose a simple yet effective solution. While having a similar runtime to dynamic link prediction, our time-window-based evaluation facilitates a fairer and more realistic evaluation that better reflects real-world scenarios.

## 2 Preliminaries and Related Work

**Temporal Graphs.** A temporal (or dynamic) graph $G = (V, E)$ is a tuple where $V$ is the set of $n = |V|$ nodes and $E = \{(u_1, v_1, t_1), \ldots, (u_m, v_m, t_m)\}$ with $t_1 \leq \cdots \leq t_m$ is a chronologically ordered sequence of $m = |E|$ time-stamped edges (Wang et al., 2021c; Yu et al., 2023). Each node $v_i$ can have static node features $\mathbf{h_i} \in H_V$ and each temporal edge $(u_i, v_j, t)$ can have edge features $\mathbf{e_{ij,t}} \in H_E$. We assume that interactions occur instantaneously with timestamps $t \in \mathbb{R}$. This type of temporal graph is often categorised as continuous-time (Kazemi et al., 2020; Skarding et al., 2021). In contrast, discrete-time temporal graphs coarse-grain time-stamped edges into a sequence of static snapshots $(G_{t_i:t_j})$, where $G_{t_i:t_j} = (V, E_{t_i:t_j})$ with $E_{t_i:t_j} = \{(u, v) \mid \exists (u, v, t) \in E : t_i \leq t < t_j\}$ (Xue et al., 2022).

**Dynamic Link Prediction.** Given time-stamped edges up to time $t$, the goal of dynamic link prediction is to predict whether an edge $(v, u, \tau)$ exists at future time $\tau > t$ (Kazemi et al., 2020; Wang et al., 2021c; Poursafaei et al., 2022; Yu et al., 2023). In practice, it is often computationally infeasible to train and evaluate models on all possible edges, one edge at a time. Thus, the chronologically ordered sequence of edges $E$ is usually divided into temporal batches $B_i^+$, where each batch has a fixed size, containing $b$ edges. Edges within the same batch are typically processed in parallel (Rossi et al., 2020; Su et al., 2024). In addition to the existing (positive) edges $(u, v, \tau) \in B_i^+$, non-existing (negative) edges $(u^-, v^-, \tau^-) \in B_i^-$ are sampled and used for training and evaluation (see Appendix A for a brief overview of negative sampling

approaches). This is done since real-world graphs are typically sparse, and using all possible negative edges between all possible node pairs at all points in time would lead to a large class imbalance and prohibitively long runtime. Formally, the dynamic link prediction task is defined as follows:

**Definition 2.1.** Let $G = (V, E)$ be a temporal graph with node features $H_V$ and edge features $H_E$. Let $b$ be the batch size and $B_i^+ := \{(u_j, v_j, t_j) \mid \exists (u_j, v_j, t_j) \in E : b \cdot i < j \leq b \cdot (i+1)\}$ the $i$-th batch containing $b$ edges. We further use $B_i^-$ to denote a set of negative edges drawn using negative sampling as described in Appendix A. For a given batch $i$ we use $\hat{E}_i = \{(u_k, v_k, t_k) \mid \exists (u_k, v_k, t_k) \in E : k \leq b \cdot i\}$ to denote the *past edges* up to batch $i$. The goal of *dynamic link prediction* is to find a model $f_\theta(u, v, \tau \mid \hat{E}_i, H_V, H_{\hat{E}_i})$ with parameters $\theta$ that, for all future edges $(u, v, \tau) \in B_i^+ \cup B_i^-$ in each batch $i$, predicts whether $(u, v, \tau) \in B_i^+$ or $(u, v, \tau) \in B_i^-$ using the past edges $\hat{E}_i$.

**State-of-the-Art TGNNs.** Current state-of-the-art dynamic link prediction methods, such as JODIE (Kumar et al., 2019), DyRep (Trivedi et al., 2019), TGN (Rossi et al., 2020) keep an up-to-date memory of temporal information in the graph by utilising recurrent neural networks. TGAT (Xu et al., 2020) extends graph attention to the temporal domain and replaces positional encodings in GAT (Velickovic et al., 2018) with a vector representation of time. TCL (Wang et al., 2021a) uses a transformer-based architecture to capture the nodes' time-evolving properties. CAWN learns temporal motifs based on causal anonymous walks (Wang et al., 2021c). GraphMixer takes an attention-free and transformer-free approach, using an MLP-based link encoder, a mean-pooling-based node encoder, and an MLP-based link classifier for predictions (Cong et al., 2023). DyGFormer combines nodes' historical co-occurrences as interaction targets of the same source node with a temporal patching approach to capture long-term histories (Yu et al., 2023). Further approaches for temporal graph learning exist, including ROLAND (You et al., 2022), DyGEM (Taheri et al., 2019), DySAT (Sankar et al., 2020), EvolveGCN (Pareja et al., 2020), and DBGNN (Qarkaxhija et al., 2022). For a recent survey of deep-learning-based dynamic link prediction, we refer to Feng et al. (2024).

### Open Challenges in Dynamic Link Prediction

**Training.** Recent works (Zhou et al., 2022; 2023; Feldman & Baskin, 2024; Su et al., 2024) identified issues in the training of memory-based TGNNs with large batch sizes: Processing edges belonging to the same batch in parallel ignores their temporal dependencies, resulting in varying performance depending on the chosen batch size. This issue, called *temporal discontinuity*, is caused by Problem (a) from above, that is, treating edges within the same batch as if they occurred simultaneously (see Appendix D). To mitigate this issue, Zhou et al. (2023) introduced a distributed training framework using smaller batch sizes. Another approach, PRES (Su et al., 2024), accounts for intra-batch temporal dependencies using a prediction-correction scheme. Feldman & Baskin (2024) proposed a model with a decoupling strategy to mitigate this problem. However, these works focus on training instead of evaluation, which is our work's focus.

Parallel to our work, Huang et al. (2024) formulate the unified temporal graph (UTG) representation, which facilitates learning on continuous-time graphs with discrete-time TGNNs by proposing conversion methods between discrete- and continuous-time temporal graphs. They also train continuous-time TGNNs on discrete-time graphs and briefly touch upon information leakage as we described in Problem (b). However, their work does not consider all Problems (a)–(d), nor include an empirical evaluation of how Problem (b) affects model performance, which we address here. Related to the above, Rossi et al. (2023) investigated how different conversion approaches from continuous- to discrete-time temporal graphs affect the graphs' structural properties.

**Evaluation.** Recent progress in terms of TGNN evaluation includes the temporal graph benchmark (TGB) (Huang et al., 2023), similar to the static open graph benchmark (OGB) (Hu et al., 2020). Poursafaei et al. (2022) identified problems with random negative sampling for dynamic link prediction and proposed novel time-dependent negative sampling techniques to improve the evaluation of TGNNs. Gastinger et al. (2023) pointed out issues in the evaluation of temporal knowledge graph forecasting. Although none of the models used for temporal knowledge graph forecasting overlap with regular TGNNs for dynamic link prediction, some of the problems are related, for example, differences in forecasting horizons leading to incomparable results.

# 3 From Link Prediction to Link Forecasting

Learning temporal patterns in a batch-oriented fashion leads to issues in continuous-time and discrete-time graphs. We demonstrate these issues in eight continuous-time and six discrete-time temporal graphs, whose characteristics are summarised in Table 1 and Appendix C. To address these issues, we formulate the *link forecasting* task based on time windows with a fixed duration.

## 3.1 Problems in Batch-based Dynamic Link Prediction

Below, we show empirically that batching causes information loss or leakage because it ignores the existing temporal order of links or induces an artificial order between them. When link densities vary over time, temporal batches lead to incoherent tasks because they produce varying prediction durations. Consequently, the results from different models become incomparable if they use different batch sizes. In practice, this happens if the batch size is treated like a hyperparameter and different models choose different batch sizes.

**(a) Information Loss.** For efficiency, edges from different timestamps are grouped into batches and processed in parallel. This parallel processing implies that earlier edges cannot inform the prediction of later edges if they are in the same batch, for example, as recent neighbours or as part of the memory in memory-based models (see Figure 1a). Therefore, assigning temporal edges with different timestamps to a single batch discards the temporal ordering of those edges, resulting in information loss. In Figure 2 we use normalised mutual information (NMI) (Vinh et al., 2010) to measure this information loss. NMI quantifies how much information observing one random variable conveys about another random variable (see Appendix B for details and an example). NMI yields values between 0, meaning "no infor-

Table 1: Characteristics of continuous- (top) and discrete-time (bottom) temporal graphs. For each dataset, we list the number of nodes $n$ and edges $m$, the resolution of timestamps, the dataset's total duration $T$, the average number of edges $\overline{|E_t|}$ with the same timestamp $t$, and the temporal density $T/m$. The US L. dataset is measured in congresses (c). The other datasets are measured in conventional units of time, i.e., seconds (s), minutes (min), days (d) or years (a). See Appendix C for more information on each dataset.

| Dataset | $n$ | $m$ | Res. | $T$ | $\overline{|E_t|}$ | $T/m$ |
|---|---|---|---|---|---|---|
| Enron | 184 | 125 235 | 1 s | 3.6 a | $5.5 \pm 16.6$ | 908.2 s |
| UCI | 1899 | 59 835 | 1 s | 193.7 d | $1.0 \pm 0.3$ | 279.7 s |
| MOOC. | 7144 | 411 749 | 1 s | 29.8 d | $1.2 \pm 0.5$ | 6.2 s |
| Wiki. | 9227 | 157 474 | 1 s | 31.0 d | $1.0 \pm 0.2$ | 17.0 s |
| LastFM | 1980 | 1 293 103 | 1 s | 4.3 a | $1.0 \pm 0.1$ | 106.0 s |
| Myket | 17 988 | 694 121 | 1 s | 197.0 d | $1.0 \pm 0.0$ | 24.5 s |
| Social | 74 | 2 099 519 | 1 s | 242.3 d | $3.7 \pm 2.5$ | 10.0 s |
| Reddit | 10 984 | 672 447 | 1 s | 31.0 d | $1.0 \pm 0.1$ | 4.0 s |
| UN V. | 201 | 1 035 742 | 1 a | 71.0 a | $14\,385.3 \pm 7142.1$ | 36.1 min |
| US L. | 225 | 60 396 | 1 c | 11.0 c | $5033.0 \pm 92.4$ | $1.8 \cdot 10^{-4}$ c |
| UN Tr. | 255 | 507 497 | 1 a | 31.0 a | $15\,859.3 \pm 3830.8$ | 32.1 min |
| Can. P. | 734 | 74 478 | 1 a | 13.0 a | $5319.9 \pm 1740.5$ | 91.8 min |
| Flights | 13 169 | 1 927 145 | 1 d | 121.0 d | $15\,796.3 \pm 4278.5$ | 5.4 s |
| Cont. | 692 | 2 426 279 | 5 min | 28.0 d | $300.9 \pm 342.4$ | 1.0 s |

mation", and 1, i.e. "full information". By treating the index $i$ of each batch $B_i$ assigned to each edge $(u, v, t) \in B_i$ as one random variable and the associated edge's timestamp $t$ as the other, we measure the temporal information that is retained after dividing edges into batches. An NMI value of 1 means that we can reconstruct the timestamps of edges correctly from their batch number, and a value of 0 means that batch numbers do not carry any information about timestamps, that is, total loss of information.

Figure 2a shows that larger batches result in higher information loss (lower NMI) for continuous-time temporal graphs where timestamps have a high resolution. This is because assigning edges that occur at different times to the same batch discards their temporal ordering—the larger the batch size, the stronger the coarse-graining effect and the more information is lost. A batch size of $b = 1$ preserves most temporal information and maximises NMI because we obtain a bijective mapping between almost all timestamps and batch numbers, except when multiple edges occur simultaneously.

Figure 2b shows the batch-size dependent NMI for discrete-time temporal graphs. The "optimal" batch size that retains most temporal information depends on the average number of links per snapshot and, thus, on the characteristics of the data. Too small batch sizes impose an ordering on the edges within the snapshots

that is not present in the data. Too large batches stretch across snapshots and discard the temporal ordering of edges from different batches.

**(b) Information Leakage.** In discrete-time graphs, the number of edges with a given timestamp often exceeds the batch size (see Appendix C.4). This means that edges with the same timestamp need to be assigned to different subsequent batches, which are processed sequentially (see Figure 1b). The same problem can arise in continuous-time graphs in moments of high activity, where the number of edges with a specific timestamp surpasses the batch size. Memory-based models utilise information from previous batches to make predictions for later batches by updating their internal node representations. But because of the issues above, this can include edges with the same timestamp and lead to an unfair advantage for memory-based models, as the already processed edges leak information about the remaining edges that have the same timestamp.

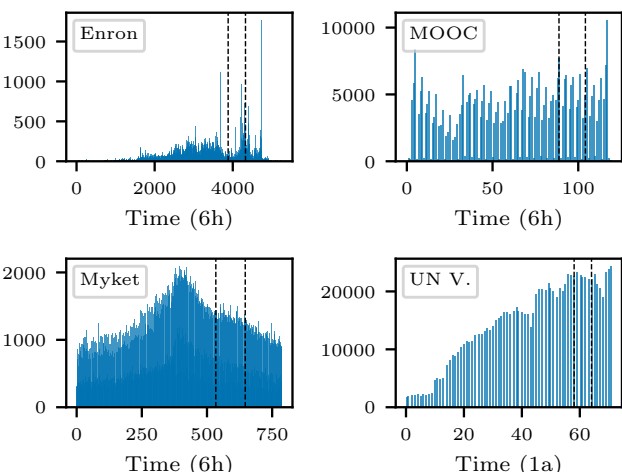

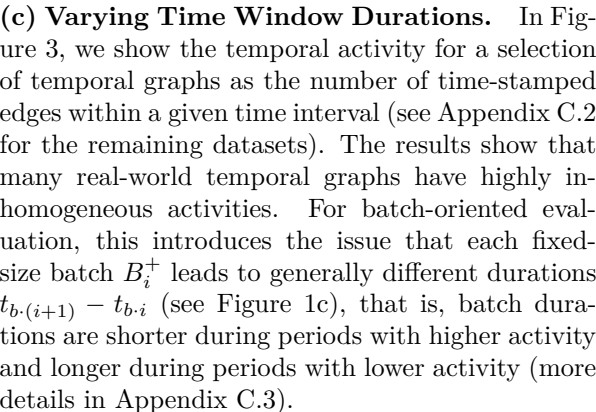

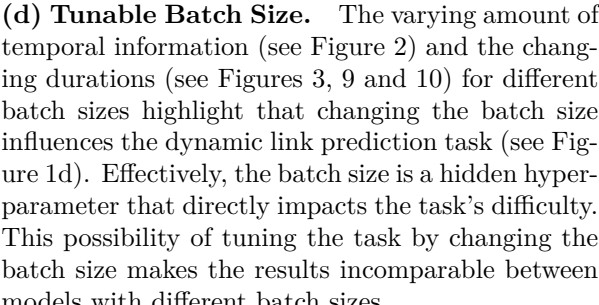

**(c) Varying Time Window Durations.** In Figure 3, we show the temporal activity for a selection of temporal graphs as the number of time-stamped edges within a given time interval (see Appendix C.2 for the remaining datasets). The results show that many real-world temporal graphs have highly inhomogeneous activities. For batch-oriented evaluation, this introduces the issue that each fixed-size batch $B_i^+$ leads to generally different durations $t_{b \cdot (i+1)} - t_{b \cdot i}$ (see Figure 1c), that is, batch durations are shorter during periods with higher activity and longer during periods with lower activity (more details in Appendix C.3).

Figure 3: Real-world datasets exhibit diverse edge occurrence patterns. For selected graphs, we visualise the edge density over time by counting the edges per time interval as indicated in the axis captions (histograms for all datasets in Appendix C.2). Dashed lines divide the datasets into 70% train, 15% validation, and 15% test sets as used in Section 4.

**(d) Tunable Batch Size.** The varying amount of temporal information (see Figure 2) and the changing durations (see Figures 3, 9 and 10) for different batch sizes highlight that changing the batch size influences the dynamic link prediction task (see Figure 1d). Effectively, the batch size is a hidden hyperparameter that directly impacts the task's difficulty. This possibility of tuning the task by changing the batch size makes the results incomparable between models with different batch sizes.

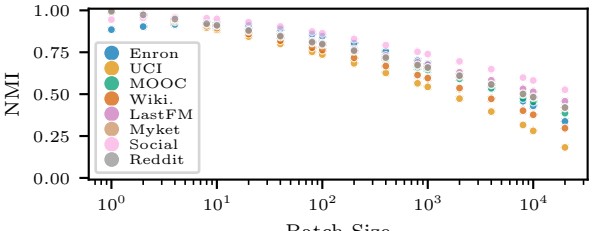

(a) **Continuous time:** Larger batch sizes assign more edges with different timestamps to the same batch, resulting in increased information loss.

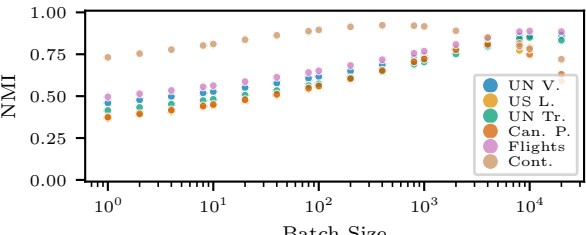

(b) **Discrete time:** Edges with the same timestamp are processed in different batches sequentially, discarding the information that edges co-occur in snapshots.

Figure 2: NMI (y-axis) measures the temporal information loss as a function of different batch sizes (x-axis), where smaller NMI values indicate more information loss.

Consider the Myket dataset (Loghmani & Fazli, 2023) as an example: It contains users $v$ and Android applications $u$, connected at time $t$ when user $v$ installs application $u$. The timestamps are provided in seconds, and edges occur roughly every 30 seconds on average (see Table 1), making the expected time range for a batch with size $b = 2$ approximately 30 seconds. Thus, the task is to predict who installs which applications during the next 30 seconds. Choosing $b = 120$ or $b = 2880$ turns the task into a prediction problem for approximately the next hour or day, respectively. For empirical validation, we compare the performance of state-of-the-art TGNNs using two different batch sizes during evaluation in Appendix D and show that the performance can vary substantially between different batch sizes.

### 3.2 Our Solution: Dynamic Link Forecasting

In real-world applications, the prediction time window is inherently connected to the problem at hand, necessitating a task formulation that is chosen carefully – and fixed – for each dataset. To address the afore-mentioned issues, we propose *link forecasting*, which replaces link prediction with a task definition utilising a *fixed-duration* time window, generally resulting in a variable number of edges per window. Compared to dynamic link prediction with a fixed batch size, this task (i) controls the window durations in which the temporal ordering of edges is lost; (ii) does not introduce an artificial temporal ordering that leaks information; (iii) has a fixed time window duration regardless of edge occurrence frequency; and (iv) does not depend on a hyperparameter that is tuned for each model.

**Motivation.** The study of temporal information is at the centre of time series forecasting (Benidis et al., 2023), where the goal is to forecast the values of a time series for a given time window in the future. We define our task accordingly and interpret the temporal edges $E$ as $n^2$ Boolean time series, each of which takes the value 1 when an edge occurs. Standard multivariate models output a value for each timestamp over a forecasting horizon $h$. In large-scale temporal graphs, it is computationally infeasible to forecast the existence of all $n^2$ possible links, thus, only a sample of negative edges is considered instead. In continuous-time dynamic graphs, observations are available at high resolution, such as seconds, however, for many practical applications, predicting at lower granularity suffices. For example, it is typically enough to predict whether a customer will purchase a certain product within the next day or week. Therefore, we consider forecasting for all timestamps $(t, t+h]$ during a time window at once instead of for each of them individually. We define the link forecasting task as follows:

**Definition 3.1.** Let $G = (V, E)$ be a temporal graph with node features $H_V$ and edge features $H_E$. Let $h$ be the time horizon and $W_i^+ := \{(u, v, \tau) \mid \exists (u, v, \tau) \in E : i \cdot h < \tau \leq (i+1) \cdot h\}$ the set of edges in the $i$-th time window. We use $W_i^-$ to denote a sample of $|W_i^+|$ negative edges that are sampled using one of the negative sampling approaches described in Appendix A and do not occur as positive edges in time window $(i \cdot h, (i+1) \cdot h]$. We further use $\hat{E}_i = \{(\hat{u}, \hat{v}, \hat{t}) \mid \exists (\hat{u}, \hat{v}, \hat{t}) \in E : \hat{t} \leq i \cdot h\}$ to denote the set of past edges up to time $i \cdot h$. The goal of *dynamic link forecasting* is to find a model $f_\theta(u, v, \tau \mid \hat{E}_i, H_V, H_{\hat{E}_i})$ with parameters $\theta$ that, for all edges $(u, v, \tau) \in W_i^+ \cup W_i^-$ at future time $\tau > i \cdot h$ in each time window $i$, forecasts whether $(u, v, \tau) \in W_i^+$ or $(u, v, \tau) \in W_i^-$.

Crucially, link forecasting makes the evaluation *independent of a batch size $b$* and, instead, introduces a time horizon $h$ that defines the forecasting time window for coarse-graining the temporal information. This requires deliberately choosing the temporal resolution based on the dataset and application (see Appendix E for examples). Similar to batch-based dynamic link prediction, adopting a time-window-based perspective enables parallel processing of edges by aggregating temporal information.

To summarise, link prediction depends on the batch size, which can become a hyperparameter, tuned for each model to get the best performance. In contrast, link forecasting is based on a *dataset- or task-specific time horizon* that controls the amount of temporal information and ensures that the information is consistent for different models, facilitating fair performance comparisons.

**Computational Cost.** We can assign all links to their corresponding time window in $O(m)$ by checking each link's interaction time. After each link is assigned, the time complexity during model evaluation is the same as for the batch-based approach. Since the number of links per time window varies, windows

| NMI | Wiki. | Reddit | MOOC | LastFM | Myket | Enron | Social | UCI | Flights | Can. P. | US L. | UN Tr. | UN V. | Cont. |
|---|---|---|---|---|---|---|---|---|---|---|---|---|---|---|
| Batch | 0.9982 | 0.9933 | 0.9908 | 0.9996 | 0.9985 | 0.9030 | 0.9449 | 0.9988 | 0.9023 | 0.8316 | 0.9864 | 0.8918 | 0.8945 | 0.9240 |
| Window | 1.0000 | 1.0000 | 1.0000 | 1.0000 | 1.0000 | 1.0000 | 1.0000 | 1.0000 | 1.0000 | 1.0000 | 1.0000 | 1.0000 | 1.0000 | 1.0000 |

Table 2: Maximum NMI as a measure for the maximum amount of information that is preserved using the optimal batch size (top row) and horizon (bottom row) for each continuous- (left) and discrete-time (right) dataset. The results show that our window-based approach preserves full temporal information.

can become large during periods when many temporal edges occur. For those periods, time windows can be subdivided into smaller chunks for GPU-based computations. To prevent information leakage between chunks of the same time window, steps such as memory updates need to be accumulated and done based on the whole time window. We provide a more detailed explanation of possible memory overflow mitigation strategies and report the empirical runtime of both approaches in Appendix G.

**Implementation.** We provide implementations for our evaluation procedure in commonly used PyTorch libraries. Specifically, we implement a new `DataLoader` called `SnapshotLoader`[1] that replaces the widely used `TemporalDataLoader` in PyTorch Geometric (Fey & Lenssen, 2019). We extend DyGLib (Yu et al., 2023) by an argument `horizon` in the evaluation pipeline[2]. The latter was used for the experiments in this work and can be used to reproduce our results.

## 4 Link Prediction vs. Forecasting

First, we verify empirically that our proposed window-based evaluation can mitigate the information loss issue inherent to batch-based evaluation. We compare the maximum amount of information preserved (that is, the maximum achievable NMI) of both approaches for each dataset in Table 2. The top row of the table shows that no batch size exists that leads to no information loss for all datasets. In contrast, our window-based approach (bottom row) does not leak any information by definition, as demonstrated by NMI values of one.

Next, we experimentally evaluate the performance of nine state-of-the-art models (Kumar et al., 2019; Trivedi et al., 2019; Xu et al., 2020; Rossi et al., 2020; Wang et al., 2021a;c; Poursafaei et al., 2022; Cong et al., 2023; Yu et al., 2023), both for the (conventional) dynamic link prediction task as well as our proposed dynamic link forecasting task. We use implementation and model configurations provided by DyGLib (Yu et al., 2023) (see Appendix F) and repeat each experiment five times with different seeds to obtain averages. We use historical negative sampling (Poursafaei et al., 2022) (see Appendix A) and train each model using batch-based training and validation with batch size $b = 200$, which was found "to be a good trade-off between speed and update granularity" (Rossi et al., 2020) and adopted in later works (Poursafaei et al., 2022; Yu et al., 2023). Afterwards, we evaluate each trained model *with weights obtained by batch-based training* using both our proposed time-window-based evaluation method and the common batch-based evaluation approach. The setup is visualised in Figure 4.

Table 3: Average links per window $\overline{|W_i^+|}$ and standard deviation for horizon $h$ used in the evaluation. We chose $b = 200$, that is, $\overline{|B_i^+|} = 200$, and appropriate horizons $h$ for each dataset to get $\overline{|W_i^+|} \approx 200$. Detailed window size histograms are presented in Figure 13. NMI uses time window and batch IDs of the test set (see Appendix B) to quantify how much the chosen chunks differ between approaches.

| Dataset | $h$ | $\overline{|W_i^+|}$ | NMI |
|---|---|---|---|
| Enron | 172 800s (48h) | 214.1 ± 274.1 | 0.8017 |
| UCI | 57 600s (16h) | 208.5 ± 335.5 | 0.8252 |
| MOOC | 1200s ($^1$/3h) | 199.3 ± 167.2 | 0.8803 |
| Wiki. | 3600s (1h) | 211.7 ± 56.3 | 0.8903 |
| LastFM | 21 600s (6h) | 204.4 ± 120.2 | 0.9089 |
| Myket | 5400s ($^3$/2h) | 220.1 ± 133.6 | 0.9139 |
| Social | 1800s ($^1$/2h) | 186.1 ± 165.3 | 0.9131 |
| Reddit | 900s ($^1$/4h) | 226.0 ± 54.5 | 0.9152 |

---

[1] https://github.com/M-Lampert/pytorch_geometric/tree/snapshot-loader
[2] https://github.com/M-Lampert/DyGLib

Figure 4: Experimental setup: All TGNNs are trained using batch-based training and evaluated using batch- and window-based evaluation to ensure as much comparability as possible.

**Continuous-Time Temporal Graphs.** Following previous work (Rossi et al., 2020; Poursafaei et al., 2022; Yu et al., 2023), we set the batch size to $b = 200$ and choose the forecasting horizon $h$ such that the resulting time windows contain 200 edges on average to facilitate comparability with the batch-oriented approach (see Table 3 for the exact values). We quantify the amount of information shared between the two approaches in Table 3 as the NMI score between time window ID and batch ID (see Appendix B for a detailed explanation).

Figure 6 shows the change of AUC-ROC scores for time-window-based link forecasting relative to the batch-based evaluation of dynamic link prediction (detailed scores are provided in Appendix H.1 and H.2). The performance changes are presented in detail as a heatmap in Figure 6a and aggregated for each dataset across all models in Figure 6b. Both figures show that the change in performance between our window-based and the batch-based approach largely depends on the dataset: The lower datasets in both figures with a similar window duration for all fixed-sized batches (quantified by NMI scores close to one in Table 3 and a narrow window size distribution in Figure 13), such as Wikipedia, Reddit, or Myket, only exhibit small performance differences.

This is expected since we chose the horizon $h$ to produce time windows close to the same average size as the fixed-sized batches. Nevertheless, we observe long-tailed distributions in Figure 13 and lower NMI values in Table 3 for datasets with inhomogeneously distributed temporal activity, such as Enron or UCI—that is, the time windows do not fit the fixed-sized batches well. These datasets (placed at the top in Figure 6) with lower NMIs show substantial performance changes across models. This highlights that the performance scores of batch-based evaluation are skewed and may not reflect the models' performance in a real-world setting on temporal datasets with inhomogeneous activities.

To explain the causes of the performance differences of both evaluation approaches, we analyse the AUC-ROC scores for each batch and time window individually in Figure 5 (and Appendix H.3 for other datasets and models). We observe that the performance between the two approaches diverges most during phases of irregular temporal activity. Specifically, the batch-based evaluation is affected by an anomaly in the Enron dataset at day 1181, where

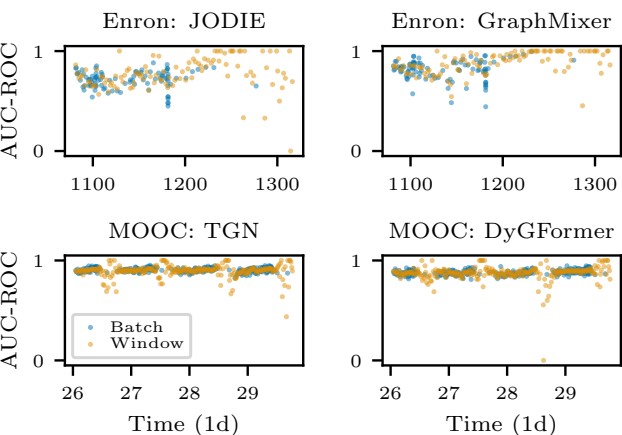

Figure 5: Model performance of selected continuous-time datasets varies over time (cf. Appendix H.3 for all datasets and models), which is visualised by AUC-ROC scores (y-axis) for each batch (blue) and time window (orange). For both datasets, the best-performing memory- and non-memory-based model is shown. The time (x-axis) is limited to the test set.

the performance of all models drops substantially. At this time, 1705 temporal edges with identical timestamps are evaluated in 9 consecutive batches. In contrast, our approach uses a single time window for all 1705 edges with identical timestamps. During the second half of the observation period for the Enron dataset (117 days), the batch-based approach yields only 5 batches, with durations of up to 73 days. In contrast, the average batch duration of the whole dataset is two days, which is consistent with the durations of the time windows of our window-based approach. Most models perform better during this period using our window-based evaluation compared to the batch-based approach. This performance increase suggests

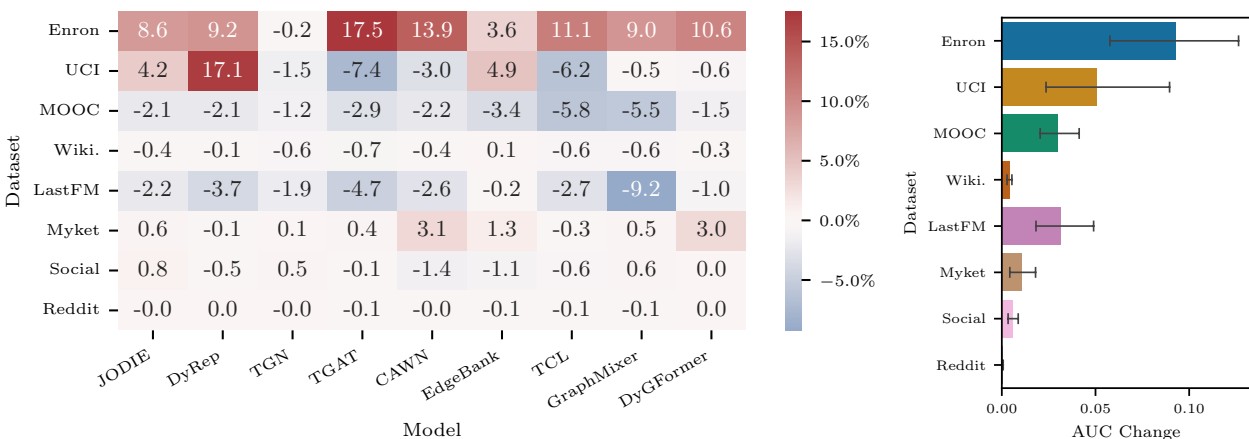

(a) Heatmap showing the change in performance in percentage. Datasets with lower NMI exhibit larger performance changes (top rows with warm or cold coloured cells), whereas datasets with NMI values close to 1 perform equally well (bottom rows with mostly white cells) for both evaluation approaches.

(b) Average absolute performance change. Datasets with lower NMI values (top) show larger performance changes.

Figure 6: Performance change of our window-based approach relative to the batch-based approach (a) for each continuous-time dataset and model combination and (b) averaged for each continuous-time dataset. The datasets are ordered by NMI as reported in Table 3.

that the loss of information that happens because edges across such a long period of time are grouped into one batch has a large impact on the performance. Additionally, when using the batch-based evaluation, the performance of the second half only has a small impact on the overall performance because it is outweighed by the other half with more batches. In contrast, our window-based approach weighs the performance of each time period equally. A similar (daily) pattern can be observed for MOOC, where each period of low activity follows a high-activity period. Our time-window-based evaluation reveals that most models are particularly unreliable during the low-activity periods, which is again outweighed by periods with high activity in the batch-based approach.

**Discrete-Time Temporal Graphs.** We set $h = 1$ to obtain one time window per snapshot with prediction time intervals ranging from five minutes to a year, depending on the dataset (see Table 1). Similar to Figure 6, Figure 7 shows the AUC-ROC score change and highlights the performance discrepancy between the two evaluation approaches—link prediction and link forecasting—for discrete-time temporal graphs. For link forecasting, the performance of memory-based models (JODIE, DyRep, TGN) decreases substantially on half of the datasets. This is expected since these models incorporate leaked information about the present snapshot by updating their memory based on prior batches, which means using part of the snapshot's edges to predict its remaining edges. This leakage happens when the batch size is smaller than the snapshot, thus splitting the snapshot into a sequence of batches that are processed sequentially. With link forecasting, we prevent this information leakage, explaining the substantial drop in performance.

We validate the effectiveness of our evaluation protocol in preventing information leakage in Appendix H.4. By shuffling the edges inside each snapshot, we destroy the implicit edge ordering within snapshots that may help memory-based models in their prediction. We compare the model performance on snapshots with and without shuffled edges and find that shuffling leads to a substantial drop in performance for link prediction but not for link forecasting. These results confirm that information leakage occurs for link prediction and that our evaluation protocol solves this issue.

**Realistic Choice of Time Horizons.** In the evaluation above, we chose the time horizon $h$ to match the batch size $b$ approximately to achieve as much comparability as possible between the link forecasting and prediction tasks. Thereby, we emphasise performance differences between both approaches since we use the same trained models and "only" group edges differently. In real-world applications, however, the models'

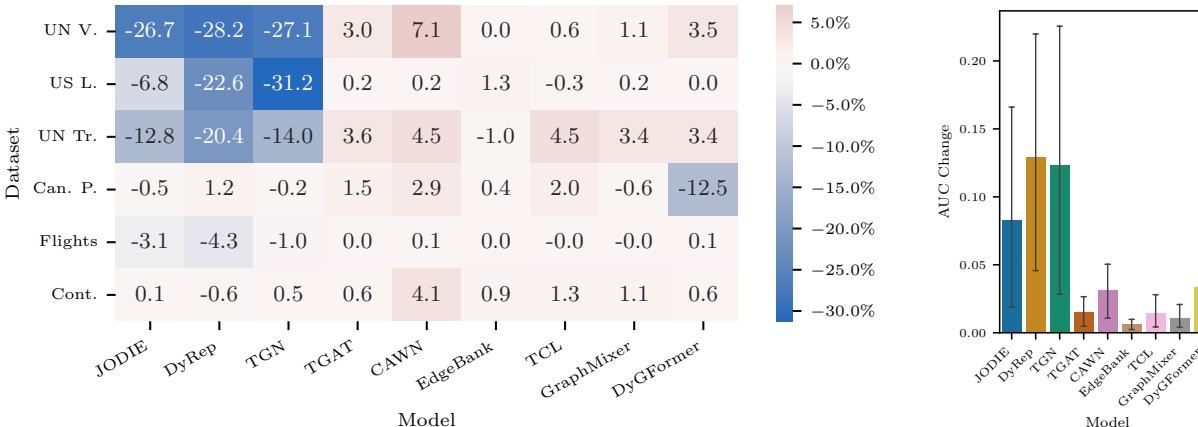

(a) Heatmap showing the change in performance in percentage. Memory-based models exhibit a large drop in performance when using our window-based evaluation, when compared to the batch-based evaluation on three discrete-time datasets.

(b) Avg. abs. perf. change. Performance for memory-based models changes substantially.

Figure 7: Performance change of our window-based approach relative to the batch-based approach (a) for each discrete-time dataset and model combination and (b) averaged for each model. Models are ordered by type. The three leftmost models are memory-based; all others are non-memory-based.

performance should be evaluated using time horizons adjusted to a specific use case or research question. We provide examples for such an evaluation using the above datasets in Appendix E. When evaluated on realistic time horizons, the performance differences (reported in Table 8) between link prediction and link forecasting are even larger than in Figure 6. Furthermore, the best-performing model using link forecasting with realistic horizons (see Table 6) compared to link prediction (see Table 15) changes for datasets such as LastFM or Myket, emphasising the need to evaluate the models in realistic scenarios to get accurate performance estimates.

## 5 Conclusion

In this work, we considered issues in current evaluation practices for dynamic link prediction in temporal graphs. For computational reasons, edges are split into fixed-size batches, which leads to multiple issues, including (a) information loss for continuous-time and (b) information leakage for discrete-time datasets. Additionally, (c) fixed-size batches have varying-length durations that depend on temporal activity patterns. (d) Treating the batch size as a tunable parameter further affects the duration of each batch, resulting in incomparable results between models that use different batch sizes.

We address these issues by formulating the *dynamic link forecasting* task. Dynamic link forecasting acknowledges the resolution at which temporal interaction data is recorded and explicitly considers a forecasting horizon corresponding to a time window with *fixed duration*. Depending on the dataset and problem setting, the horizon may span minutes, hours, or longer, but crucially, time windows always span the same length. We compared the dynamic link prediction and forecasting performance of nine state-of-the-art temporal graph learning approaches on 14 real-world datasets. We found substantial performance differences, especially for memory-based TGNNs on discrete-time datasets due to information leakage, which has been overlooked so far. On datasets with periods of irregular activity, we also observed large performance differences, indicating that most TGNNs are unreliable in these periods. However, periods with irregular activity are common in many real-world settings, like opportunistic networks (Huang et al., 2015). By providing a novel evaluation method – with implementations in commonly used PyTorch libraries – that is better suited to capture the performance in such data, our work aids the development of better methods for these tasks, e.g. cellular traffic offloading (Trifunovic et al., 2017).

**Limitations and Future Work**   Limitations of our work include that the used metrics treat dynamic link prediction and forecasting as a binary classification problem, using one negative sample for each positive edge. Other approaches (You et al., 2022) consider the problem as a ranking task, e.g., using the mean reciprocal rank (MRR), and compute the ranking of each positive edge inside a single batch with a large number of negative samples. Such a ranking-based approach provides the following advantages: (I) A large number of negative samples provides a better estimate of the models' precision, (II) using only one positive sample (with a single timestamp) per batch alleviates the problems of varying and tunable batch sizes, and (III) there is no information loss since no positive edges (with potentially different timestamps) are grouped. While this mitigates three out of the four problems in Figure 1, the problem of information leakage could become more severe because multiple edges with the same timestamps are assigned to their own batch. Most importantly, ranking-based evaluation results in a substantially longer runtime, as it introduces a large number of additional negative samples. In contrast, our window-based approach enables the parallel processing of many positive samples in each time window, leading to a considerably faster evaluation. Moreover, we argue that a window-based forecast is closer to real-world applications, which commonly consider forecasts for a fixed time horizon that has lower temporal resolution than the data. For example, it may not be required to predict whether a customer will purchase a certain product within the next second; making such a prediction for the next day or week may be sufficient. A promising direction for future work is combining the advantages of both approaches. In particular, ranking-based metrics that include multiple positive samples, such as the Hits@K metric (Hu et al., 2020), could be extended to the link forecasting task.

Furthermore, our reformulation of the dynamic link prediction task suggests a time-window-based model training approach, which was not considered in our experiments. We hypothesise that a window-based training approach changes model performance in two ways: First, memory-based models may perform better since they can learn from the actual temporal patterns in the data rather than from leaked information (see Appendix H.4 showing that memory-based models learn leaked information). Second, we expect the models to become more robust against bursty patterns. With a window-based training algorithm, the gradients for updating the model weights will be accumulated for each time window instead of for each equally sized batch. This means each time window will have an equally large influence on the model weights, which we expect to result in more consistent model performance across time. We leave the investigation of the expected changes to model performance introduced by a window-based training to future work and note that a misalignment between the training objective and the downstream task is common practice in model evaluation, for example, in dynamic link prediction with ranking-based metrics (You et al., 2022).

## Broader Impact Statement

We believe that a more practice-oriented, window-based evaluation of dynamic link forecasting methods, as proposed in this work, is crucial for the adoption of proposed models in real-world applications and for investigating the potential ethical implications of existing models.

## Acknowledgments

Moritz Lampert acknowledges funding from the Federal Ministry of Research, Technology and Space in Germany, Grant No. 100582863 (TissueNet). Ingo Scholtes and Christopher Blöcker acknowledge funding from the Federal Ministry of Research, Technology and Space in Germany, via the project "Software Campus 3.0" (FKZ: 01IS24030). Ingo Scholtes and Christopher Blöcker acknowledge funding through the Swiss National Science Foundation, Grant No. 176938.

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

# A  Negative Sampling Approaches

## A.1  Definitions

Dynamic link prediction is typically framed as a binary classification problem to predict class 1 for existing links during a certain time window and 0 otherwise. Due to the sparsity of most real-world graphs, it usually suffices to train and evaluate using all existing (positive) edges and a sample of non-existing (negative) edges out of all possible edges $V^2$. In static link prediction, negative edges are typically sampled randomly from $V^2$ without replacement, but Poursafaei et al. (2022) showed that this technique is ill-suited for dynamic link prediction. One reason is rooted in the characteristics of temporal graphs, where already-seen interactions tend to repeat several times during the observation period. To address this issue, Poursafaei et al. (2022) introduced negative sampling, which we cover in the following.

Let $t_{test}$ be the time that splits the temporal edges $E$ into training and test sets. We define $E_{\text{train}} = \{(u_j, v_j) | \exists (u_j, v_j, t_j) \in E : t_j < t_{test}\}$ and $E_{\text{test}} = \{(u_j, v_j) | \exists (u_j, v_j, t_j) \in E : t_j \geq t_{test}\}$ that contain the edges occurring before and after the split time $t_{test}$, respectively. We define the following commonly used sampling strategies for drawing negative samples $B_i^-$ corresponding to a set of positive samples $B_i^+ := \{(u_j, v_j, t_j) \mid \exists (u_j, v_j, t_j) \in E : b \cdot i < j \leq b \cdot (i+1)\}$ with $|B_i^+| = |B_i^-|$ (Poursafaei et al., 2022; Yu et al., 2023).

- **Random:** Sample $B_i^-$ from $V^2$ without replacement and assign the timestamps according to the positive samples $B_i^+$. The subgraph corresponding to $B_i^+$ is assumed to be sparse, making it unlikely to sample a positive edge $e \in B_i$ as negative.

- **Historic:** Sample the set of negative samples $B_i^-$ from a set of historical edges $E_{\text{hist}}$ without replacement with timestamp assignment according to the positive samples $B_i^+$. The historical edges $E_{\text{hist}}$ consist of all training edges except the ones appearing at the same time as the edges in $B_i^+$:

$$E_{\text{hist}} = E_{\text{train}} \setminus \left\{ (u_j, v_j) | \exists (u_j, v_j, t_j) \in E : t_{b \cdot i} < t_j \leq t_{b \cdot (i+1)} \right\}. \tag{1}$$

  If $|E_{\text{hist}}| < |B_i^+|$, draw the remaining edges randomly as described above.

- **Inductive:** Sample $B_i^-$ without replacement from a set of inductive edges $E_{\text{ind}}$ with timestamp assignment according to the positive samples $B_i^+$. Inductive edges $E_{\text{ind}}$ are all test set edges that have not occurred in the training set and do not occur at the same time as edges in $B_i^+$:

$$E_{\text{ind}} = E_{\text{test}} \setminus (E_{\text{train}} \cup \left\{ (u_j, v_j) | \exists (u_j, v_j, t_j) \in E : t_{b \cdot i} < t_j \leq t_{b \cdot (i+1)} \right\}). \tag{2}$$

  If $|E_{\text{ind}}| < |B_i^+|$, draw the remaining edges randomly as described above.

Note that we leave out the validation set $E_{\text{val}}$ for simplicity. Negative edges for $E_{\text{val}}$ can be sampled as for $E_{\text{test}}$.

## A.2  Influence of Link Forecasting on Negative Sampling

The above definitions of historic and inductive negative sampling exclude edges from the same time as positive edges from each batch or window from the edge sets to sample from. The sampled negative edges may be different for both approaches since the edges in the batches and windows differ. Different negative samples could explain the performance differences we observed in the experiments in Section 4. We investigate how much the negative samples differ between the two approaches in the following.

Figure 8 counts the number of edges that could be sampled for each batch and time window. For most datasets, the plots show straight lines for historic and inductive negative sampling. By definition (see Equations (1) and (2)), the negative samples are essentially generated from the set of train and test edges, respectively, but excluding the edges in the current batch or window. Consequently, the sets of available negative edges for sampling are nearly identical across all batches or windows, resulting in the observed straight-line patterns in the plots. Moreover, the number of possible negative samples is nearly identical

for the window-based and the batch-based approaches in all continuous-time datasets. For discrete-time datasets, the number of possible negative edges is identical in all points where both approaches share the same $x$-coordinate. The batch-based approach has additional points which are a small distance below the linear interpolation between the points of the window-based method. This behaviour is caused by batches that contain edges from two different snapshots, resulting in more edges that need to be excluded from the set of possible negative samples. In particular, all edges occurring at the same time as any edge in the batch have to be excluded as per the definition of historic and inductive negative sampling. Thus, our window-based approach excludes only edges within the given window, whereas the batch-based approach must exclude edges from one or two snapshots.

Overall, Figure 8 shows that the number of possible negative samples is very similar for all batches and windows across datasets, and we expect the generated negative samples to be very similar for both approaches.

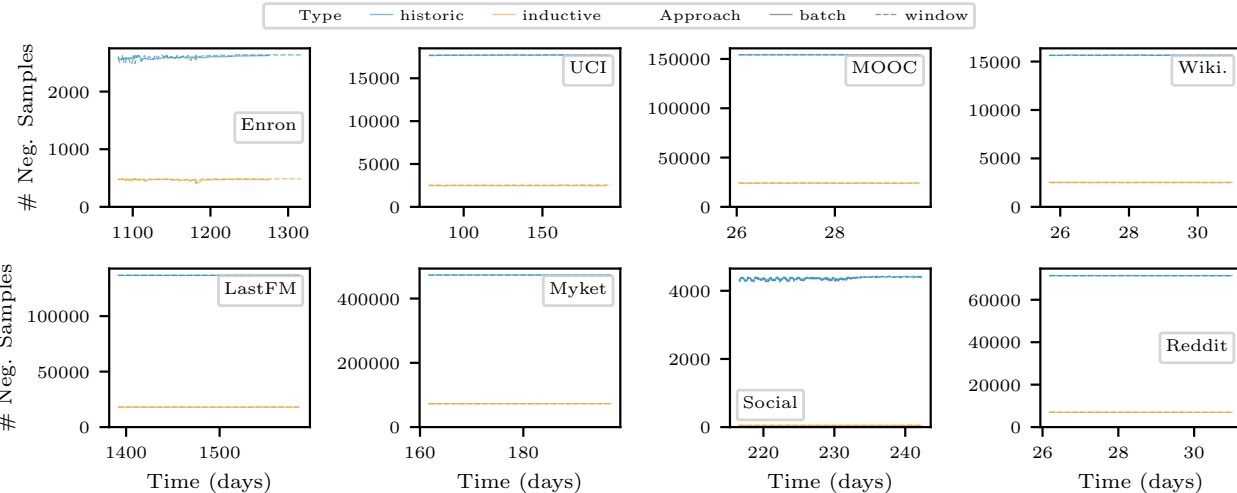

(a) Number of edges that can be sampled as negative samples for all continuous-time datasets. The $x$-axis counts the time in days starting with the first edge occurrence in the training set. Note that for some datasets, the lines for the different approaches overlay each other.

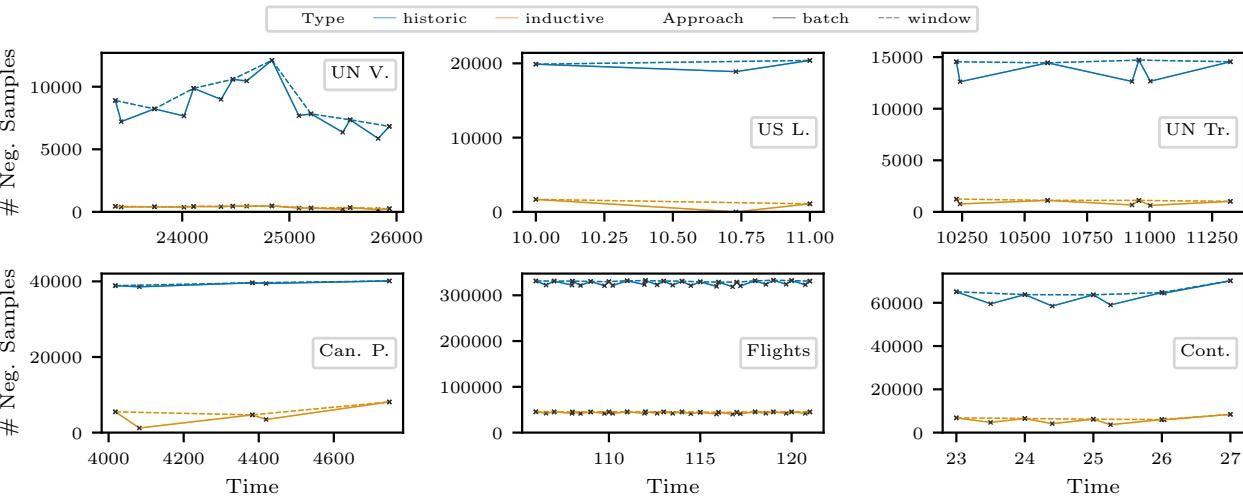

(b) Number of edges that can be sampled as negative samples for all discrete-time datasets. Since the snapshots capture different time ranges for each dataset (see Table 1), each dataset has its own temporal resolution counted from the beginning of the training set.

Figure 8: Number of edges that can be sampled as negative samples for each batch and each time window. The $x$-axis shows the arithmetic mean of all timestamps inside each batch/window from the test set.

## B   Normalised Mutual Information

Normalised mutual information is an information-theoretic measure based on mutual information. It is based on mutual information, which for two random variables $X$ and $Y$ captures the bits of information we gain about the outcome of $Y$ if we know the outcome of $X$ and vice versa. A formal definition is given in the following:

**Definition B.1** (Mutual Information)**.** Consider two random variables, $X$ and $Y$ with joint probability mass function $p(x, y)$ and marginal probability mass functions $p(x)$ and $p(y)$. Mutual information is the reduction in the uncertainty of $X$ due to the knowledge of $Y$ defined as (Cover & Thomas, 2006)

$$I(X, Y) = \sum_{x \in X} \sum_{y \in Y} p(x, y) \log \frac{p(x, y)}{p(x) p(y)}.$$

Note that mutual information can be defined using different logarithms. The intuitive understanding described above using bits of information is defined using $\log_2$. This work utilises the implementation of the Python library scikit-learn (Pedregosa et al., 2011), which uses the natural logarithm $\log_e$.

The specific value of mutual information depends on the entropy

$$H(X) = - \sum_{x \in X} p(x) \log p(x)$$

of the underlying random variables, and is thus difficult to compare across different settings. To address this issue, normalised mutual information provides a measure between zero and one that is normalised based on the entropies of the underlying random variables (Vinh et al., 2010). We use the following normalisation as implemented in scikit-learn's function `normalized_mutual_info_score`:

$$\text{NMI}(X, Y) = \frac{I(X, Y)}{\frac{1}{2} \cdot (H(X) + H(Y))}$$

In the context of our work, we use the NMI to capture the loss of temporal information. In Figure 2, we use the NMI to measure how much information about the edges' timestamps is lost by grouping the edges into batches. Specifically, we use the number $i$ of each batch $B_i^+ \cup B_i^-$ of the definition of dynamic link prediction in Section 2 assigned to each edge in the corresponding batch as one random variable and the timestamps of the edges as the other. With this setup, we can measure how much information about the timestamps of edges we gain – or keep – based on the batch number only.
Additionally, we use the NMI in Table 3 to quantify the difference between the assignments of edges to time windows and batches, respectively. Similar to above, we assign to each edge in the test set of each dataset its batch number and also a time window number corresponding to the time window the edge belongs to and then compute the NMI between those two variables.

In the following, we present an educational example that illustrates the usage of NMI in the context of our work to facilitate comprehension among readers who may not be familiar with the measure:

**Example B.1.** Let $G$ be a temporal graph with nodes $V = \{a, b, c\}$ and temporal edges

$$E = ((a, b, 1), (b, c, 2), (c, a, 2), (a, c, 4), (a, b, 5), (b, a, 5)).$$

We define the timestamps $t_i$ of the edges $e_i \in E$, the batch number for $b = 2$ and the window number with $h = 1$ as random variables $X, Y$ and $Z$, respectively. We obtain the following realizations of $X, Y$ and $Z$:

$$\mathbf{x} = (1, 2, 2, 4, 5, 5)$$
$$\mathbf{y} = (1, 1, 2, 2, 3, 3)$$
$$\mathbf{z} = (1, 2, 2, 3, 4, 4)$$

In the next step, we compute the marginal probability mass functions $p(X = x), p(Y = y)$ and $p(Z = z)$:

$$p(X = 1) = \frac{1}{6}, \qquad p(X = 2) = \frac{2}{6}, \qquad p(X = 4) = \frac{1}{6}, \qquad p(X = 5) = \frac{2}{6},$$

$$p(Y = 1) = \frac{2}{6}, \qquad p(Y = 2) = \frac{2}{6}, \qquad p(Y = 3) = \frac{2}{6},$$

$$p(Z = 1) = \frac{1}{6}, \qquad p(Z = 2) = \frac{2}{6}, \qquad p(Z = 3) = \frac{1}{6}, \qquad p(Z = 4) = \frac{2}{6}.$$

Afterward, we compute the joint probability mass functions $p(X = x, Y = y)$, $p(X = x, Z = z)$, and $p(Y = y, Z = z)$:

$$p(X = 1, Y = 1) = \frac{1}{6}, \qquad\qquad p(X = 2, Y = 1) = \frac{1}{6}, \qquad\qquad \cdots$$

$$p(X = 1, Z = 1) = \frac{1}{6}, \qquad\qquad p(X = 2, Z = 2) = \frac{2}{6}, \qquad\qquad \cdots$$

$$p(Y = 1, Z = 1) = \frac{1}{6}, \qquad\qquad p(Y = 1, Z = 2) = \frac{1}{6}, \qquad\qquad \cdots$$

Using these marginal and joint probabilities, we compute the mutual information $I(X, Y)$, $I(X, Z)$, and $I(Y, Z)$ using the formula from above:

$$I(X, Y) = \frac{1}{6} \log_e \frac{\frac{1}{6}}{\frac{1}{6} \cdot \frac{2}{6}} + \frac{1}{6} \log_e \frac{\frac{1}{6}}{\frac{2}{6} \cdot \frac{2}{6}} + \cdots \approx 0.868$$

$$I(X, Z) = \frac{1}{6} \log_e \frac{\frac{1}{6}}{\frac{1}{6} \cdot \frac{1}{6}} + \frac{2}{6} \log_e \frac{\frac{2}{6}}{\frac{2}{6} \cdot \frac{2}{6}} + \cdots \approx 1.330$$

$$I(Y, Z) = \frac{1}{6} \log_e \frac{\frac{1}{6}}{\frac{2}{6} \cdot \frac{1}{6}} + \frac{1}{6} \log_e \frac{\frac{1}{6}}{\frac{2}{6} \cdot \frac{2}{6}} + \cdots \approx 0.868$$

Finally, we obtain the normalised mutual information (NMI) by normalising based on the entropies of both random variables:

$$\mathrm{NMI}(X, Y) = \frac{0.868}{\frac{1}{2}(1.330 + 1.099)} \approx 0.715$$

$$\mathrm{NMI}(X, Z) = \frac{1.330}{\frac{1}{2}(1.330 + 1.330)} \approx 1.0$$

$$\mathrm{NMI}(Y, Z) = \frac{0.868}{\frac{1}{2}(1.099 + 1.330)} \approx 0.715$$

Note that the NMI calculations in the main part of the paper are done accordingly. Table 3 reports NMI values that follow the example calculation of $\mathrm{NMI}(Y, Z)$.

## C  Datasets

In this work, we use eight continuous-time and six discrete-time datasets, listed in Table 1. In the following, we describe what systems were observed to create the datasets and analyse the data in more detail.

### C.1  Descriptions

We list details about each dataset in the following:[3]

- **Enron** (Shetty & Adibi, 2004) is a bipartite continuous-time graph where nodes are users and the temporal edges represent emails sent between users. Emails with multiple recipients are recorded as separate and simultaneously occurring edges, one per recipient. The temporal edges are resolved at the second level, and the dataset spans approximately 3.6 years.

- **UCI** (Panzarasa et al., 2009) is a unipartite continuous-time social network dataset from an online platform at the University of California at Irvine. The nodes represent students, and the time-stamped edges represent communication between the students. The dataset spans approximately six and a half months.

- **MOOC** (massive open online course) (Kumar et al., 2019) is a bipartite continuous-time graph where nodes represent users and units in an online course, such as problems or videos. Temporal edges are resolved at the second level and encode when a user interacts with a unit of the online course. The dataset spans approximately one month.

- **Wikipedia** (Wiki.) (Kumar et al., 2019) is a bipartite continuous-time graph where nodes represent editors and Wikipedia articles. The time-stamped edges are resolved at the second level and represent when an editor has edited an article. The dataset spans approximately one month.

- **LastFM** is a bipartite continuous-time graph where nodes represent users and songs. Temporal edges are resolved at the second level and model the users' listening behaviour and represent when a user has listened to a song. The dataset was originally published by Celma (2010) and later filtered by Kumar et al. (2019) for use in a temporal graph learning context.

- **Myket**[4] (Loghmani & Fazli, 2023) is a bipartite continuous-time graph where nodes represent users and Android applications. The time-stamped edges represent when a user installed an application. The dataset spans approximately six and a half months.

- **Social Evolution** (Social) (Madan et al., 2012) is a unipartite continuous-time graph of the proximity between the students in a dormitory, collected between October 2008 and May 2009 using mobile phones. Temporal edges connect students when they are in proximity and are resolved in seconds.

- **Reddit** (Kumar et al., 2019) is a bipartite continuous-time graph where nodes represent Reddit users and their posts. The time-stamped edges are resolved in seconds and represent when a user has made a post on Reddit. The dataset spans approximately one month.

- **UN Vote** (UN V.)(Voeten et al., 2009; Poursafaei et al., 2022) is a weighted unipartite discrete-time graph of votes in the United Nations General Assembly between 1946 and 2020. Nodes represent countries and edges connect countries if they both vote "yes". The dataset is resolved at the year level, and edge weights represent how many times the two connected countries have both voted "yes" in the same vote.

- **US Legislators** (US L.) (Huang et al., 2020; Fowler, 2006; Poursafaei et al., 2022) is a weighted unipartite discrete-time graph of interactions between legislators in the US Senate. Nodes represent legislators and edges represent co-sponsorship, i.e., edges connect legislators who co-sponsor the same bill. The dataset is resolved at the congress level, and edge weights encode the number of co-sponsorships during a congress.

---

[3]Each dataset is part of the Zenodo package provided by Poursafaei et al. (2022) under the Creative Commons Attribution 4.0 International (CC BY 4.0) license, if no other license information is provided.
[4]License: Attribution-NonCommercial 4.0 International (CC BY-NC 4.0)

- **UN Trade** (UN Tr.) (MacDonald et al., 2015; Poursafaei et al., 2022) is a directed and weighted unipartite discrete-time graph of food and agricultural trade between countries, where nodes represent countries. The dataset spans 30 years and is resolved at the year level. Weighted edges encode the sum of normalised agriculture imports or exports between two countries during a given year.

- **Canadian Parliament** (Can. P.) (Huang et al., 2020; Poursafaei et al., 2022) is a weighted unipartite discrete-time political network where nodes represent Members of the Canadian Parliament (MPs) and an edge between two MPs means that they have both voted "yes" on a bill. The dataset is resolved at the year level, and the edges' weights represent how often the two connected MPs voted "yes" on the same bill during a year.

- **Flights** (Schäfer et al., 2014; Poursafaei et al., 2022) is a directed and weighted unipartite discrete-time graph where nodes represent airports and edges represent flights during the COVID-19 pandemic. The edges are resolved at the day level, and their weights are given by the number of flights between two airports during the respective day.

- **Contacts** (Cont.) (Sapiezynski et al., 2019; Poursafaei et al., 2022) is a unipartite discrete-time proximity network between university students. Nodes represent students who are connected by an edge if they were in close proximity during a time window. The dataset is resolved at the 5-minute level and spans one month.

## C.2  Edge Activity patterns

Similar to Figure 3, we show the link densities over time for all datasets in Figure 9. While some datasets, such as Enron, UCI, and LastFM, contain very inhomogeneously distributed activity, others exhibit recurring or seasonal patterns, e.g. MOOC, Wikipedia, and Reddit. In addition to seasonality, we also observe trends like a growing number of edges in the Myket, UN Vote, and UN Trade datasets, which are also often found in standard time series data (Brockwell & Davis, 2016).

## C.3  Time Window Durations for Different Batch Sizes

In Figure 10, we evaluate the dependency between batch size and time window for empirical temporal graphs. The figure shows the distribution of batch durations (y-axis) for different batch sizes (x-axis). We observe that, both in continuous- and discrete-time temporal graphs, a single batch size can create time windows with varying durations even within the same dataset. For continuous-time temporal graphs, we typically have much bigger batches than edges per timestamp, such that the time window defined by the batches becomes long (cf. Table 1). The number of edges per snapshot in discrete-time temporal graphs is generally larger than the batch size $b$ in any period, regardless of the density (Table 1). This means that edges in a batch often belong to the same snapshot, leading to small window durations.

## C.4  Number of Links per Timestamp

In Figure 11, we show how often each number of links with the same timestamp (x-axis) occurs in each dataset. We can observe that all continuous-time datasets have a large number of edges that appear at a unique point in time. Additionally, the number of edges that occur at once for many continuous-time datasets is less than ten (MOOC, Wiki., Myket, Reddit) and less than 50 for most datasets, except for Enron, which has outliers with more than 1000 links at a certain timestamp. For discrete-time datasets, many edges occur in each snapshot.

## C.5  Window Sizes for Different Window Durations

Figure 12 shows the distribution of the number of links per time window for continuous- and discrete-time data sets. We can see that the distribution of window sizes depends on the distribution of edges across time as visualised in Figure 9. Datasets with a more uniform activity pattern, such as Wikipedia and Reddit, show a similar number of edges across windows of the same duration. Datasets with inhomogeneously distributed activities, e.g. Enron or UCI, exhibit a wide range of window sizes. For large values of $h$, i.e. horizons that

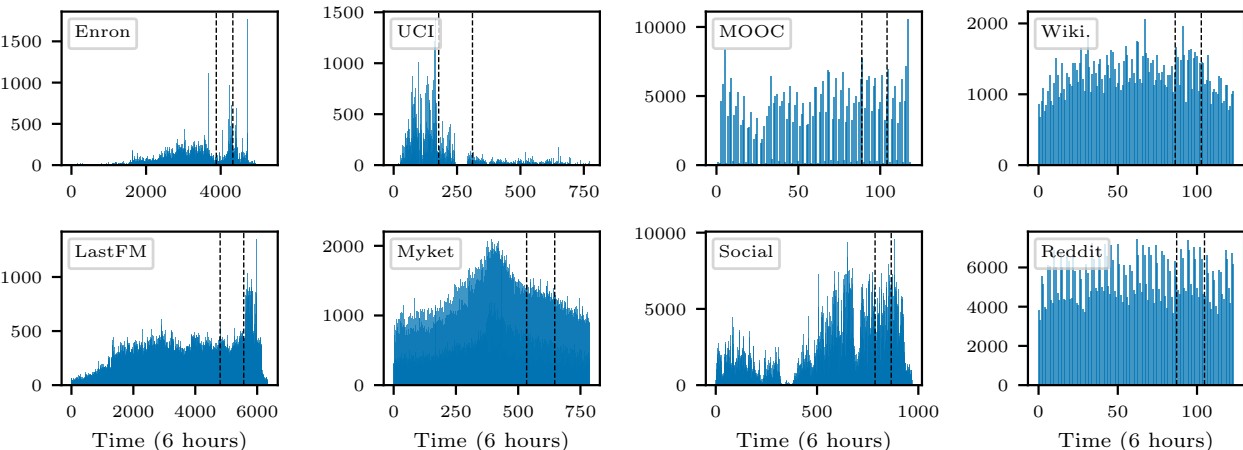

(a) Continuous-time temporal graphs resolved in seconds. For visualisation purposes, edges are binned into 6-hour time periods and then represented as one bar in the histogram.

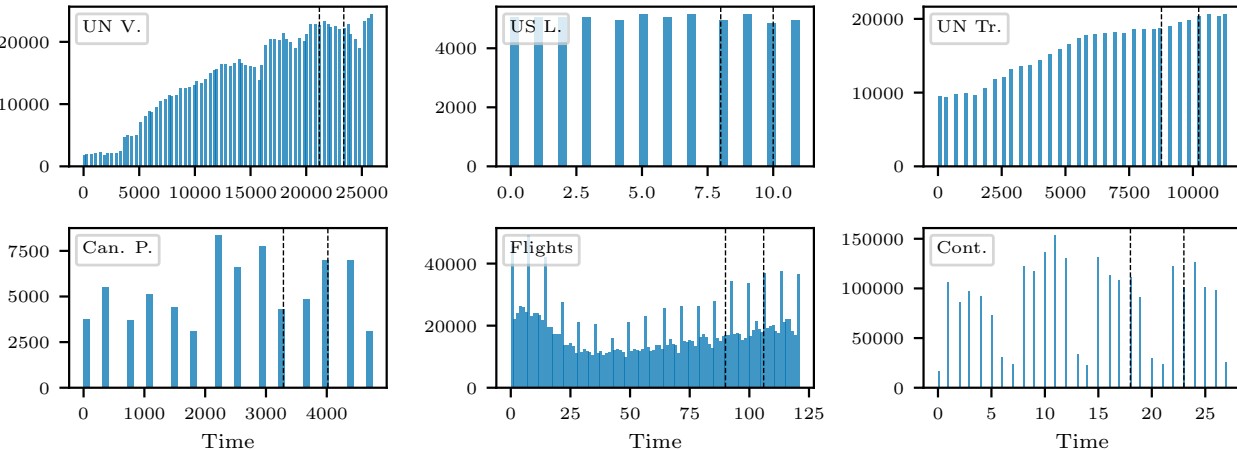

(b) Discrete-time temporal graphs where each bar in the histogram corresponds to one snapshot. Since the snapshots capture different time ranges for each dataset (see Table 1), each dataset has its own temporal resolution.

Figure 9: Real-world datasets exhibit diverse edge occurrence patterns that are visualised using the edge density across time, i.e., histograms counting the number of edges per timestamp. Dashed lines divide the datasets into 70% train, 15% validation, and 15% test sets as used in Section 4.

include more than half of the observed time, the window sizes diverge because the window of the last time window that still includes observed links gets smaller.

Figure 13 shows the histograms of window sizes for the horizons used in the experiments of Section 4 (see Table 3). We observe that continuous-time datasets with only small performance differences (in particular, Wikipedia and Reddit) between our window-based and batch-based evaluations exhibit a narrow distribution of window sizes. Datasets with larger changes in performance (MOOC, LastFM, UCI, and Enron) have an increasingly longer tail. This supports our hypothesis that a batch-based evaluation results in skewed model performance estimates for temporal datasets with inhomogeneous temporal activity, that is, datasets with long-tailed window-size distributions.

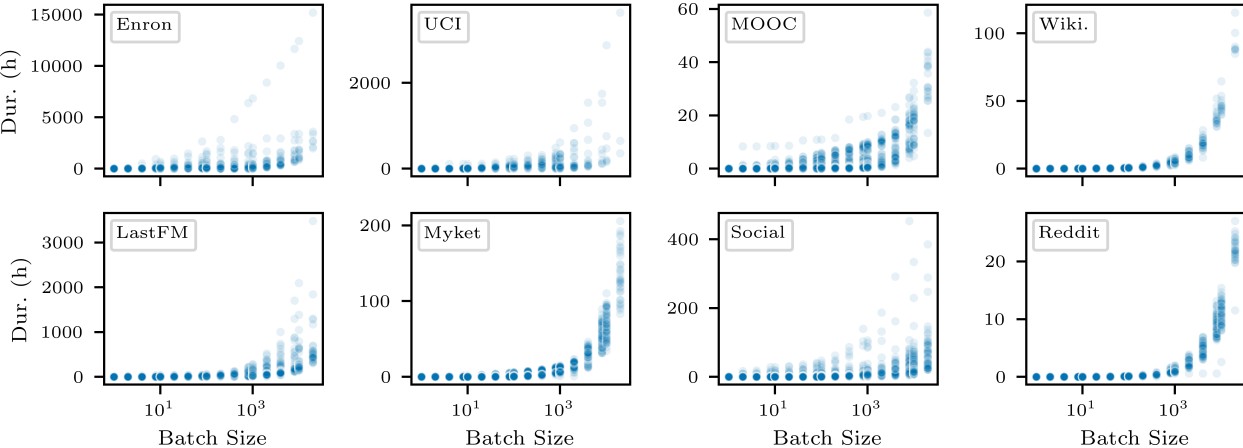

(a) Continuous-time temporal graphs: Batch size $b$ determines the average time window length. However, a single batch size creates time windows with various lengths within and across datasets.

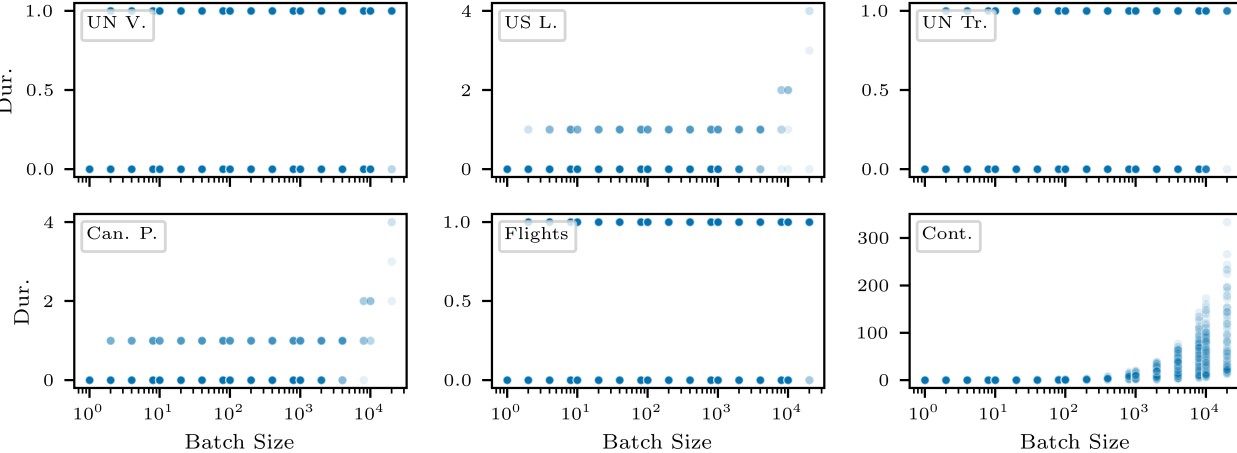

(b) Discrete-time temporal graphs: Fixed-size batches fall mostly within snapshots when the batches are much smaller than the snapshots. Depending on the dataset, larger batches can also span across many snapshots.

Figure 10: Using a low opacity value for individual points, the distribution of time window durations is shown for different batch sizes. I.e. points appear less see-through with an increasing number of points with the same duration and batch size stacked on top of each other.

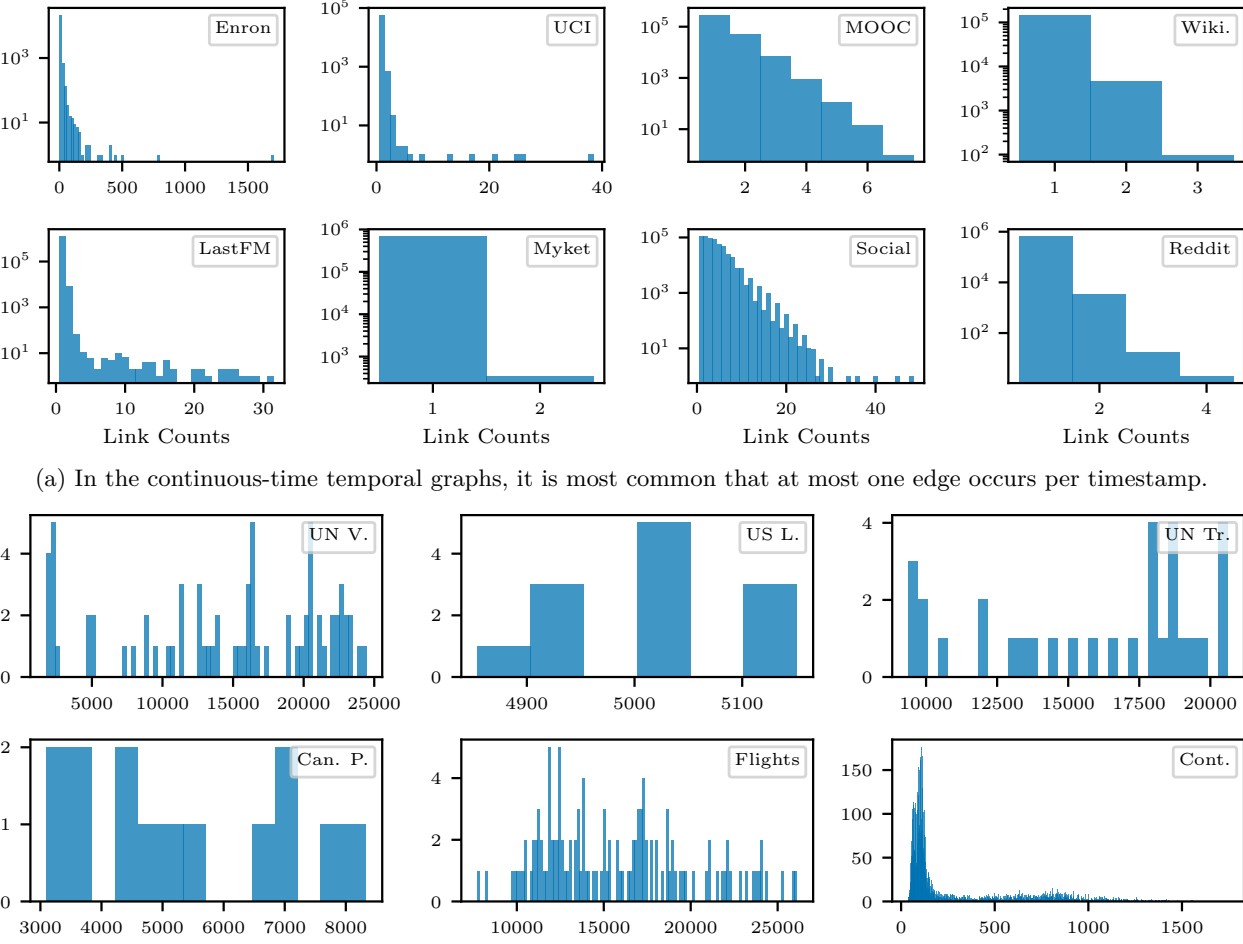

(a) In the continuous-time temporal graphs, it is most common that at most one edge occurs per timestamp.

(b) The snapshots in discrete-time temporal graphs contain large numbers of edges, typically much larger than commonly utilised batch sizes. The Contacts dataset has fewer links per snapshot due to its much higher resolution than the remaining discrete-time datasets.

Figure 11: The link count histograms show how many edges occur per timestamp in continuous-time temporal graphs and per snapshot in discrete-time temporal graphs, respectively.

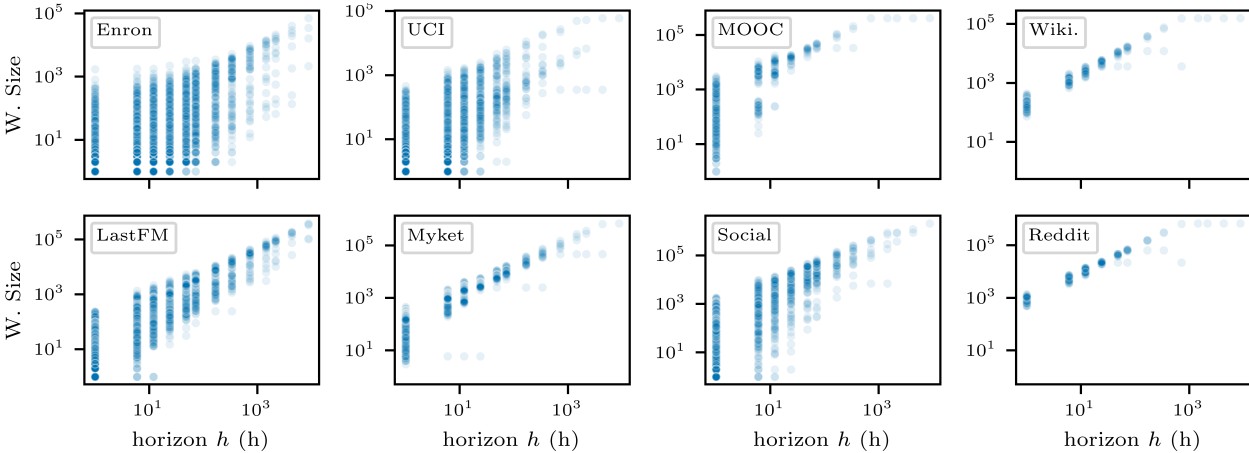

(a) The number of links per window in continuous-time temporal graphs for horizons ranging from one second to one year. The $x$-axis is labelled in hours.

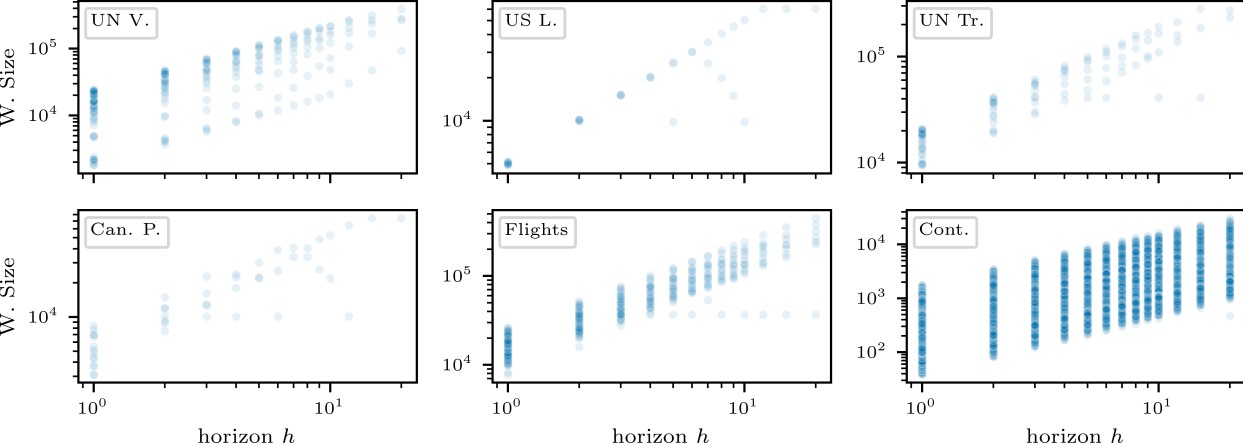

(b) Window sizes for discrete-time temporal graphs where $h$ represents the number of snapshots.

Figure 12: Window sizes, i.e. number of links per time window, for different horizons.

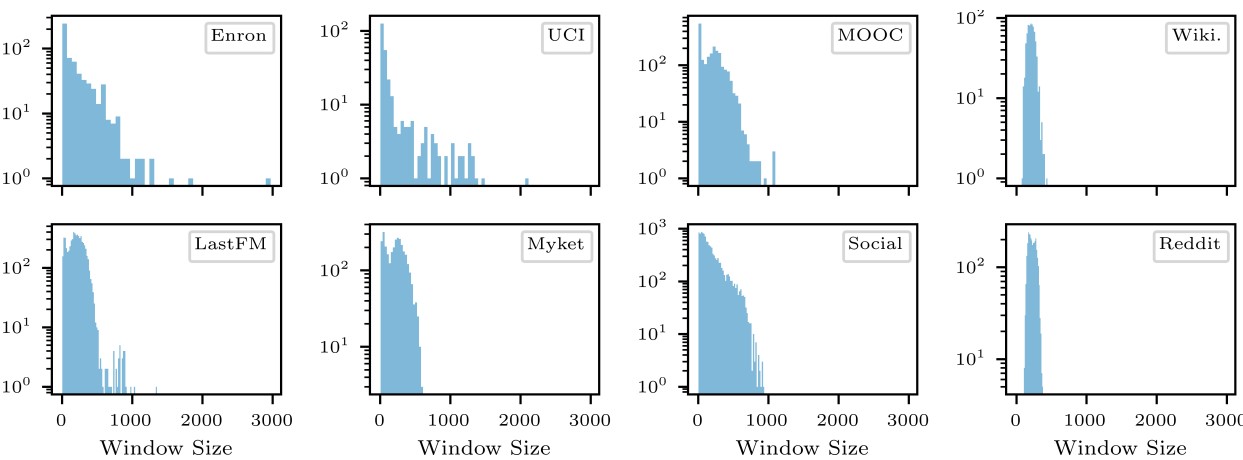

Figure 13: Histogram of window sizes, i.e. number of links per time window, for continuous-time temporal graphs.

# D   Link Prediction using Different Batch Sizes

The results in Table 4 and 5 show the differences in AUC-ROC and average precision scores using two different batch sizes. While the same models with the same trained weights are used for both evaluations, the results show substantial differences in performance. This is in line with our expectations as described in Problem (d).

As expected due to Problem (a), the performance differences are especially large for memory-based models, and the performance drops almost consistently for all datasets when using a larger batch size. Other models show no clear preference for whether larger or smaller batch sizes lead to better performance. This drop in performance for memory-based models specifically can be explained because they update their internal node representations after processing each batch, thus, larger batches lead to less frequent updates and more "out-of-date" internal representations (Su et al., 2024; Feldman & Baskin, 2024). In other words, larger batches lead to a larger information loss because more edges at the beginning of each batch are missing in the internal representation for the prediction of edges at the end of each batch.

Table 4: Mean AUC-ROC performance for dynamic link prediction ($b = 800$) over five runs for the continuous- (top) and discrete-time (bottom) datasets. Values in parentheses show the relative change compared to an evaluation using batch size $b = 200$.

| Dataset | JODIE | DyRep | TGN | TGAT | CAWN | EdgeBank | TCL | GraphMixer | DyGFormer | $\mu \pm \sigma$ |
|---|---|---|---|---|---|---|---|---|---|---|
| Enron | 74.8(↓3.4%) | 67.9(↓7.7%) | 68.4(↑0.5%) | 59.4(↑1.2%) | 67.1(↑1.1%) | 78.5(↓1.7%) | 68.4(↑1.1%) | 82.2(↑1.1%) | 77.5(↑1.4%) | 2.1%±2.2% |
| UCI | 79.9(↓4.1%) | 47.7(↓7.3%) | 61.0(↓3.3%) | 59.2(↓0.7%) | 57.6(↓0.9%) | 64.6(↓6.6%) | 59.6(↓0.6%) | 79.9(↓0.9%) | 76.0(↓0.2%) | 2.7%±2.7% |
| MOOC | 82.0(↓3.4%) | 77.9(↓3.5%) | 87.5(↓1.1%) | 82.5(↑0.2%) | 70.5(↑0.1%) | 57.2(↓7.6%) | 72.6(↓0.0%) | 74.5(↑0.1%) | 81.3(↑0.1%) | 1.8%±2.6% |
| Wiki. | 77.4(↓5.4%) | 74.5(↓4.9%) | 83.7(↓0.5%) | 83.5(↑0.1%) | 72.0(↑0.5%) | 76.5(↓0.8%) | 85.2(↑0.1%) | 88.0(↑0.2%) | 80.4(↑0.5%) | 1.4%±2.1% |
| LastFM | 76.5(↓2.0%) | 70.4(↓1.8%) | 80.1(↓0.8%) | 68.4(↑0.0%) | 67.9(↓0.2%) | 77.4(↓1.0%) | 64.3(↑0.1%) | 65.9(↓0.1%) | 78.9(↓0.0%) | 0.7%±0.8% |
| Myket | 64.4(↑0.7%) | 63.6(↓1.0%) | 60.7(↓0.7%) | 57.4(↓0.3%) | 32.6(↑0.3%) | 52.0(↑0.1%) | 58.4(↓0.0%) | 59.4(↓0.2%) | 33.0(↓0.5%) | 0.4%±0.3% |
| Social | 87.3(↓4.4%) | 88.2(↓4.8%) | 92.0(↑0.3%) | 92.7(↑0.1%) | 87.8(↓0.2%) | 86.0(↑0.2%) | 95.3(↑0.1%) | 94.1(↑0.1%) | 97.4(↑0.1%) | 1.1%±2.0% |
| Reddit | 79.0(↓1.9%) | 76.4(↓3.9%) | 80.3(↓0.1%) | 78.7(↑0.1%) | 80.3(↑0.1%) | 78.5(↓0.2%) | 76.2(↑0.0%) | 77.2(↑0.1%) | 80.2(↑0.0%) | 0.7%±1.3% |
| UN V. | 71.3(↓3.1%) | 66.9(↓7.8%) | 67.2(↓4.4%) | 52.8(↑0.0%) | 50.2(↑0.2%) | 89.6(↑0.1%) | 53.0(↓0.1%) | 56.5(↑0.4%) | 63.3(↑0.4%) | 1.8%±2.7% |
| US L. | 56.6(↑0.6%) | 73.5(↓7.9%) | 76.7(↓8.7%) | 78.6(↑0.1%) | 82.6(↑0.9%) | 67.2(↓0.4%) | 77.0(↑1.9%) | 90.3(↑0.1%) | 90.2(↑0.8%) | 2.4%±3.4% |
| UN Tr. | 65.7(↓0.6%) | 63.2(↑0.1%) | 61.5(↓2.6%) | 61.4(↓0.5%) | 64.6(↓0.2%) | 86.5(↑0.1%) | 60.9(↓0.1%) | 66.2(↓0.1%) | 68.2(↓0.2%) | 0.5%±0.8% |
| Can. P. | 62.2(↓2.7%) | 64.6(↓3.0%) | 72.9(↓0.7%) | 71.1(↓0.7%) | 67.9(↓0.2%) | 62.1(↓1.4%) | 68.2(↑0.0%) | 81.0(↓0.2%) | 93.3(↓4.5%) | 1.5%±1.6% |
| Flights | 68.9(↓0.9%) | 68.4(↓0.9%) | 68.5(↓0.4%) | 72.7(↑0.1%) | 66.0(↑0.0%) | 74.6(↑0.0%) | 70.6(↓0.0%) | 70.7(↓0.0%) | 68.7(↓0.0%) | 0.3%±0.4% |
| Cont. | 93.4(↓2.2%) | 93.3(↓2.2%) | 95.5(↓0.6%) | 95.5(↑0.1%) | 83.3(↑0.1%) | 91.7(↓0.5%) | 94.6(↑0.1%) | 94.2(↑0.1%) | 97.2(↑0.1%) | 0.7%±0.9% |
| $\mu \pm \sigma$ | 2.5%±1.5% | 4.1%±2.8% | 1.8%±2.4% | 0.3%±0.4% | 0.4%±0.4% | 1.5%±2.4% | 0.3%±0.6% | 0.3%±0.3% | 0.6%±1.2% | |

Table 5: Mean AP performance for dynamic link prediction ($b = 800$) over five runs for the continuous- (top) and discrete-time (bottom) datasets. Values in parentheses show the relative change compared to an evaluation using batch size $b = 200$.

| Dataset | JODIE | DyRep | TGN | TGAT | CAWN | EdgeBank | TCL | GraphMixer | DyGFormer | $\mu \pm \sigma$ |
|---|---|---|---|---|---|---|---|---|---|---|
| Enron | 68.2(↓5.4%) | 62.4(↓10.4%) | 65.2(↓0.2%) | 63.2(↑0.1%) | 66.8(↑1.2%) | 75.8(↓1.5%) | 70.7(↑0.6%) | 83.1(↑0.9%) | 77.2(↑1.0%) | 2.4%±3.4% |
| UCI | 77.5(↓4.5%) | 46.3(↓5.5%) | 70.1(↓1.5%) | 68.3(↓0.4%) | 65.2(↑0.1%) | 61.0(↓6.2%) | 68.9(↓0.2%) | 85.5(↓0.4%) | 81.1(↑0.6%) | 2.2%±2.5% |
| MOOC | 80.8(↓3.1%) | 74.7(↓3.1%) | 85.6(↓1.6%) | 84.6(↑0.1%) | 73.5(↑0.0%) | 56.1(↓7.5%) | 78.3(↓0.2%) | 77.8(↓0.1%) | 82.4(↑0.0%) | 1.7%±2.5% |
| Wiki. | 79.7(↓5.3%) | 77.3(↓4.3%) | 88.5(↓0.3%) | 87.9(↑0.1%) | 75.4(↑0.6%) | 72.6(↓0.7%) | 89.6(↑0.1%) | 91.3(↑0.2%) | 83.6(↑0.7%) | 1.4%±2.0% |
| LastFM | 75.3(↓3.0%) | 68.6(↓3.9%) | 80.1(↓0.3%) | 75.2(↑0.3%) | 69.8(↑0.0%) | 72.5(↓1.0%) | 71.9(↑0.3%) | 74.2(↑0.2%) | 81.2(↑0.2%) | 1.0%±1.4% |
| Myket | 63.7(↑0.4%) | 61.0(↓2.4%) | 61.6(↓0.8%) | 56.6(↓0.9%) | 45.0(↓0.3%) | 51.2(↓0.1%) | 57.9(↓0.7%) | 58.8(↓0.6%) | 44.6(↓0.3%) | 0.7%±0.7% |
| Social | 83.2(↓5.8%) | 85.7(↓6.5%) | 93.5(↑0.3%) | 94.9(↑0.1%) | 86.3(↑0.1%) | 80.8(↑0.3%) | 96.3(↑0.1%) | 95.5(↑0.0%) | 97.5(↑0.2%) | 1.5%±2.7% |
| Reddit | 78.3(↓2.3%) | 75.7(↓4.4%) | 80.3(↓0.3%) | 78.5(↓0.2%) | 81.1(↓0.0%) | 73.5(↓0.2%) | 76.5(↓0.1%) | 77.4(↓0.1%) | 82.7(↓0.0%) | 0.8%±1.5% |
| UN V. | 65.1(↓4.0%) | 62.2(↓8.3%) | 63.1(↓4.5%) | 51.8(↓0.8%) | 50.3(↓0.7%) | 84.9(↑0.1%) | 52.9(↓0.7%) | 53.7(↓0.5%) | 60.0(↑0.0%) | 2.2%±2.8% |
| US L. | 48.2(↑0.0%) | 67.8(↓7.2%) | 74.4(↓8.3%) | 70.8(↓0.6%) | 81.3(↑0.7%) | 63.1(↓0.3%) | 78.8(↑1.8%) | 86.2(↑0.2%) | 86.7(↑1.1%) | 2.2%±3.2% |
| UN Tr. | 57.9(↓1.7%) | 58.5(↓1.2%) | 57.2(↓2.8%) | 57.2(↓1.2%) | 57.4(↓0.8%) | 81.1(↑0.1%) | 55.7(↓0.9%) | 63.7(↓0.0%) | 64.5(↑0.4%) | 1.0%±0.9% |
| Can. P. | 51.4(↓2.6%) | 60.4(↓2.2%) | 69.1(↑0.8%) | 68.1(↑0.6%) | 64.5(↑0.9%) | 61.7(↓3.3%) | 65.0(↑1.2%) | 78.9(↑2.2%) | 93.1(↓4.1%) | 2.0%±1.2% |
| Flights | 65.8(↓1.4%) | 65.7(↓1.5%) | 68.2(↓0.2%) | 73.0(↑0.4%) | 64.8(↓0.3%) | 70.4(↓0.1%) | 70.8(↓0.0%) | 71.4(↑0.3%) | 68.4(↓0.2%) | 0.5%±0.6% |
| Cont. | 91.4(↓3.0%) | 93.3(↓2.3%) | 95.6(↓0.6%) | 96.1(↑0.1%) | 84.5(↑0.1%) | 88.4(↓0.5%) | 95.1(↑0.1%) | 94.4(↑0.1%) | 97.8(↑0.1%) | 0.8%±1.1% |
| $\mu \pm \sigma$ | 3.0%±1.8% | 4.5%±2.8% | 1.6%±2.3% | 0.4%±0.4% | 0.4%±0.4% | 1.6%±2.4% | 0.5%±0.5% | 0.4%±0.6% | 0.6%±1.1% | |

# E   Evaluation based on Realistic Time Horizons

In our main experiments, we determined the time horizon based on the average number of links per time window to enable a fair comparison to the batch-based approach. In Section 3.1, we discussed that the time horizon for a realistic evaluation should instead be chosen carefully for each individual dataset. In the following, we provide guidelines for selecting the time horizon in practice and, as an example of such a realistic evaluation, assign a reasonable horizon for each dataset used in this work.

## E.1   Guidelines for Selecting a Good Time Horizon

Three factors restrict the choice of possible time horizons for an application domain: (Q1) Which time horizons are reasonable in the context of the task? (Q2) Which time horizons fit the characteristics of the dataset, and (Q3) Which time horizons are possible given the available computational resources?

The first question (Q1) should be the primary focus for the selection. We list typical ranges for common application domains of link forecasting models in the following:

- **Recommender systems:** Recommender systems are employed in a multitude of different areas, and depending on the area and the type of recommendation to give, the time horizons can change drastically (Wang et al., 2021b). Session-based recommendations are made on timescales from seconds to hours, e.g., for a single browser session. Recommendations for short-term preferences use a time horizon of days to weeks and usually model evolving preferences, in which a user, for example, researches a specific product category before purchasing a product from that category. Lastly, recommendations for long-term or static preferences use a time horizon of months or years and capture user behaviour, such as a preference for buying products from a certain brand.

- **Traffic forecasting:** Similar time ranges as for recommender systems also exist for traffic forecasting, depending on the goal of the forecast. It can range from seconds to minutes for real-time management or incident detection, from minutes to hours for traffic control systems, or from days to months for infrastructure planning (Mystakidis et al., 2025).

- **Opportunistic networks:** The time granularity for forecasts in opportunistic networks is usually in the range of minutes (Shu et al., 2022) to hours (Gou & Wu, 2022; Huang et al., 2015).

Note that even within the same application domain, e.g., traffic forecasting, the time range is highly task-dependent, making it difficult to provide general recommendations. Therefore, we select appropriate time horizons for all datasets used in this paper's experiments and explain the rationale for our choices, providing examples to guide time horizon selection in other scenarios in Appendix E.2.

Given a reasonable time horizon for an application, the second question (Q2) must additionally be considered. In particular, the results in Section 4 show that especially bursty or highly irregular temporal edge-occurrence patterns lead to skewed model performance when using batch-based evaluation. We further observe that the model's performance for individual time windows can vary substantially with the window size, that is, the number of edges per window. Therefore, an evaluation can highlight the model's ability to handle irregular edge-occurrence patterns by choosing a time horizon that reflects the dataset's non-uniform patterns in its window-size distribution. On the other hand, if we are not interested in the model's robustness to irregular edge-occurrence patterns in the dataset, we can choose a time horizon that "destroys" these patterns. For example, the window size distribution would be uniform for a dataset with a seasonal pattern if we choose the period of the seasonal pattern as the time horizon. Additionally, the task context might have already been a deciding factor in the dataset's temporal resolution during its creation, making the dataset's temporal resolution a natural choice for the time horizon.

Given the time horizons possible under (Q1) and (Q2), we must additionally consider (Q3) to determine the final time horizon. Depending on the dataset's temporal edge density, the time horizon should usually be a multiple of the dataset's timestamp resolution to enable parallel processing for many edges. Datasets with high temporal resolution (continuous-time) are often sparse—where many timestamps have no or only

a single edge—requiring a large time horizon in terms of multiples of the resolution to enable high GPU utilisation. Ideally, the time horizon required by the task enables high parallelisation for continuous-time temporal graphs. If this is not the case, one needs to compromise between computational requirements and how well the horizon fits the task. The temporal resolution for discrete-time datasets (low resolution) often represents a natural choice for the time horizon because each snapshot already contains enough edges for high parallelisation. Note that we suggest mitigation strategies in Appendix G if the chosen time horizon contains too many edges and causes memory overflows.

### E.2 Selecting Exemplary Time Horizons

We use 14 real-world datasets to evaluate our approach in Section 4. For each dataset, we assign an appropriate horizon. Note that this enables a fair model comparison since we use the same horizon for all models across each dataset. The temporal resolution of most discrete-time datasets is limited by the data collection process. The US Legislators dataset, for example, contains one snapshot for each Congress, which provides a natural horizon of one snapshot. Thus, we will only consider continuous-time datasets and discrete-time datasets where the duration of a snapshot is not yet a natural time horizon for evaluation, i.e. Contacts.

We list the considered datasets and the chosen horizon, with the reasoning behind it as follows:

- **Enron** is a network of users with edges representing emails sent between them. We choose 24 hours as a reasonable horizon since 90% of all email replies are typically sent within a day (Kooti et al., 2015).

- **UCI** is a social network based on student communications. We use 30 min as the horizon since users receiving a text message typically feel pressured to reply between the next 20 minutes and the end of the day (Aranda & Baig, 2018).

- **MOOC** connects students to units of an online course based on their interactions. We set $h = 6$ min since Guo et al. (2014) recommend keeping the learning video of a unit shorter than this time frame.

- **Wikipedia** represents the editing behaviour of users in a graph. Since editing Wikipedia articles is unpaid, we do not expect frequent interactions from each user. We assume that users come from different time zones and consider that the total duration of the dataset is only one month. Therefore, we see the typical working time of 8 hours as an appropriate time horizon to take into account that interactions won't appear very frequently, but there are still enough time windows for evaluation.

- **LastFM** connects users to the songs they listen to. We select 24 hours as the horizon to evaluate this dataset on a task where the goal is to predict the songs that users will listen to tomorrow based on their listening behaviour during the last days.

- **Myket** represents users and Android applications that are connected when an application is installed. Considering that, similar to the Wikipedia dataset, no frequent interactions are expected, we choose 24 hours as the time horizon since the total duration of the dataset is longer than the total duration of the Wikipedia dataset.

- **Social Evolution** is a proximity network gathered from students in a dormitory. Thus, we select 2 hours – a typical duration of a lecture including breaks – as the time horizon.

- **Reddit** contains the posting behaviour of Reddit users. Since dynamics in a social network are typically fast, we select a time horizon of 15 minutes.

- **Contacts:** Similar to Social Evolution, we select 2 hours as the time horizon since the network also captures the proximity of students.

We use the trained models from the experiments of the main part of our work and reevaluate the specified datasets with the selected time horizons. The results are presented in Table 6 and 7. For completeness,

Table 6: Average AUC-ROC performance and standard deviation over five runs using the trained models from the above experiments and the window-based evaluation with the horizons specified above. The tables also include the datasets that have not been reevaluated using performance scores obtained from the main experiments' time horizons. The best performance score for each dataset is highlighted in bold.

| Datasets | JODIE | DyRep | TGN | TGAT | CAWN | EdgeBank | TCL | GraphMixer | DyGFormer |
|---|---|---|---|---|---|---|---|---|---|
| Enron | 84.2 ± 4.8 | 80.4 ± 2.5 | 70.0 ± 4.5 | 68.3 ± 1.8 | 75.5 ± 0.4 | 83.1 ± 0.0 | 74.2 ± 4.8 | **87.9 ± 0.6** | 83.9 ± 0.6 |
| UCI | **89.4 ± 0.7** | 70.4 ± 2.8 | 64.0 ± 1.3 | 52.2 ± 1.1 | 56.0 ± 0.2 | 75.3 ± 0.0 | 55.3 ± 1.3 | 82.7 ± 0.9 | 75.4 ± 0.4 |
| MOOC | 84.6 ± 4.1 | 81.3 ± 3.3 | **87.6 ± 1.3** | 80.7 ± 0.9 | 69.3 ± 1.5 | 62.9 ± 0.0 | 68.1 ± 0.9 | 70.4 ± 1.6 | 79.9 ± 10.0 |
| Wiki. | 75.5 ± 0.8 | 73.2 ± 0.9 | 83.5 ± 0.6 | 83.5 ± 0.2 | 71.9 ± 0.8 | 76.1 ± 0.0 | 85.2 ± 0.6 | **87.9 ± 0.3** | 80.6 ± 1.6 |
| LastFM | 74.9 ± 2.1 | 67.5 ± 1.4 | 78.6 ± 2.9 | 66.0 ± 0.9 | 67.2 ± 0.3 | 77.5 ± 0.0 | 63.2 ± 6.7 | 60.6 ± 1.3 | **78.9 ± 0.6** |
| Myket | **64.5 ± 1.8** | 62.7 ± 3.1 | 60.4 ± 2.3 | 57.8 ± 0.4 | 32.2 ± 0.4 | 51.4 ± 0.0 | 58.3 ± 2.0 | 59.9 ± 0.3 | 32.8 ± 0.9 |
| Social | 87.0 ± 2.0 | 84.4 ± 4.7 | 92.3 ± 2.7 | 92.6 ± 0.5 | 86.7 ± 0.0 | 85.1 ± 0.0 | 94.7 ± 0.5 | 94.7 ± 0.2 | **97.4 ± 0.1** |
| Reddit | **80.6 ± 0.1** | 79.5 ± 0.8 | 80.4 ± 0.4 | 78.6 ± 0.7 | 80.2 ± 0.3 | 78.6 ± 0.0 | 76.2 ± 0.4 | 77.1 ± 0.4 | 80.2 ± 1.1 |
| Avg. Rank | 3.0 | 5.2 | 3.8 | 6.1 | 6.9 | 6.0 | 6.1 | 4.2 | 3.6 |
| UN V. | 54.0 ± 1.8 | 52.2 ± 2.0 | 51.3 ± 7.1 | 54.4 ± 3.6 | 53.7 ± 2.1 | **89.6 ± 0.0** | 53.4 ± 1.0 | 56.9 ± 1.6 | 65.2 ± 1.1 |
| US L. | 52.5 ± 1.8 | 61.8 ± 3.5 | 57.7 ± 1.8 | 78.6 ± 7.9 | 82.0 ± 4.0 | 68.4 ± 0.0 | 75.4 ± 5.3 | **90.4 ± 1.5** | 89.4 ± 0.9 |
| UN Tr. | 57.7 ± 3.3 | 50.3 ± 1.4 | 54.3 ± 1.5 | 64.0 ± 1.3 | 67.6 ± 1.2 | **85.6 ± 0.0** | 63.7 ± 1.6 | 68.6 ± 2.6 | 70.7 ± 2.6 |
| Can. P. | 63.6 ± 0.8 | 67.5 ± 8.5 | 73.2 ± 1.1 | 72.7 ± 2.2 | 70.0 ± 1.4 | 63.2 ± 0.0 | 69.5 ± 3.1 | 80.7 ± 0.9 | **85.5 ± 3.5** |
| Flights | 67.4 ± 2.0 | 66.0 ± 1.9 | 68.1 ± 1.7 | 72.6 ± 0.2 | 66.0 ± 1.7 | **74.6 ± 0.0** | 70.6 ± 0.1 | 70.7 ± 0.3 | 68.8 ± 1.0 |
| Cont. | 85.4 ± 0.5 | 74.5 ± 3.1 | 94.6 ± 0.6 | 96.0 ± 0.2 | 86.6 ± 0.1 | 85.8 ± 0.0 | 95.7 ± 0.4 | 95.2 ± 0.2 | **97.8 ± 0.0** |
| Avg. Rank | 7.3 | 8.0 | 6.5 | 3.5 | 5.5 | 4.2 | 5.2 | 2.7 | 2.2 |

both tables also include the performance scores of the discrete-time datasets that have not been reevaluated because we determined the horizon used above as realistic.

The results using both the AUC-ROC as well as the average precision score mostly agree on a best-performing model, yet for different datasets, there is no clear winner among the models. For continuous-time datasets, JODIE, GraphMixer, and DyGFormer are among the best-performing models, while EdgeBank, GraphMixer, and DyGFormer performed best for discrete-time data.

DyGFormer performs best for both Contacts and Social Evolution, suggesting that DyGFormer is best suited for proximity networks among all models. For UN Vote, UN Trade, and Flights, EdgeBank – a simple baseline model that predicts an edge if it has occurred before – is among the best. These datasets are highly repetitive because they are, e.g. based on a schedule or relations among countries that rarely change. Thus, since they are all outperformed by simple baselines, none of the proposed TGNN models adequately address the task of these datasets, i.e. finding edges that do not follow the schedule or some other recurring pattern. For other types of datasets like communication (e.g. Enron or UCI) or user-interaction networks (e.g. Wikipedia or MOOC), no clear patterns are visible.

Table 7: Mean average precision scores and standard deviation following Table 6.

| Datasets | JODIE | DyRep | TGN | TGAT | CAWN | EdgeBank | TCL | GraphMixer | DyGFormer |
|---|---|---|---|---|---|---|---|---|---|
| Enron | 82.2 ± 4.8 | 80.0 ± 3.4 | 71.2 ± 4.2 | 72.6 ± 1.3 | 77.1 ± 0.3 | 81.6 ± 0.0 | 77.9 ± 2.6 | **89.2 ± 0.4** | 85.0 ± 0.6 |
| UCI | **93.0 ± 0.5** | 81.5 ± 1.9 | 78.0 ± 0.9 | 71.6 ± 0.7 | 73.3 ± 0.2 | 75.6 ± 0.0 | 73.2 ± 0.7 | 89.6 ± 0.5 | 85.2 ± 0.3 |
| MOOC | 84.1 ± 5.4 | 79.4 ± 3.8 | **87.4 ± 1.6** | 83.9 ± 0.8 | 73.8 ± 1.1 | 62.1 ± 0.0 | 75.8 ± 0.5 | 75.6 ± 0.9 | 82.6 ± 8.9 |
| Wiki. | 78.3 ± 1.2 | 76.4 ± 0.7 | 88.3 ± 0.4 | 88.0 ± 0.2 | 75.7 ± 1.1 | 72.4 ± 0.0 | 89.7 ± 0.4 | **91.3 ± 0.2** | 84.0 ± 1.2 |
| LastFM | 73.6 ± 2.0 | 64.8 ± 1.8 | 78.6 ± 3.7 | 73.0 ± 0.8 | 69.2 ± 0.5 | 72.9 ± 0.0 | 71.2 ± 6.7 | 71.0 ± 1.1 | **81.3 ± 0.9** |
| Myket | **62.9 ± 1.5** | 60.2 ± 1.9 | 61.1 ± 1.7 | 56.8 ± 0.4 | 44.9 ± 0.2 | 51.1 ± 0.0 | 57.8 ± 2.2 | 59.0 ± 0.2 | 44.4 ± 1.7 |
| Social | 82.5 ± 4.0 | 81.6 ± 5.7 | 94.0 ± 1.8 | 95.0 ± 0.3 | 85.8 ± 0.1 | 79.9 ± 0.0 | 96.2 ± 0.3 | 95.8 ± 0.2 | **97.8 ± 0.1** |
| Reddit | 80.1 ± 0.3 | 79.2 ± 0.9 | 80.6 ± 0.6 | 78.6 ± 1.0 | 81.3 ± 0.4 | 73.5 ± 0.0 | 76.5 ± 0.6 | 77.5 ± 0.5 | **82.8 ± 0.8** |
| Avg. Rank | 3.4 | 5.8 | 3.8 | 5.5 | 6.8 | 7.2 | 5.4 | 4.0 | 3.2 |
| UN V. | 52.6 ± 1.8 | 49.6 ± 1.9 | 49.7 ± 3.9 | 52.7 ± 2.6 | 52.4 ± 2.0 | **84.2 ± 0.0** | 52.4 ± 0.9 | 54.0 ± 1.4 | 62.4 ± 1.7 |
| US L. | 46.0 ± 0.9 | 62.5 ± 3.6 | 58.6 ± 2.4 | 71.0 ± 8.9 | 80.7 ± 3.7 | 63.2 ± 0.0 | 77.5 ± 4.3 | **86.5 ± 1.9** | 86.1 ± 1.0 |
| UN Tr. | 52.7 ± 3.0 | 49.4 ± 0.9 | 53.2 ± 1.5 | 58.9 ± 2.7 | 59.2 ± 1.7 | **79.0 ± 0.0** | 57.5 ± 1.9 | 65.8 ± 1.9 | 67.1 ± 2.7 |
| Can. P. | 52.1 ± 0.5 | 61.0 ± 7.6 | 69.9 ± 0.8 | 70.8 ± 1.6 | 68.3 ± 2.3 | 59.4 ± 0.0 | 68.2 ± 1.6 | 80.9 ± 0.5 | **83.2 ± 2.9** |
| Flights | 65.2 ± 2.7 | 63.9 ± 2.8 | 68.3 ± 2.2 | **73.5 ± 0.3** | 65.5 ± 1.3 | 70.4 ± 0.0 | 71.0 ± 0.4 | 71.9 ± 0.8 | 69.1 ± 1.8 |
| Cont. | 83.3 ± 0.7 | 69.7 ± 3.7 | 95.0 ± 0.8 | 96.9 ± 0.2 | 88.1 ± 0.2 | 82.3 ± 0.0 | 96.7 ± 0.5 | 95.8 ± 0.1 | **98.5 ± 0.0** |
| Avg. Rank | 7.7 | 8.3 | 6.3 | 3.3 | 5.2 | 4.7 | 4.8 | 2.5 | 2.2 |

Table 8: Test AUC-ROC scores for link forecasting with realistic horizons and the relative performance change compared to link prediction using the trained models from the experiments in Section 4. The table compares the AUC-ROC scores for all datasets that have been reevaluated on a different horizon. The last row/column provides the mean $\mu$ and standard deviation $\sigma$ of the absolute values (without sign) of the relative change per column/row.

| Model | JODIE | DyRep | TGN | TGAT | CAWN | EdgeBank | TCL | GraphMixer | DyGFormer | $\mu \pm \sigma$ |
|---|---|---|---|---|---|---|---|---|---|---|
| Enron | 84.2(↑8.8%) | 80.4(↑9.3%) | 70.0(↑2.8%) | 68.3(↑16.4%) | 75.5(↑13.7%) | 83.1(↑4.1%) | 74.2(↑9.7%) | 87.9(↑8.2%) | 83.9(↑9.8%) | 9.2%±4.2% |
| UCI | 89.4(↑7.3%) | 70.4(↑36.9%) | 64.0(↑1.5%) | 52.2(↓12.5%) | 56.0(↓3.8%) | 75.3(↑8.9%) | 55.3(↓7.8%) | 82.7(↑2.5%) | 75.4(↓1.0%) | 9.1%±11.1% |
| MOOC | 84.6(↓0.2%) | 81.3(↑0.7%) | 87.6(↓1.0%) | 80.7(↓1.9%) | 69.3(↓1.5%) | 62.9(↑1.6%) | 68.1(↓6.3%) | 70.4(↓5.4%) | 79.9(↓1.6%) | 2.2%±2.1% |
| Wiki. | 75.5(↓7.8%) | 73.2(↓6.6%) | 83.5(↓0.8%) | 83.5(↑0.1%) | 71.9(↑0.4%) | 76.1(↓1.3%) | 85.2(↑0.1%) | 87.9(↑0.0%) | 80.6(↑0.7%) | 2.0%±3.0% |
| LastFM | 74.9(↓4.0%) | 67.5(↓5.7%) | 78.6(↓2.6%) | 66.0(↓3.5%) | 67.2(↓1.3%) | 77.5(↓0.9%) | 63.2(↓1.6%) | 60.6(↓8.1%) | 78.9(↓0.1%) | 3.1%±2.6% |
| Myket | 64.5(↑0.8%) | 62.7(↓2.4%) | 60.4(↓1.1%) | 57.8(↑0.3%) | 32.2(↓0.8%) | 51.4(↓1.0%) | 58.3(↓0.2%) | 59.9(↑0.7%) | 32.8(↑0.0%) | 0.8%±0.7% |
| Social | 87.0(↓4.7%) | 84.4(↓9.0%) | 92.3(↑0.7%) | 92.6(↓0.1%) | 86.7(↓1.2%) | 85.1(↓0.9%) | 94.7(↓0.5%) | 94.7(↑0.6%) | 97.4(↑0.1%) | 2.0%±3.0% |
| Reddit | 80.6(↓0.0%) | 79.5(↑0.0%) | 80.4(↓0.0%) | 78.6(↓0.1%) | 80.2(↓0.0%) | 78.6(↓0.1%) | 76.2(↓0.1%) | 77.1(↓0.1%) | 80.2(↑0.0%) | 0.0%±0.1% |
| Cont. | 85.4(↓10.7%) | 74.5(↓21.9%) | 94.6(↓1.5%) | 96.0(↑0.6%) | 86.6(↑4.0%) | 85.8(↓7.0%) | 95.7(↑1.3%) | 95.2(↑1.2%) | 97.8(↑0.7%) | 5.4%±7.1% |
| $\mu \pm \sigma$ | 4.9%±4.0% | 10.3%±11.9% | 1.3%±0.9% | 3.9%±6.1% | 3.0%±4.3% | 2.9%±3.1% | 3.1%±3.8% | 3.0%±3.4% | 1.6%±3.1% | |

Table 9: Test average precision scores for link forecasting with realistic horizons and the relative performance change compared to link prediction using the trained models from the experiments in Section 4. The table compares the average precision scores for all datasets that have been reevaluated on a different horizon. The last row/column provides the mean $\mu$ and standard deviation $\sigma$ of the absolute values (without sign) of the relative change per column/row.

| Model | JODIE | DyRep | TGN | TGAT | CAWN | EdgeBank | TCL | GraphMixer | DyGFormer | $\mu \pm \sigma$ |
|---|---|---|---|---|---|---|---|---|---|---|
| Enron | 82.2(↑14.0%) | 80.0(↑14.7%) | 71.2(↑9.1%) | 72.6(↑14.9%) | 77.1(↑16.8%) | 81.6(↑6.1%) | 77.9(↑11.0%) | 89.2(↑8.4%) | 85.0(↑11.2%) | 11.8%±3.5% |
| UCI | 93.0(↑14.6%) | 81.5(↑66.5%) | 78.0(↑9.6%) | 71.6(↑4.3%) | 73.3(↑12.6%) | 75.6(↑16.3%) | 73.2(↑6.0%) | 89.6(↑4.3%) | 85.2(↑5.7%) | 15.5%±19.6% |
| MOOC | 84.1(↑0.9%) | 79.4(↑2.9%) | 87.4(↑0.5%) | 83.9(↓0.7%) | 73.8(↑0.4%) | 62.1(↑2.3%) | 75.8(↓3.4%) | 75.6(↓3.0%) | 82.6(↑0.2%) | 1.6%±1.3% |
| Wiki. | 78.3(↓6.9%) | 76.4(↓5.5%) | 88.3(↓0.5%) | 88.0(↑0.1%) | 75.7(↑0.9%) | 72.4(↓0.9%) | 89.7(↑0.2%) | 91.3(↑0.1%) | 84.0(↑1.2%) | 1.8%±2.5% |
| LastFM | 73.6(↓5.1%) | 64.8(↓9.2%) | 78.6(↓2.2%) | 73.0(↓2.7%) | 69.2(↓0.9%) | 72.9(↓0.4%) | 71.2(↓0.6%) | 71.0(↓4.1%) | 81.3(↑0.3%) | 2.8%±2.9% |
| Myket | 62.9(↑0.8%) | 60.2(↓3.8%) | 61.1(↓1.6%) | 56.8(↓0.6%) | 44.9(↓0.5%) | 51.1(↓0.3%) | 57.8(↓0.9%) | 59.0(↓0.3%) | 44.4(↓0.5%) | 1.0%±1.1% |
| Social | 82.5(↓6.6%) | 81.6(↓10.9%) | 94.0(↑0.9%) | 95.0(↑0.2%) | 85.8(↓0.4%) | 79.9(↓0.9%) | 96.2(↑0.0%) | 95.8(↑0.4%) | 97.8(↑0.5%) | 2.3%±3.8% |
| Reddit | 80.1(↓0.0%) | 79.2(↑0.0%) | 80.6(↑0.0%) | 78.6(↓0.1%) | 81.3(↑0.2%) | 73.5(↓0.2%) | 76.5(↓0.0%) | 77.5(↓0.0%) | 82.8(↑0.1%) | 0.1%±0.1% |
| Cont. | 83.3(↓11.6%) | 69.7(↓27.0%) | 95.0(↓1.3%) | 96.9(↑0.9%) | 88.1(↑4.4%) | 82.3(↓7.4%) | 96.7(↑1.8%) | 95.8(↑1.7%) | 98.5(↑0.8%) | 6.3%±8.6% |
| $\mu \pm \sigma$ | 6.7%±5.7% | 15.6%±20.7% | 2.9%±3.7% | 2.7%±4.8% | 4.1%±6.2% | 3.9%±5.4% | 2.7%±3.7% | 2.5%±2.8% | 2.3%±3.8% | |

In Table 8 and 9, the differences in performance between link forecasting with a realistic horizon and link prediction are reported, similar to Table 13 in Section 4. The results show that for most datasets, the difference in performance between the current link prediction evaluation and our proposed link forecasting is even larger when choosing a realistic horizon based on the application scenario. This emphasises that the model performance is skewed if the models are evaluated with link prediction without taking the application scenario into account.

# F    Experimental Details

For reproducibility, we provide a Python package extending the dynamic graph learning library DyGLib[5] (Yu et al., 2023) as a supplement, including a bash script to run the experiments[6].

We use the best hyperparameters reported by Yu et al. (2023) and, for completeness, list these hyperparameters for the 13 datasets used by Yu et al. (2023) below. However, the Myket dataset (Loghmani & Fazli, 2023) was not included in the study. Therefore, for Myket, we use each method's default parameters as suggested by the respective authors.

We use 9 state-of-the-art dynamic graph learning models and baselines (JODIE (Kumar et al., 2019), DyRep (Trivedi et al., 2019), TGAT (Xu et al., 2020), TGN (Rossi et al., 2020), CAWN (Wang et al., 2021c), Edge-Bank (Poursafaei et al., 2022), TCL (Wang et al., 2021a), GraphMixer (Cong et al., 2023) and DyGFormer (Yu et al., 2023)). The neural-network-based approaches (all except EdgeBank) are trained five times for 100 epochs using the Adam optimiser with a learning rate of 0.0001. An early-stopping strategy with a patience of 5 is employed to avoid overfitting. For training and validation, a batch size of 200 is used. The training, validation and test sets of each dataset contain 70%, 15% and 15% of the edges, respectively. The sets are split based on time, i.e., the training set contains the edges that occurred first, while the test set comprises the most recent edges.

The experiments were conducted on a variety of machines with different CPUs and GPUs. A list of machine specifications is provided in Table 10.

Table 10: Hardware details of the machines used for the experiments.

(a) CPUs

| CPU |
| --- |
| AMD Ryzen Threadripper PRO 5965WX 24 Cores |
| AMD Ryzen 9 7900X 12 Cores |
| 11th Gen Intel(R) Core(TM) i9-11900K 8 Cores |
| AMD Ryzen 9 7950X 16 Cores |
| 13th Gen Intel(R) Core(TM) i9-13900H 14 Cores |
| AMD Epyc 7543 (3rd Gen) |
| AMD Epyc 7763 (3rd Gen) |
| AMD Epyc 7713 (3rd Gen) |
| Intel Xeon Platinum 8480+ (4th Gen) |

(b) GPUs

| GPU |
| --- |
| NVIDIA GeForce RTX 3090 Ti |
| NVIDIA GeForce RTX 4080 |
| NVIDIA GeForce RTX 3090 |
| NVIDIA GeForce RTX 4090 |
| NVIDIA GeForce RTX 4060 (Laptop) |
| NVIDIA A100 |
| NVIDIA GeForce RTX 2080 Ti |
| NVIDIA TITAN Xp |
| NVIDIA TITAN X |
| NVIDIA Quadro RTX 8000 |
| NVIDIA H100 |
| NVIDIA L40 |
| NVIDIA L40s |

For all model architectures, time-related representations use a size of 100 dimensions, while all other non-time-related representations are set to 172. An exception is DyGFormer, where the neighbour co-occurrence encoding and the aligned encoding each have 50 dimensions. We use eight attention heads for CAWN, and two attention heads for all other attention-based methods. The memory-based models either use a vanilla recurrent neural network (JODIE and DyRep), or a gated recurrent unit (GRU) to update their memory. Other model-specific parameters are provided in Table 11 .

---

[5] https://github.com/yule-BUAA/DyGLib (MIT License)
[6] https://github.com/M-Lampert/DyGLib/blob/master/run_experiments.sh

Table 11: Specific hyperparameters for different models and datasets.

(a) Hyperparameters for neighbourhood sampling-based models. $n_{\text{Neighbours}}$ is the number of sampled neighbours using the specified neighbour sampling strategy. $n_{\text{L}}$ is the number of transformer layers (for TCL), the number of MLP-Mixer layers (for GraphMixer) or the number of GNN layers otherwise.

| Dataset | Model | Neigh. Sampling | $n_{\text{Neigh.}}$ | $n_{\text{L}}$ | Dropout |
|---|---|---|---|---|---|
| Wikipedia | DyRep | recent | 10 | 1 | 0.1 |
| | TGAT | recent | 20 | 2 | 0.1 |
| | TGN | recent | 10 | 1 | 0.1 |
| | TCL | recent | 20 | 2 | 0.1 |
| | GraphMixer | recent | 30 | 2 | 0.5 |
| Reddit | DyRep | recent | 10 | 1 | 0.1 |
| | TGAT | uniform | 20 | 2 | 0.1 |
| | TGN | recent | 10 | 1 | 0.1 |
| | TCL | uniform | 20 | 2 | 0.1 |
| | GraphMixer | recent | 10 | 2 | 0.5 |
| MOOC | DyRep | recent | 10 | 1 | 0.0 |
| | TGAT | recent | 20 | 2 | 0.1 |
| | TGN | recent | 10 | 1 | 0.2 |
| | TCL | recent | 20 | 2 | 0.1 |
| | GraphMixer | recent | 20 | 2 | 0.4 |
| LastFM | DyRep | recent | 10 | 1 | 0.0 |
| | TGAT | recent | 20 | 2 | 0.1 |
| | TGN | recent | 10 | 1 | 0.3 |
| | TCL | recent | 20 | 2 | 0.1 |
| | GraphMixer | recent | 10 | 2 | 0.0 |
| Enron | DyRep | recent | 10 | 1 | 0.0 |
| | TGAT | recent | 20 | 2 | 0.2 |
| | TGN | recent | 10 | 1 | 0.0 |
| | TCL | recent | 20 | 2 | 0.1 |
| | GraphMixer | recent | 20 | 2 | 0.5 |
| Social Evo. | DyRep | recent | 10 | 1 | 0.1 |
| | TGAT | recent | 20 | 2 | 0.1 |
| | TGN | recent | 10 | 1 | 0.0 |
| | TCL | recent | 20 | 2 | 0.0 |
| | GraphMixer | recent | 20 | 2 | 0.3 |
| UCI | DyRep | recent | 10 | 1 | 0.0 |
| | TGAT | recent | 20 | 2 | 0.1 |
| | TGN | recent | 10 | 1 | 0.1 |
| | TCL | recent | 20 | 2 | 0.0 |
| | GraphMixer | recent | 20 | 2 | 0.4 |
| Myket | DyRep | recent | 10 | 1 | 0.1 |
| | TGAT | recent | 20 | 2 | 0.1 |
| | TGN | recent | 10 | 1 | 0.1 |
| | TCL | recent | 20 | 2 | 0.1 |
| | GraphMixer | recent | 20 | 2 | 0.1 |
| Flights | DyRep | recent | 10 | 1 | 0.1 |
| | TGAT | recent | 20 | 2 | 0.1 |
| | TGN | recent | 10 | 1 | 0.1 |
| | TCL | recent | 20 | 2 | 0.1 |
| | GraphMixer | recent | 20 | 2 | 0.2 |
| Can. Parl. | DyRep | uniform | 10 | 1 | 0.0 |
| | TGAT | uniform | 20 | 2 | 0.2 |
| | TGN | uniform | 10 | 1 | 0.3 |
| | TCL | uniform | 20 | 2 | 0.2 |
| | GraphMixer | uniform | 20 | 2 | 0.2 |
| US Legis. | DyRep | recent | 10 | 1 | 0.0 |
| | TGAT | recent | 20 | 2 | 0.1 |
| | TGN | recent | 10 | 1 | 0.1 |
| | TCL | uniform | 20 | 2 | 0.3 |
| | GraphMixer | recent | 20 | 2 | 0.4 |
| UN Trade | DyRep | recent | 10 | 1 | 0.1 |
| | TGAT | uniform | 20 | 2 | 0.1 |
| | TGN | recent | 10 | 1 | 0.2 |
| | TCL | uniform | 20 | 2 | 0.0 |
| | GraphMixer | uniform | 20 | 2 | 0.1 |
| UN Vote | DyRep | recent | 10 | 1 | 0.1 |
| | TGAT | recent | 20 | 2 | 0.2 |
| | TGN | uniform | 10 | 1 | 0.1 |
| | TCL | uniform | 20 | 2 | 0.0 |
| | GraphMixer | uniform | 20 | 2 | 0.0 |
| Contacts | DyRep | recent | 10 | 1 | 0.0 |
| | TGAT | recent | 20 | 2 | 0.1 |
| | TGN | recent | 10 | 1 | 0.1 |
| | TCL | recent | 20 | 2 | 0.0 |
| | GraphMixer | recent | 20 | 2 | 0.1 |

(b) Hyperparameters DyGFormer.

| Dataset | Model | Sequence Length | Patch Size | Dropout |
|---|---|---|---|---|
| Wikipedia | DyGFormer | 32 | 1 | 0.1 |
| Reddit | DyGFormer | 64 | 2 | 0.2 |
| MOOC | DyGFormer | 256 | 8 | 0.1 |
| LastFM | DyGFormer | 512 | 16 | 0.1 |
| Enron | DyGFormer | 256 | 8 | 0.0 |
| Social Evo. | DyGFormer | 32 | 1 | 0.1 |
| UCI | DyGFormer | 32 | 1 | 0.1 |
| Myket | DyGFormer | 32 | 1 | 0.1 |
| Flights | DyGFormer | 256 | 8 | 0.1 |
| Can. Parl. | DyGFormer | 2048 | 64 | 0.1 |
| US Legis. | DyGFormer | 256 | 8 | 0.0 |
| UN Trade | DyGFormer | 256 | 8 | 0.0 |
| UN Vote | DyGFormer | 128 | 4 | 0.2 |
| Contacts | DyGFormer | 32 | 1 | 0.0 |

(c) Hyperparameters CAWN.

| Dataset | Model | Walk Length | Time Scale | Dropout |
|---|---|---|---|---|
| Wikipedia | CAWN | 1 | 0.000001 | 0.1 |
| Reddit | CAWN | 1 | 0.000001 | 0.1 |
| MOOC | CAWN | 1 | 0.000001 | 0.1 |
| LastFM | CAWN | 1 | 0.000001 | 0.1 |
| Enron | CAWN | 1 | 0.000001 | 0.1 |
| Social Evo. | CAWN | 1 | 0.000001 | 0.1 |
| UCI | CAWN | 1 | 0.000001 | 0.1 |
| Myket | CAWN | 1 | 0.000001 | 0.1 |
| Flights | CAWN | 1 | 0.000001 | 0.1 |
| Can. Parl. | CAWN | 1 | 0.000001 | 0.0 |
| US Legis. | CAWN | 1 | 0.000001 | 0.1 |
| UN Trade | CAWN | 1 | 0.000001 | 0.1 |
| UN Vote | CAWN | 1 | 0.000001 | 0.1 |
| Contacts | CAWN | 1 | 0.000001 | 0.1 |

(d) Hyperparameters EdgeBank

| Dataset | Model | Neg. Sampling | Memory Mode | Time Window |
|---|---|---|---|---|
| Wikipedia | EdgeBank | random | unlimited | - |
| | | historical | repeat threshold | - |
| | | inductive | repeat threshold | - |
| Reddit | EdgeBank | random | unlimited | - |
| | | historical | repeat threshold | - |
| | | inductive | repeat threshold | - |
| MOOC | EdgeBank | random | time window | fixed proportion |
| | | historical | time window | repeat interval |
| | | inductive | repeat threshold | - |
| LastFM | EdgeBank | random | time window | fixed proportion |
| | | historical | time window | repeat interval |
| | | inductive | repeat threshold | - |
| Enron | EdgeBank | random | time window | fixed proportion |
| | | historical | time window | repeat interval |
| | | inductive | repeat threshold | - |
| Social Evo. | EdgeBank | random | repeat threshold | - |
| | | historical | repeat threshold | - |
| | | inductive | repeat threshold | - |
| UCI | EdgeBank | random | unlimited | - |
| | | historical | time window | fixed proportion |
| | | inductive | time window | repeat interval |
| Myket | EdgeBank | random | unlimited | - |
| | | historical | repeat threshold | |
| | | inductive | repeat threshold | |
| Flights | EdgeBank | random | unlimited | - |
| | | historical | repeat threshold | - |
| | | inductive | repeat threshold | - |
| Can. Parl. | EdgeBank | random | time window | fixed proportion |
| | | historical | time window | fixed proportion |
| | | inductive | repeat threshold | - |
| US Legis. | EdgeBank | random | time window | fixed proportion |
| | | historical | time window | fixed proportion |
| | | inductive | time window | fixed proportion |
| UN Trade | EdgeBank | random | time window | repeat interval |
| | | historical | time window | repeat interval |
| | | inductive | repeat threshold | - |
| UN Vote | EdgeBank | random | time window | repeat interval |
| | | historical | time window | repeat interval |
| | | inductive | time window | repeat interval |
| Contacts | EdgeBank | random | time window | repeat interval |
| | | historical | time window | repeat interval |
| | | inductive | repeat threshold | - |

# G   Runtime and Memory Implications

In the following, we discuss how to avoid memory overflows in datasets with time windows with many edges. Additionally, we present the empirical runtime of the experiments in Section 4 for dynamic link prediction and forecasting averaged across five runs.

## G.1   Avoiding Memory Overflows

In highly bursty datasets, the window size can become very large during periods of extreme density. This can lead to memory overflows if the evaluation is implemented on a GPU. The easiest mitigation strategy to avoid overflows, if the runtime is not a deciding factor, is to move the evaluation to the CPU. This can mitigate the problem because CPU memory is usually larger than GPU memory, and if the CPU memory is also full, strategies such as paging or swapping can prevent overflows. While this mitigation strategy was sufficient for the experiments conducted in this work, stricter runtime requirements or larger datasets might require a faster and more reliable solution.

We explain a faster and more reliable solution without memory overflow problems that can be implemented on the GPU in the following: To enable adaptation to any model and framework, we provide a conceptual implementation with pseudocode since the specific implementation may vary across TGNNs and frameworks. We first present pseudocode for a basic implementation of our window-based evaluation approach in Listing 1 for reference. In this pseudocode, we iterate over time windows containing target temporal edges (with timestamps) that can be positive samples with label `1` or negative samples with label `0`. For each window, we first sample the neighbourhood of all nodes that are sources or destinations of the target edges based on the recent history of the temporal graph. Next, we can optionally include a memory for each node to enable the evaluation of memory-based models and then do message passing as defined by the TGNN model on the sampled recent neighbourhood. Afterwards, the resulting node representations are combined to obtain a forecast for each target temporal edge, which can be used to compute arbitrary performance metrics for each window. Lastly, the memory needs to be updated if the model uses memory.

We extend this pseudocode to avoid memory overflows by subdividing the time window into smaller chunks in Listing 2. This subdivision into smaller chunks is possible because the temporal information is coarse-grained into time windows, thereby deliberately discarding temporal information within each time window. Thus, we do not need to save any temporal information, but only need to ensure that our subdivision introduces no information leakage. We implement this subdivision with an additional loop compared to Listing 1. The outer loop encompasses all steps that could potentially leak information, including sampling the recent neighbourhood, calculating performance metrics, and updating memory. The inner loop comprises resampling to reduce the size of the recent neighbourhood graph and inferring whether an edge occurs using the TGNN model. Lastly, the results for each edge and the intermediate node representations are saved to enable the metric calculation and memory update of the whole window, combined in the outer loop.

## G.2   Empirical Runtime

We compare the empirical runtime of window-based link forecasting against the batch-based link prediction in Figure 14. The runtime of both approaches is generally comparable, with both approaches being faster than the other on some datasets and models. For continuous-time datasets, we can see that the runtime is often similar and large differences are caused by specific characteristics of the datasets. The model evaluation on UCI, for example, is faster using the batch-based approach because it consistently evaluates batches of size $b = 200$ while the time windows are mostly smaller, which leads to less parallelisation. This is because the duration of each window was chosen so that the average window size across the whole dataset is comparable to the chosen batch size of $b = 200$. For this specific dataset, there are large bursts of activity in the first third of the total dataset duration and low activity afterwards. This leads to small window sizes for the temporal edges of the test set (see Figure 9).

For discrete-time datasets, each snapshot contains more than 200 edges for most datasets, leading to better parallelisation for link forecasting and, thus, smaller runtime. However, the basic implementation of our window-based approach leads to GPU memory overflows on datasets with many edges per time windows

```python
def evaluate_tgnn(tgnn, t_graph):
    # Initialise a window loader that iterates over all windows
    test_window_loader = get_window_loader(
        t_graph,  # temporal graph for evaluation
        horizon=3600,  # specify time horizon as 1 hour
        test_ratio=0.15,  # use the last 15% of the time as test set
        n_negatives=1  # sample 1 negative edge for each positive sample
    )
    # Iterate over all time windows with positive and negative samples
    for target_t_edges, t_edge_label in test_window_loader:
        # Sample recent neighbourhood that occurs before all `target_t_edges` of the window
        neighbour_t_graph = t_graph.get_recent_neighbours(target_t_edges)
        # Get updated memory of all nodes involved in the window's computation.
        z = tgnn.get_memory(neighbour_t_graph)
        # Use the TGNN to get temporal node features
        z = tgnn(z, neighbour_t_graph)
        # Get forecast for each edge using the temporal node feature
        # of the edge's source and destination node
        out = forecast_links(z, target_t_edges)
        # Compute performance metrics using the forecast and the ground truth
        metrics = calculate_metrics(out, t_edge_label)
        # Update memory with ground-truth state
        tgnn.update_memory(z, t_edge_label)
```

Listing 1: Python-like pseudocode for a basic implementation of our window-based evaluation approach as a reference.

for some model implementations—in particular TGAT, CAWN, GraphMixer and DyGFormer. For those models, the employed mitigation strategy (running the evaluation on CPU) resulted in a larger runtime for the window-based evaluation. Table 12 shows the detailed runtime of the evaluation of all models and datasets for dynamic link prediction and forecasting.

Table 12: Mean and standard deviation in seconds of the evaluation runtime of all models for dynamic link prediction and forecasting averaged across five runs.

| Dataset | Approach | JODIE | DyRep | TGN | TGAT | CAWN | EdgeBank | TCL | GraphMixer | DyGFormer |
|---|---|---|---|---|---|---|---|---|---|---|
| Enron | Forec. | $3.42 \pm 0.26$ | $4.15 \pm 0.30$ | $4.20 \pm 0.31$ | $36.16 \pm 0.35$ | $67.98 \pm 0.76$ | $9.00 \pm 0.11$ | $7.76 \pm 0.26$ | $12.44 \pm 0.33$ | $43.58 \pm 0.43$ |
| | Pred. | $3.47 \pm 1.57$ | $4.16 \pm 1.99$ | $4.29 \pm 1.92$ | $37.26 \pm 2.03$ | $73.06 \pm 1.38$ | $7.74 \pm 0.09$ | $7.61 \pm 1.18$ | $12.65 \pm 1.24$ | $47.79 \pm 1.69$ |
| UCI | Forec. | $6.84 \pm 0.47$ | $8.40 \pm 0.50$ | $8.44 \pm 0.50$ | $37.72 \pm 0.44$ | $97.97 \pm 1.58$ | $4.56 \pm 0.17$ | $11.36 \pm 0.30$ | $13.90 \pm 0.31$ | $23.85 \pm 0.39$ |
| | Pred. | $2.66 \pm 1.02$ | $3.08 \pm 0.72$ | $3.09 \pm 0.72$ | $24.58 \pm 0.83$ | $88.00 \pm 2.72$ | $1.15 \pm 0.02$ | $4.86 \pm 0.97$ | $8.03 \pm 0.99$ | $18.29 \pm 0.78$ |
| MOOC | Forec. | $55.41 \pm 0.52$ | $59.23 \pm 0.37$ | $58.56 \pm 0.38$ | $210.16 \pm 0.28$ | $539.03 \pm 2.59$ | $264.74 \pm 2.79$ | $89.27 \pm 0.20$ | $132.41 \pm 0.42$ | $226.80 \pm 1.38$ |
| | Pred. | $61.10 \pm 0.56$ | $64.53 \pm 0.43$ | $65.49 \pm 0.35$ | $230.79 \pm 1.74$ | $555.57 \pm 3.37$ | $301.55 \pm 5.18$ | $95.60 \pm 1.19$ | $140.46 \pm 1.36$ | $254.47 \pm 0.93$ |
| Wiki. | Forec. | $15.16 \pm 0.73$ | $16.29 \pm 0.79$ | $16.46 \pm 0.65$ | $44.58 \pm 0.88$ | $90.17 \pm 1.10$ | $10.27 \pm 0.46$ | $17.19 \pm 0.70$ | $19.80 \pm 0.71$ | $50.72 \pm 1.13$ |
| | Pred. | $13.67 \pm 1.08$ | $14.63 \pm 0.84$ | $13.91 \pm 0.91$ | $45.38 \pm 1.74$ | $89.25 \pm 1.72$ | $9.46 \pm 0.43$ | $15.99 \pm 1.44$ | $19.01 \pm 1.46$ | $49.27 \pm 1.50$ |
| LastFM | Forec. | $271.34 \pm 0.97$ | $281.58 \pm 1.81$ | $338.74 \pm 1.73$ | $985.54 \pm 34.93$ | $3316.72 \pm 21.18$ | $1413.11 \pm 18.08$ | $578.48 \pm 3.48$ | $702.40 \pm 1.33$ | $1037.08 \pm 8.49$ |
| | Pred. | $319.89 \pm 1.76$ | $340.54 \pm 12.63$ | $400.65 \pm 11.46$ | $1180.93 \pm 83.27$ | $3472.23 \pm 79.24$ | $1723.99 \pm 28.39$ | $674.63 \pm 12.96$ | $832.52 \pm 3.00$ | $1193.28 \pm 3.40$ |
| Myket | Forec. | $467.24 \pm 3.94$ | $465.55 \pm 6.40$ | $445.55 \pm 11.19$ | $680.26 \pm 15.71$ | $851.62 \pm 21.51$ | $479.92 \pm 6.43$ | $507.72 \pm 7.10$ | $602.63 \pm 7.31$ | $643.33 \pm 8.06$ |
| | Pred. | $425.36 \pm 6.72$ | $424.50 \pm 7.38$ | $403.12 \pm 9.27$ | $701.21 \pm 8.81$ | $876.41 \pm 17.09$ | $445.81 \pm 5.62$ | $475.26 \pm 8.32$ | $566.69 \pm 19.39$ | $628.24 \pm 7.03$ |
| Social | Forec. | $408.85 \pm 1.89$ | $424.96 \pm 3.62$ | $419.62 \pm 2.44$ | $1208.04 \pm 12.30$ | $4020.21 \pm 4.69$ | $1090.46 \pm 1.08$ | $709.80 \pm 2.45$ | $899.58 \pm 2.20$ | $1025.54 \pm 2.26$ |
| | Pred. | $500.15 \pm 3.65$ | $509.14 \pm 2.35$ | $510.64 \pm 3.11$ | $1544.19 \pm 2.59$ | $4308.14 \pm 8.00$ | $1379.71 \pm 0.74$ | $955.07 \pm 1.34$ | $1203.01 \pm 1.67$ | $1263.80 \pm 4.42$ |
| Reddit | Forec. | $120.79 \pm 0.77$ | $124.65 \pm 0.74$ | $131.35 \pm 2.35$ | $543.41 \pm 1.25$ | $561.97 \pm 2.74$ | $174.94 \pm 0.50$ | $161.26 \pm 1.12$ | $213.45 \pm 0.73$ | $248.81 \pm 1.26$ |
| | Pred. | $120.22 \pm 0.46$ | $124.28 \pm 0.41$ | $129.58 \pm 0.67$ | $585.98 \pm 53.62$ | $575.36 \pm 2.94$ | $191.38 \pm 0.59$ | $167.83 \pm 1.06$ | $227.86 \pm 2.07$ | $254.34 \pm 1.45$ |
| UN V. | Forec. | $4.25 \pm 0.30$ | $9.37 \pm 0.28$ | $20.11 \pm 0.36$ | $161.34 \pm 1.82$ | $5804.24 \pm 9.62$ | $6.88 \pm 0.05$ | $40.01 \pm 0.29$ | $831.89 \pm 9.65$ | $137.97 \pm 0.97$ |
| | Pred. | $158.35 \pm 0.46$ | $161.49 \pm 0.70$ | $174.23 \pm 0.68$ | $560.17 \pm 4.39$ | $5599.27 \pm 92.68$ | $684.81 \pm 8.57$ | $310.53 \pm 0.78$ | $1223.42 \pm 8.09$ | $406.54 \pm 0.94$ |
| US L. | Forec. | $1.01 \pm 0.26$ | $1.33 \pm 0.31$ | $1.32 \pm 0.31$ | $13.25 \pm 0.30$ | $32.43 \pm 0.55$ | $0.05 \pm 0.01$ | $3.19 \pm 0.25$ | $3.63 \pm 0.26$ | $18.67 \pm 0.26$ |
| | Pred. | $2.55 \pm 1.82$ | $2.93 \pm 1.76$ | $2.84 \pm 1.77$ | $19.40 \pm 0.82$ | $42.72 \pm 1.06$ | $0.71 \pm 0.02$ | $5.73 \pm 0.98$ | $6.87 \pm 1.25$ | $26.85 \pm 1.21$ |
| UN Tr. | Forec. | $1.95 \pm 0.26$ | $4.13 \pm 0.24$ | $3.97 \pm 0.12$ | $271.15 \pm 3.19$ | $2813.05 \pm 2.68$ | $1.83 \pm 0.06$ | $19.42 \pm 0.26$ | $507.83 \pm 14.74$ | $84.71 \pm 0.52$ |
| | Pred. | $34.63 \pm 0.42$ | $37.36 \pm 0.44$ | $36.37 \pm 0.44$ | $466.44 \pm 2.36$ | $2992.96 \pm 384.16$ | $161.85 \pm 0.63$ | $90.81 \pm 0.43$ | $719.00 \pm 70.07$ | $213.38 \pm 1.02$ |
| Can. P. | Forec. | $0.56 \pm 0.22$ | $2.00 \pm 0.24$ | $1.99 \pm 0.24$ | $60.15 \pm 0.31$ | $1034.22 \pm 1.06$ | $0.11 \pm 0.02$ | $4.11 \pm 0.21$ | $20.65 \pm 0.30$ | $464.45 \pm 25.61$ |
| | Pred. | $2.38 \pm 0.46$ | $3.68 \pm 0.54$ | $3.80 \pm 0.56$ | $90.50 \pm 1.56$ | $1242.03 \pm 185.41$ | $2.02 \pm 0.03$ | $7.64 \pm 0.63$ | $30.36 \pm 1.72$ | $380.33 \pm 2.02$ |
| Flights | Forec. | $343.75 \pm 10.64$ | $346.74 \pm 9.28$ | $341.56 \pm 9.66$ | $357.77 \pm 5.58$ | $10363.33 \pm 2098.15$ | $21.36 \pm 0.44$ | $68.14 \pm 0.78$ | $1611.70 \pm 33.57$ | $994.11 \pm 7.16$ |
| | Pred. | $1567.52 \pm 79.75$ | $1550.22 \pm 94.86$ | $1581.06 \pm 47.29$ | $2359.88 \pm 54.11$ | $3420.00 \pm 57.27$ | $2132.99 \pm 103.26$ | $1933.46 \pm 84.54$ | $2113.83 \pm 20.74$ | $2701.10 \pm 31.46$ |
| Cont. | Forec. | $656.31 \pm 2.85$ | $673.36 \pm 3.62$ | $683.82 \pm 2.96$ | $1966.25 \pm 10.58$ | $3878.32 \pm 11.95$ | $2646.55 \pm 37.98$ | $1139.57 \pm 4.54$ | $1378.58 \pm 4.22$ | $1518.44 \pm 12.00$ |
| | Pred. | $938.48 \pm 6.81$ | $940.55 \pm 7.94$ | $953.46 \pm 8.02$ | $2528.31 \pm 34.70$ | $4358.78 \pm 21.10$ | $3854.40 \pm 32.23$ | $1523.29 \pm 26.55$ | $1848.28 \pm 40.53$ | $1873.09 \pm 11.28$ |

```python
1   def evaluate_tgnn(tgnn, t_graph):
2       test_window_loader = get_window_loader(
3           t_graph,
4           horizon=3600,
5           test_ratio=0.15,
6           n_negatives=1
7       )
8       for target_t_edges, t_edge_label in test_window_loader:
9           # Initialise lists to save the individual forecasts and
10          # intermediate node respresentations
11          window_out, window_z = [], []
12          neighbour_t_graph = t_graph.get_recent_neighbours(target_t_edges)
13          # Divide the time window into chunks of size 10
14          for chunk_t_edges in subdivide(target_t_edges, chunk_size=10):
15              # Resample neighourhood to remove unnecessary nodes from neighbourhood
16              # Sample from the `neighour_t_graph` to make sure no information is leaked,
17              # i.e. no edges from the same batch are sampled
18              chunk_t_graph = neighbour_t_graph.get_recent_neighbours(chunk_t_edges)
19              z = tgnn.get_memory(chunk_t_graph)
20              z = tgnn(z, chunk_t_graph)
21              out = forecast_links(z, chunk_t_edges)
22              # Save the edge forecasts of this chunk
23              window_out.append(out)
24              # Save the intermediate node representations
25              window_z.append(z)
26          metrics = calculate_metrics(window_out, t_edge_label)
27          tgnn.update_memory(window_z, t_edge_label)
```

Listing 2: Python-like pseudocode explaining how to subdivide windows to prevent GPU memory overflows. We only add comments to the code sections that differ from the basic implementation in Listing 1.

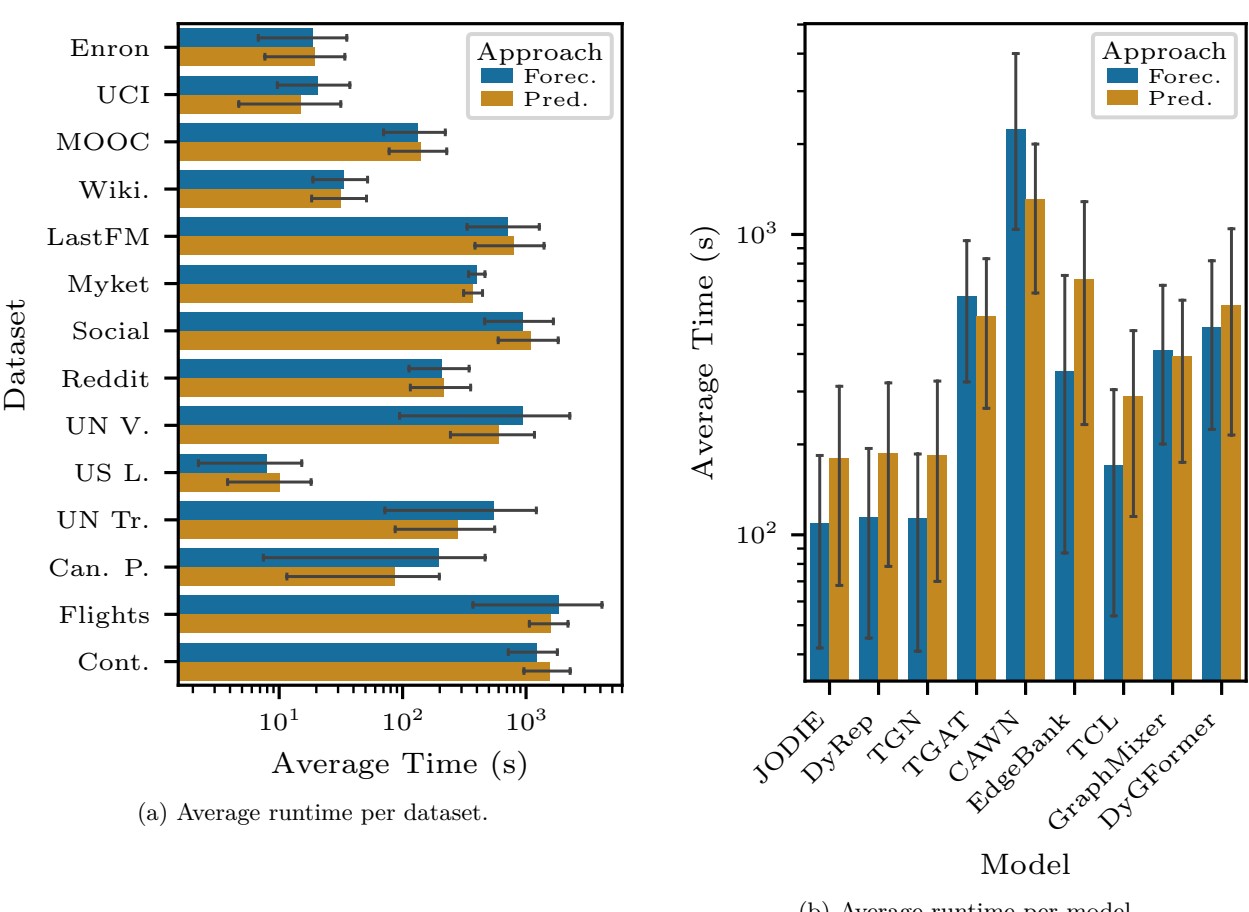

(a) Average runtime per dataset.

(b) Average runtime per model.

Figure 14: Runtimes of the experiments in Section 4 averaged across datasets or models.

# H Detailed AUC-ROC and Average Precision Results

Here, we provide detailed tabulated results for all models' AUC-ROC and average precision performance across five runs, including standard deviations. We further include visualisations investigating the causes of the large performance differences between dynamic link prediction and forecasting.

## H.1 AUC-ROC

The following reports the complete results (including standard deviations) of the experiments conducted in Section 4. For easier comparison between the performance of the batch-based and our window-based approach, we visualise the results in two different table formats. Table 13 and Table 14 show AUC-ROC scores for time-window-based link forecasting and the relative change compared to the batch-based evaluation of dynamic link prediction. Table 15 and Table 16 show the full results, including standard deviations for both approaches separately.

Table 13: Test AUC-ROC scores for link forecasting and the relative performance change compared to link prediction for continuous-time graphs using the *same trained models* (standard deviations in Appendix H.1). The last row/column provides the mean $\mu$ and standard deviation $\sigma$ of the absolute values (without sign) of the relative change per column/row.

| Dataset | JODIE | DyRep | TGN | TGAT | CAWN | EdgeBank | TCL | GraphMixer | DyGFormer | $\mu \pm \sigma$ |
|---|---|---|---|---|---|---|---|---|---|---|
| Enron | 84.0(↑8.6%) | 80.3(↑9.2%) | 67.9(↓0.2%) | 69.0(↑17.5%) | 75.7(↑13.9%) | 82.7(↑3.6%) | 75.1(↑11.1%) | 88.6(↑9.0%) | 84.5(↑10.6%) | 9.3%±5.1% |
| UCI | 86.8(↑4.2%) | 60.2(↑17.1%) | 62.1(↓1.5%) | 55.2(↓7.4%) | 56.5(↓3.0%) | 72.5(↑4.9%) | 56.3(↓6.2%) | 80.2(↓0.5%) | 75.7(↓0.6%) | 5.0%±5.1% |
| MOOC | 83.1(↓2.1%) | 79.0(↓2.1%) | 87.4(↓1.2%) | 79.9(↓2.9%) | 68.8(↓2.2%) | 59.8(↓3.4%) | 68.4(↓5.8%) | 70.3(↓5.5%) | 80.0(↓1.5%) | 3.0%±1.7% |
| Wiki. | 81.5(↓0.4%) | 78.3(↓0.1%) | 83.7(↓0.6%) | 82.9(↓0.7%) | 71.3(↓0.4%) | 77.2(↑0.1%) | 84.6(↓0.6%) | 87.3(↓0.6%) | 79.8(↓0.3%) | 0.4%±0.2% |
| LastFM | 76.3(↓2.2%) | 69.0(↓3.7%) | 79.2(↓1.9%) | 65.2(↓4.7%) | 66.3(↓2.6%) | 78.0(↓0.2%) | 62.5(↓2.7%) | 59.9(↓9.2%) | 78.2(↓1.0%) | 3.1%±2.6% |
| Myket | 64.4(↑0.6%) | 64.1(↑0.1%) | 61.2(↑0.1%) | 57.8(↑0.4%) | 33.5(↑3.1%) | 52.6(↑1.3%) | 58.3(↑0.3%) | 59.8(↑0.5%) | 33.8(↑3.0%) | 1.0%±1.2% |
| Social | 92.1(↑0.8%) | 92.2(↓0.5%) | 92.2(↑0.5%) | 92.5(↓0.1%) | 86.5(↓1.4%) | 84.9(↓1.1%) | 94.7(↓0.6%) | 94.6(↑0.6%) | 97.3(↑0.0%) | 0.6%±0.4% |
| Reddit | 80.6(↓0.0%) | 79.5(↑0.0%) | 80.4(↓0.0%) | 78.6(↓0.1%) | 80.2(↓0.0%) | 78.6(↓0.1%) | 76.2(↓0.1%) | 77.1(↓0.1%) | 80.2(↑0.0%) | 0.0%±0.1% |
| $\mu \pm \sigma$ | 2.4%±2.9% | 4.1%±6.1% | 0.8%±0.7% | 4.2%±6.0% | 3.3%±4.4% | 1.8%±1.9% | 3.4%±4.0% | 3.2%±4.0% | 2.1%±3.6% | |

Table 14: Test AUC-ROC scores as in Table 13 for discrete-time graphs.

| Dataset | JODIE | DyRep | TGN | TGAT | CAWN | EdgeBank | TCL | GraphMixer | DyGFormer | $\mu \pm \sigma$ |
|---|---|---|---|---|---|---|---|---|---|---|
| UN V. | 54.0(↓26.7%) | 52.2(↓28.2%) | 51.3(↓27.1%) | 54.4(↑3.0%) | 53.7(↑7.1%) | 89.6(↑0.0%) | 53.4(↑0.6%) | 56.9(↑1.1%) | 65.2(↑3.5%) | 10.8%±12.6% |
| US L. | 52.5(↓6.8%) | 61.8(↓22.6%) | 57.7(↓31.2%) | 78.6(↑0.2%) | 82.0(↑0.2%) | 68.4(↑1.3%) | 75.4(↓0.3%) | 90.4(↑0.2%) | 89.4(↑0.0%) | 7.0%±11.7% |
| UN Tr. | 57.7(↓12.8%) | 50.3(↓20.4%) | 54.3(↓14.0%) | 64.0(↑3.6%) | 67.6(↑4.5%) | 85.6(↓1.0%) | 63.7(↑4.5%) | 68.6(↑3.4%) | 70.7(↑3.4%) | 7.5%±6.6% |
| Can. P. | 63.6(↓0.5%) | 67.5(↑1.2%) | 73.2(↓0.2%) | 72.7(↑1.5%) | 70.0(↑2.9%) | 63.2(↑0.4%) | 69.5(↑2.0%) | 80.7(↓0.6%) | 85.5(↓12.5%) | 2.4%±3.9% |
| Flights | 67.4(↓3.1%) | 66.0(↓4.3%) | 68.1(↓1.0%) | 72.6(↑0.0%) | 66.0(↑0.1%) | 74.6(↑0.0%) | 70.6(↓0.0%) | 70.7(↓0.0%) | 68.8(↑0.1%) | 1.0%±1.6% |
| Cont. | 95.6(↑0.1%) | 94.9(↓0.6%) | 96.6(↑0.5%) | 95.9(↑0.6%) | 86.7(↑4.1%) | 93.0(↑0.9%) | 95.7(↑1.3%) | 95.2(↑1.1%) | 97.7(↑0.6%) | 1.1%±1.2% |
| $\mu \pm \sigma$ | 8.3%±10.2% | 12.9%±12.2% | 12.3%±14.1% | 1.5%±1.5% | 3.1%±2.7% | 0.6%±0.5% | 1.5%±1.7% | 1.1%±1.2% | 3.4%±4.8% | |

Table 15: Average AUC-ROC performance over five runs for the test set of the continuous-time datasets, including standard deviations. For discrete- and continuous-time datasets, the average rank of each model is reported in the last row, respectively. The best performance score for each dataset is highlighted in bold.

| Eval. | Dataset | JODIE | DyRep | TGN | TGAT | CAWN | EdgeBank | TCL | GraphMixer | DyGFormer |
|---|---|---|---|---|---|---|---|---|---|---|
| Forec. | Enron | 84.0 ± 5.1 | 80.3 ± 1.4 | 67.9 ± 7.1 | 69.0 ± 1.6 | 75.7 ± 0.5 | 82.7 ± 0.0 | 75.1 ± 5.2 | **88.6 ± 0.5** | 84.5 ± 0.6 |
| | UCI | **86.8 ± 1.0** | 60.2 ± 2.8 | 62.1 ± 1.3 | 55.2 ± 1.4 | 56.5 ± 0.5 | 72.5 ± 0.0 | 56.3 ± 1.0 | 80.2 ± 1.0 | 75.7 ± 0.5 |
| | MOOC | 83.1 ± 4.2 | 79.0 ± 4.5 | **87.4 ± 1.9** | 79.9 ± 0.8 | 68.8 ± 1.6 | 59.8 ± 0.0 | 68.4 ± 1.4 | 70.3 ± 1.2 | 80.0 ± 9.0 |
| | Wiki. | 81.5 ± 0.4 | 78.3 ± 0.4 | 83.7 ± 0.6 | 82.9 ± 0.3 | 71.3 ± 0.8 | 77.2 ± 0.0 | 84.6 ± 0.5 | **87.3 ± 0.3** | 79.8 ± 1.6 |
| | LastFM | 76.3 ± 0.8 | 69.0 ± 1.4 | **79.2 ± 2.7** | 65.2 ± 0.9 | 66.3 ± 0.3 | 78.0 ± 0.0 | 62.5 ± 6.4 | 59.9 ± 1.4 | 78.2 ± 0.6 |
| | Myket | **64.4 ± 2.2** | 64.1 ± 2.9 | 61.2 ± 2.6 | 57.8 ± 0.5 | 33.5 ± 0.4 | 52.6 ± 0.0 | 58.3 ± 2.2 | 59.8 ± 0.4 | 33.8 ± 0.9 |
| | Social | 92.1 ± 1.9 | 92.2 ± 0.7 | 92.2 ± 2.6 | 92.5 ± 0.5 | 86.5 ± 0.0 | 84.9 ± 0.0 | 94.7 ± 0.5 | 94.6 ± 0.2 | **97.3 ± 0.1** |
| | Reddit | **80.6 ± 0.1** | 79.5 ± 0.8 | 80.4 ± 0.4 | 78.6 ± 0.7 | 80.2 ± 0.3 | 78.6 ± 0.0 | 76.2 ± 0.4 | 77.1 ± 0.4 | 80.2 ± 1.1 |
| | Avg. Rank | 3.0 | 5.1 | 3.6 | 6.0 | 7.0 | 6.4 | 6.1 | 4.2 | 3.5 |
| Pred. | Enron | 77.4 ± 3.6 | 73.5 ± 2.4 | 68.0 ± 2.9 | 58.7 ± 1.2 | 66.4 ± 0.4 | 79.8 ± 0.0 | 67.6 ± 5.5 | **81.3 ± 0.8** | 76.4 ± 0.5 |
| | UCI | **83.3 ± 1.4** | 51.4 ± 7.8 | 63.0 ± 1.3 | 59.6 ± 1.5 | 58.2 ± 0.6 | 69.1 ± 0.0 | 60.0 ± 0.9 | 80.6 ± 0.8 | 76.2 ± 0.6 |
| | MOOC | 84.8 ± 3.1 | 80.7 ± 3.2 | **88.5 ± 1.6** | 82.3 ± 0.6 | 70.4 ± 1.3 | 61.9 ± 0.0 | 72.6 ± 0.6 | 74.4 ± 1.4 | 81.2 ± 8.9 |
| | Wiki. | 81.8 ± 0.4 | 78.4 ± 0.4 | 84.1 ± 0.6 | 83.5 ± 0.2 | 71.6 ± 0.8 | 77.1 ± 0.0 | 85.2 ± 0.5 | **87.8 ± 0.3** | 80.0 ± 1.6 |
| | LastFM | 78.0 ± 0.7 | 71.7 ± 1.1 | **80.7 ± 2.4** | 68.4 ± 0.7 | 68.1 ± 0.3 | 78.2 ± 0.0 | 64.3 ± 6.0 | 65.9 ± 1.7 | 78.9 ± 0.6 |
| | Myket | 64.0 ± 2.1 | **64.2 ± 2.7** | 61.1 ± 2.6 | 57.6 ± 0.4 | 32.5 ± 0.4 | 52.0 ± 0.0 | 58.4 ± 2.0 | 59.5 ± 0.4 | 32.8 ± 1.0 |
| | Social | 91.3 ± 2.1 | 92.7 ± 0.5 | 91.7 ± 3.3 | 92.6 ± 0.5 | 87.7 ± 0.1 | 85.8 ± 0.0 | 95.2 ± 0.2 | 94.1 ± 0.2 | **97.3 ± 0.1** |
| | Reddit | **80.6 ± 0.1** | 79.5 ± 0.8 | 80.4 ± 0.4 | 78.7 ± 0.6 | 80.2 ± 0.3 | 78.6 ± 0.0 | 76.2 ± 0.4 | 77.1 ± 0.4 | 80.2 ± 1.1 |
| | Avg. Rank | 3.1 | 5.1 | 3.4 | 5.8 | 7.6 | 6.1 | 5.9 | 4.1 | 3.9 |

Table 16: Average AUC-ROC performance over five runs for the test set of the discrete-time datasets, including standard deviation. For discrete- and continuous-time datasets, the average rank of each model is reported in the last row, respectively.

| Eval. | Dataset | JODIE | DyRep | TGN | TGAT | CAWN | EdgeBank | TCL | GraphMixer | DyGFormer |
|---|---|---|---|---|---|---|---|---|---|---|
| Forec. | UN V. | 54.0 ± 1.8 | 52.2 ± 2.0 | 51.3 ± 7.1 | 54.4 ± 3.6 | 53.7 ± 2.1 | **89.6 ± 0.0** | 53.4 ± 1.0 | 56.9 ± 1.6 | 65.2 ± 1.1 |
| | US L. | 52.5 ± 1.8 | 61.8 ± 3.5 | 57.7 ± 1.8 | 78.6 ± 7.9 | 82.0 ± 4.0 | 68.4 ± 0.0 | 75.4 ± 5.3 | **90.4 ± 1.5** | 89.4 ± 0.9 |
| | UN Tr. | 57.7 ± 3.3 | 50.3 ± 1.4 | 54.3 ± 1.5 | 64.0 ± 1.3 | 67.6 ± 1.2 | **85.6 ± 0.0** | 63.7 ± 1.6 | 68.6 ± 2.6 | 70.7 ± 2.6 |
| | Can. P. | 63.6 ± 0.8 | 67.5 ± 8.5 | 73.2 ± 1.1 | 72.7 ± 2.2 | 70.0 ± 1.4 | 63.2 ± 0.0 | 69.5 ± 3.1 | 80.7 ± 0.9 | **85.5 ± 3.5** |
| | Flights | 67.4 ± 2.0 | 66.0 ± 1.9 | 68.1 ± 1.7 | 72.6 ± 0.2 | 66.0 ± 1.7 | **74.6 ± 0.0** | 70.6 ± 0.1 | 70.7 ± 0.3 | 68.8 ± 1.0 |
| | Cont. | 95.6 ± 0.8 | 94.9 ± 0.3 | 96.6 ± 0.3 | 95.9 ± 0.2 | 86.7 ± 0.1 | 93.0 ± 0.0 | 95.7 ± 0.5 | 95.2 ± 0.2 | **97.7 ± 0.0** |
| | Avg. Rank | 6.8 | 7.7 | 6.0 | 3.7 | 6.0 | 4.3 | 5.3 | 3.0 | 2.2 |
| Pred. | UN V. | 73.7 ± 2.4 | 72.6 ± 1.5 | 70.3 ± 4.3 | 52.8 ± 3.6 | 50.1 ± 1.6 | **89.5 ± 0.0** | 53.0 ± 1.6 | 56.2 ± 2.0 | 63.0 ± 1.1 |
| | US L. | 56.3 ± 1.9 | 79.9 ± 1.1 | 84.0 ± 2.2 | 78.5 ± 7.8 | 81.8 ± 4.0 | 67.5 ± 0.0 | 75.6 ± 5.4 | **90.2 ± 1.6** | 89.4 ± 0.9 |
| | UN Tr. | 66.1 ± 3.0 | 63.2 ± 2.1 | 63.1 ± 1.2 | 61.7 ± 1.3 | 64.7 ± 1.3 | **86.4 ± 0.0** | 60.9 ± 1.3 | 66.3 ± 2.5 | 68.3 ± 2.3 |
| | Can. P. | 63.9 ± 0.7 | 66.6 ± 2.5 | 73.4 ± 3.5 | 71.6 ± 2.6 | 68.0 ± 1.0 | 62.9 ± 0.0 | 68.2 ± 3.6 | 81.2 ± 1.0 | **97.7 ± 0.7** |
| | Flights | 69.5 ± 2.2 | 69.0 ± 1.0 | 68.8 ± 1.6 | 72.6 ± 0.2 | 66.0 ± 1.7 | **74.6 ± 0.0** | 70.6 ± 0.1 | 70.7 ± 0.3 | 68.8 ± 1.0 |
| | Cont. | 95.5 ± 0.5 | 95.4 ± 0.2 | 96.1 ± 0.8 | 95.4 ± 0.3 | 83.3 ± 0.0 | 92.2 ± 0.0 | 94.5 ± 0.5 | 94.1 ± 0.2 | **97.1 ± 0.0** |
| | Avg. Rank | 5.2 | 5.3 | 4.3 | 5.3 | 7.0 | 4.7 | 6.3 | 3.7 | 3.2 |

## H.2 Average precision

Average precision scores as an additional metric to evaluate experiments in Section 4. The results are comparable to the results using the AUC-ROC score.

Table 17: Mean average precision performance for dynamic link forecasting (window-based) over five runs for the continuous-time datasets. Values in parentheses show the relative change as compared to the average precision performance for dynamic link prediction (batch-based).

| Dataset | JODIE | DyRep | TGN | TGAT | CAWN | EdgeBank | TCL | GraphMixer | DyGFormer | $\mu \pm \sigma$ |
|---|---|---|---|---|---|---|---|---|---|---|
| Enron | 80.8(↑12.0%) | 78.3(↑12.4%) | 68.6(↑5.0%) | 71.7(↑13.5%) | 76.0(↑15.1%) | 81.1(↑5.5%) | 78.2(↑11.3%) | 89.8(↑9.2%) | 85.3(↑11.6%) | 10.6%±3.4% |
| UCI | 87.0(↑7.2%) | 59.5(↑21.4%) | 69.2(↓2.8%) | 64.4(↓6.2%) | 64.0(↓1.6%) | 68.6(↑5.4%) | 65.2(↓5.5%) | 85.3(↓0.6%) | 80.5(↓0.2%) | 5.7%±6.4% |
| MOOC | 82.4(↓1.1%) | 76.9(↓0.3%) | 86.3(↓0.8%) | 82.7(↓2.1%) | 72.3(↓1.7%) | 59.0(↓2.7%) | 74.9(↓4.6%) | 74.4(↓4.5%) | 82.1(↓0.4%) | 2.0%±1.6% |
| Wiki. | 84.1(↓0.1%) | 80.9(↑0.1%) | 88.5(↓0.4%) | 87.5(↓0.4%) | 75.1(↑0.1%) | 73.3(↑0.3%) | 89.2(↓0.4%) | 90.8(↓0.4%) | 83.1(↑0.0%) | 0.2%±0.2% |
| LastFM | 76.7(↓1.1%) | 69.4(↓2.7%) | 78.8(↓1.9%) | 72.1(↓3.9%) | 68.2(↓2.3%) | 73.4(↑0.3%) | 70.3(↓1.8%) | 70.0(↓5.4%) | 80.5(↓0.7%) | 2.2%±1.6% |
| Myket | 64.5(↑1.6%) | 63.1(↑0.9%) | 62.8(↑1.2%) | 57.9(↑1.4%) | 46.6(↑3.3%) | 51.9(↑1.3%) | 58.9(↑1.0%) | 60.0(↑1.5%) | 46.1(↑3.3%) | 1.7%±0.9% |
| Social | 89.4(↑1.3%) | 91.9(↑0.3%) | 93.9(↑0.8%) | 95.0(↑0.2%) | 85.6(↓0.6%) | 79.7(↓1.1%) | 96.1(↓0.1%) | 95.8(↑0.4%) | 97.7(↑0.3%) | 0.6%±0.4% |
| Reddit | 80.1(↓0.0%) | 79.2(↑0.0%) | 80.6(↑0.0%) | 78.6(↓0.1%) | 81.3(↑0.2%) | 73.5(↓0.2%) | 76.5(↓0.0%) | 77.5(↓0.0%) | 82.8(↑0.1%) | 0.1%±0.1% |
| $\mu \pm \sigma$ | 3.0%±4.3% | 4.8%±7.9% | 1.6%±1.6% | 3.5%±4.6% | 3.1%±5.0% | 2.1%±2.2% | 3.1%±3.9% | 2.8%±3.3% | 2.1%±4.0% | |

Table 18: Mean average precision performance for dynamic link forecasting (window-based) over five runs for the discrete-time datasets. Values in parentheses show the relative change as compared to the average precision performance for dynamic link prediction (batch-based).

| Dataset | JODIE | DyRep | TGN | TGAT | CAWN | EdgeBank | TCL | GraphMixer | DyGFormer | $\mu \pm \sigma$ |
|---|---|---|---|---|---|---|---|---|---|---|
| UN V. | 52.6(↓22.4%) | 49.6(↓26.8%) | 49.7(↓24.8%) | 52.7(↑0.9%) | 52.4(↑3.4%) | 84.2(↓0.7%) | 52.4(↓1.7%) | 54.0(↑0.1%) | 62.4(↑4.0%) | 9.4%±11.6% |
| US L. | 46.0(↓4.5%) | 62.5(↓14.5%) | 58.6(↓27.8%) | 71.0(↓0.3%) | 80.7(↓0.1%) | 63.2(↓0.2%) | 77.5(↑0.1%) | 86.5(↑0.6%) | 86.1(↑0.4%) | 5.4%±9.6% |
| UN Tr. | 52.7(↓10.6%) | 49.4(↓16.6%) | 53.2(↓9.7%) | 58.9(↑1.7%) | 59.2(↑2.4%) | 79.0(↓2.6%) | 57.5(↑2.3%) | 65.8(↑3.3%) | 67.1(↑4.4%) | 6.0%±5.2% |
| Can. P. | 52.1(↓1.3%) | 61.0(↓1.3%) | 69.9(↑2.0%) | 70.8(↑4.5%) | 68.3(↑6.9%) | 59.4(↓6.8%) | 68.2(↑6.2%) | 80.9(↑4.8%) | 83.2(↑14.3%) | 5.3%±4.0% |
| Flights | 65.2(↓2.2%) | 63.9(↓4.3%) | 68.3(↓0.0%) | 73.5(↑1.1%) | 65.5(↑0.8%) | 70.4(↓0.2%) | 71.0(↑0.4%) | 71.9(↑1.0%) | 69.1(↑0.7%) | 1.2%±1.3% |
| Cont. | 94.0(↓0.2%) | 95.8(↑0.3%) | 97.0(↑0.8%) | 96.8(↑0.8%) | 88.2(↑4.4%) | 89.4(↑0.6%) | 96.6(↑1.8%) | 95.7(↑1.6%) | 98.3(↑0.6%) | 1.2%±1.3% |
| $\mu \pm \sigma$ | 6.9%±8.5% | 10.6%±10.4% | 10.8%±12.5% | 1.6%±1.5% | 3.0%±2.5% | 1.8%±2.6% | 2.1%±2.2% | 1.9%±1.8% | 4.1%±5.3% | |

Table 19: Mean average precision performance over five runs for the test set of the continuous-time datasets, including standard deviations. For discrete- and continuous-time datasets, the average rank of each model is reported in the last row, respectively.

| Eval. | Dataset | JODIE | DyRep | TGN | TGAT | CAWN | EdgeBank | TCL | GraphMixer | DyGFormer |
|---|---|---|---|---|---|---|---|---|---|---|
| | Enron | 80.8 ± 5.3 | 78.3 ± 2.3 | 68.6 ± 5.6 | 71.7 ± 1.2 | 76.0 ± 0.7 | 81.1 ± 0.0 | 78.2 ± 2.9 | **89.8 ± 0.4** | 85.3 ± 0.6 |
| | UCI | **87.0 ± 1.9** | 59.5 ± 2.3 | 69.2 ± 1.0 | 64.4 ± 1.1 | 64.0 ± 0.7 | 68.6 ± 0.0 | 65.2 ± 0.8 | 85.3 ± 0.6 | 80.5 ± 0.9 |
| | MOOC | 82.4 ± 4.9 | 76.9 ± 4.3 | **86.3 ± 2.3** | 82.7 ± 0.7 | 72.3 ± 1.3 | 59.0 ± 0.0 | 74.9 ± 0.7 | 74.4 ± 0.6 | 82.1 ± 8.7 |
| | Wiki. | 84.1 ± 0.5 | 80.9 ± 0.4 | 88.5 ± 0.4 | 87.5 ± 0.2 | 75.1 ± 1.0 | 73.3 ± 0.0 | 89.2 ± 0.3 | **90.8 ± 0.2** | 83.1 ± 1.2 |
| Forec. | LastFM | 76.7 ± 0.6 | 69.4 ± 1.8 | 78.8 ± 3.5 | 72.1 ± 0.8 | 68.2 ± 0.5 | 73.4 ± 0.0 | 70.3 ± 6.5 | 70.0 ± 1.1 | **80.5 ± 0.9** |
| | Myket | **64.5 ± 1.8** | 63.1 ± 1.5 | 62.8 ± 2.2 | 57.9 ± 0.4 | 46.6 ± 0.2 | 51.9 ± 0.0 | 58.9 ± 2.6 | 60.0 ± 0.2 | 46.1 ± 1.7 |
| | Social | 89.4 ± 4.7 | 91.9 ± 1.0 | 93.9 ± 1.7 | 95.0 ± 0.3 | 85.6 ± 0.1 | 79.7 ± 0.0 | 96.1 ± 0.4 | 95.8 ± 0.2 | **97.7 ± 0.1** |
| | Reddit | 80.1 ± 0.3 | 79.2 ± 0.9 | 80.6 ± 0.6 | 78.6 ± 1.0 | 81.3 ± 0.4 | 73.5 ± 0.0 | 76.5 ± 0.6 | 77.5 ± 0.5 | **82.8 ± 0.8** |
| | Avg. Rank | 3.5 | 5.9 | 3.8 | 5.2 | 7.2 | 6.9 | 5.1 | 4.0 | 3.4 |
| | Enron | 72.1 ± 3.0 | 69.7 ± 3.7 | 65.3 ± 3.2 | 63.2 ± 0.5 | 66.0 ± 0.5 | 76.9 ± 0.0 | 70.2 ± 3.4 | **82.3 ± 0.6** | 76.4 ± 0.4 |
| | UCI | 81.1 ± 3.3 | 49.0 ± 4.5 | 71.2 ± 1.1 | 68.6 ± 1.1 | 65.1 ± 0.6 | 65.0 ± 0.0 | 69.0 ± 0.8 | **85.9 ± 0.5** | 80.7 ± 1.1 |
| | MOOC | 83.4 ± 4.3 | 77.1 ± 3.8 | **87.0 ± 2.1** | 84.5 ± 0.7 | 73.5 ± 1.0 | 60.7 ± 0.0 | 78.5 ± 0.5 | 77.9 ± 0.8 | 82.4 ± 9.3 |
| | Wiki. | 84.1 ± 0.5 | 80.9 ± 0.3 | 88.8 ± 0.4 | 87.9 ± 0.2 | 75.0 ± 1.1 | 73.1 ± 0.0 | 89.5 ± 0.3 | **91.2 ± 0.2** | 83.1 ± 1.1 |
| Pred. | LastFM | 77.6 ± 0.6 | 71.4 ± 1.7 | 80.3 ± 3.2 | 75.0 ± 0.7 | 69.8 ± 0.5 | 73.2 ± 0.0 | 71.6 ± 6.1 | 74.1 ± 1.3 | **81.1 ± 0.9** |
| | Myket | **63.4 ± 1.7** | 62.5 ± 1.4 | 62.1 ± 2.3 | 57.1 ± 0.4 | 45.1 ± 0.2 | 51.3 ± 0.0 | 58.3 ± 2.2 | 59.1 ± 0.2 | 44.7 ± 1.6 |
| | Social | 88.3 ± 4.8 | 91.6 ± 0.7 | 93.2 ± 2.4 | 94.8 ± 0.3 | 86.2 ± 0.2 | 80.6 ± 0.0 | 96.2 ± 0.2 | 95.4 ± 0.1 | **97.3 ± 0.1** |
| | Reddit | 80.1 ± 0.3 | 79.2 ± 0.9 | 80.5 ± 0.5 | 78.6 ± 1.0 | 81.1 ± 0.4 | 73.7 ± 0.0 | 76.5 ± 0.6 | 77.5 ± 0.5 | **82.7 ± 0.8** |
| | Avg. Rank | 3.6 | 6.2 | 3.6 | 5.1 | 7.1 | 7.4 | 4.9 | 3.5 | 3.5 |

Table 20: Mean average precision performance over five runs for the test set of the discrete-time datasets, including standard deviations. For discrete- and continuous-time datasets, the average rank of each model is reported in the last row, respectively.

| Eval. | Dataset | JODIE | DyRep | TGN | TGAT | CAWN | EdgeBank | TCL | GraphMixer | DyGFormer |
|---|---|---|---|---|---|---|---|---|---|---|
| | UN V. | $52.6 \pm 1.8$ | $49.6 \pm 1.9$ | $49.7 \pm 3.9$ | $52.7 \pm 2.6$ | $52.4 \pm 2.0$ | $\mathbf{84.2 \pm 0.0}$ | $52.4 \pm 0.9$ | $54.0 \pm 1.4$ | $62.4 \pm 1.7$ |
| | US L. | $46.0 \pm 0.9$ | $62.5 \pm 3.6$ | $58.6 \pm 2.4$ | $71.0 \pm 8.9$ | $80.7 \pm 3.7$ | $63.2 \pm 0.0$ | $77.5 \pm 4.3$ | $\mathbf{86.5 \pm 1.9}$ | $86.1 \pm 1.0$ |
| | UN Tr. | $52.7 \pm 3.0$ | $49.4 \pm 0.9$ | $53.2 \pm 1.5$ | $58.9 \pm 2.7$ | $59.2 \pm 1.7$ | $\mathbf{79.0 \pm 0.0}$ | $57.5 \pm 1.9$ | $65.8 \pm 1.9$ | $67.1 \pm 2.7$ |
| Forec. | Can. P. | $52.1 \pm 0.5$ | $61.0 \pm 7.6$ | $69.9 \pm 0.8$ | $70.8 \pm 1.6$ | $68.3 \pm 2.3$ | $59.4 \pm 0.0$ | $68.2 \pm 1.6$ | $80.9 \pm 0.5$ | $\mathbf{83.2 \pm 2.9}$ |
| | Flights | $65.2 \pm 2.7$ | $63.9 \pm 2.8$ | $68.3 \pm 2.2$ | $\mathbf{73.5 \pm 0.3}$ | $65.5 \pm 1.3$ | $70.4 \pm 0.0$ | $71.0 \pm 0.4$ | $71.9 \pm 0.8$ | $69.1 \pm 1.8$ |
| | Cont. | $94.0 \pm 2.6$ | $95.8 \pm 0.4$ | $97.0 \pm 0.5$ | $96.8 \pm 0.2$ | $88.2 \pm 0.2$ | $89.4 \pm 0.0$ | $96.6 \pm 0.4$ | $95.7 \pm 0.2$ | $\mathbf{98.3 \pm 0.0}$ |
| | Avg. Rank | 7.7 | 7.7 | 5.8 | 3.5 | 5.7 | 4.7 | 5.0 | 2.8 | 2.2 |
| | UN V. | $67.8 \pm 1.9$ | $67.8 \pm 1.7$ | $66.1 \pm 3.9$ | $52.3 \pm 2.5$ | $50.7 \pm 1.4$ | $\mathbf{84.8 \pm 0.0}$ | $53.3 \pm 1.3$ | $53.9 \pm 1.7$ | $59.9 \pm 1.4$ |
| | US L. | $48.2 \pm 1.0$ | $73.1 \pm 2.2$ | $81.2 \pm 2.1$ | $71.2 \pm 8.2$ | $80.8 \pm 3.5$ | $63.3 \pm 0.0$ | $77.4 \pm 4.5$ | $\mathbf{86.0 \pm 2.0}$ | $85.8 \pm 1.0$ |
| | UN Tr. | $58.9 \pm 3.1$ | $59.3 \pm 1.8$ | $58.9 \pm 1.5$ | $57.9 \pm 2.4$ | $57.9 \pm 2.1$ | $\mathbf{81.1 \pm 0.0}$ | $56.2 \pm 1.5$ | $63.8 \pm 1.6$ | $64.3 \pm 2.2$ |
| Pred. | Can. P. | $52.8 \pm 0.5$ | $61.8 \pm 1.1$ | $68.5 \pm 2.1$ | $67.7 \pm 1.7$ | $63.9 \pm 1.3$ | $63.8 \pm 0.0$ | $64.2 \pm 2.0$ | $77.2 \pm 0.4$ | $\mathbf{97.1 \pm 0.7}$ |
| | Flights | $66.7 \pm 3.3$ | $66.8 \pm 1.6$ | $68.3 \pm 1.8$ | $\mathbf{72.7 \pm 0.2}$ | $65.0 \pm 1.3$ | $70.5 \pm 0.0$ | $70.8 \pm 0.5$ | $71.2 \pm 0.7$ | $68.6 \pm 1.7$ |
| | Cont. | $94.2 \pm 1.3$ | $95.5 \pm 0.3$ | $96.3 \pm 1.1$ | $96.0 \pm 0.3$ | $84.5 \pm 0.2$ | $88.8 \pm 0.0$ | $95.0 \pm 0.6$ | $94.2 \pm 0.1$ | $\mathbf{97.7 \pm 0.1}$ |
| | Avg. Rank | 6.5 | 5.3 | 4.0 | 5.0 | 7.5 | 4.8 | 5.7 | 3.5 | 2.7 |

### H.3 AUC-ROC Scores over Time

We present the AUC-ROC scores across the test time for all models and continuous-time datasets with an average relative change in performance (rightmost column in Table 13) larger than one, i.e. Enron (Figure 15), UCI (Figure 16), MOOC (Figure 17), and LastFM (Figure 18). We observe that the patterns seen in Figure 5 for the Enron and MOOC datasets extend to all models. For UCI in Figure 16, we can see the AUC-ROC scores for many time windows and comparably fewer batches. This is because the test set for this dataset contains 15% of all temporal edges but two thirds of the whole observation time.

Figure 18 shows that the LastFM dataset contains several outliers in the first half of the test time, where the performance is better compared to the average. These outliers correspond to the peaks in activity shown in Figure 9. Similar to what can be observed for the one anomaly in the Enron dataset, these peaks in activity lead to many batches but only a small number of time windows. This is because these bursts occur in a short period of time, leading to an overrepresentation of these outliers using the batch-based approach. At the end of the LastFM dataset, fewer edges occur, leading to a less reliable model performance, which is underrepresented in the batch-based evaluation due to the smaller number of edges.

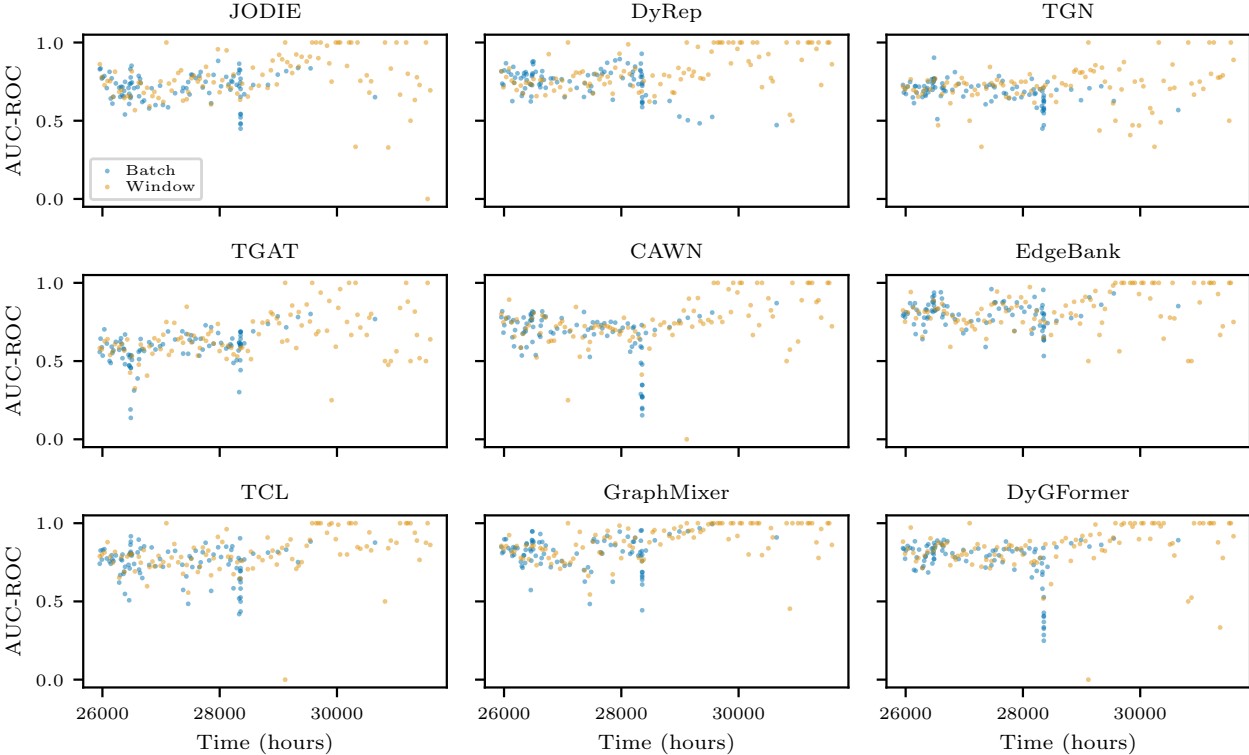

Figure 15: Model performance of Enron dataset over time, which is visualised by the AUC-ROC scores for each batch and time window individually. The time (x-axis) is limited to the test set.

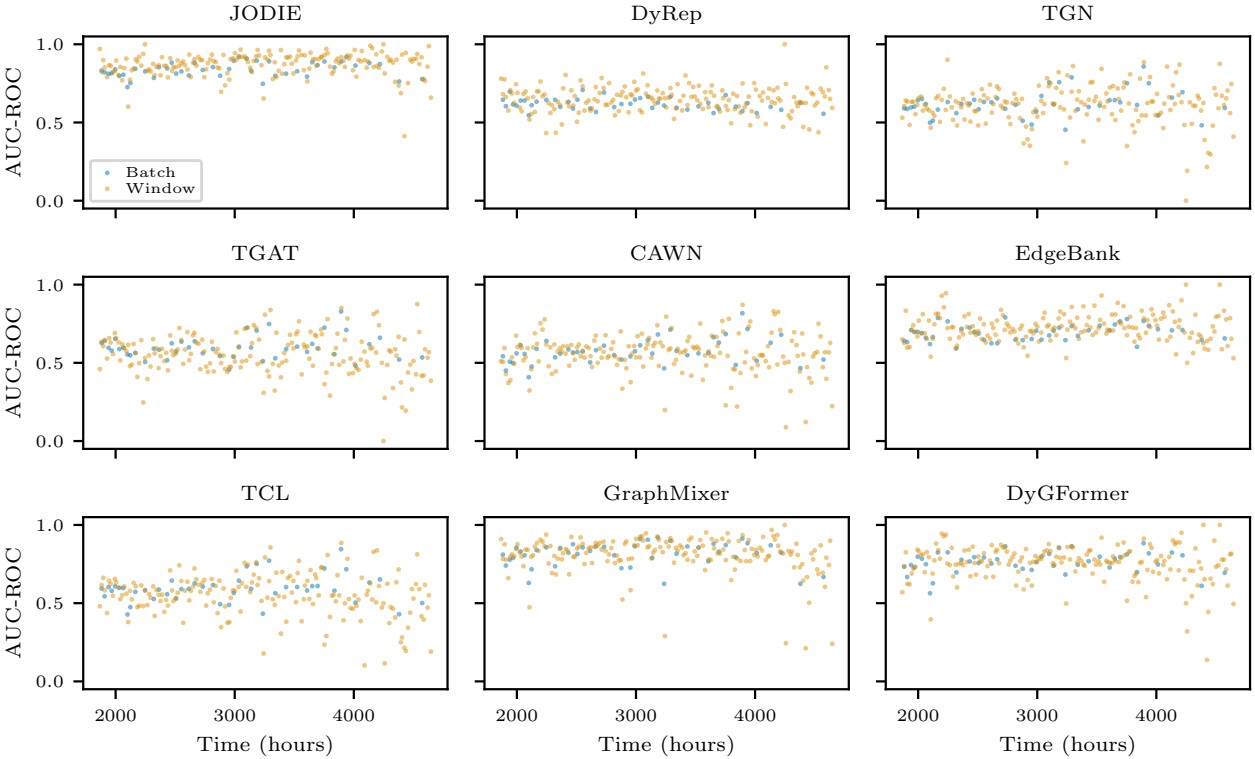

Figure 16: Model performance of UCI over time.

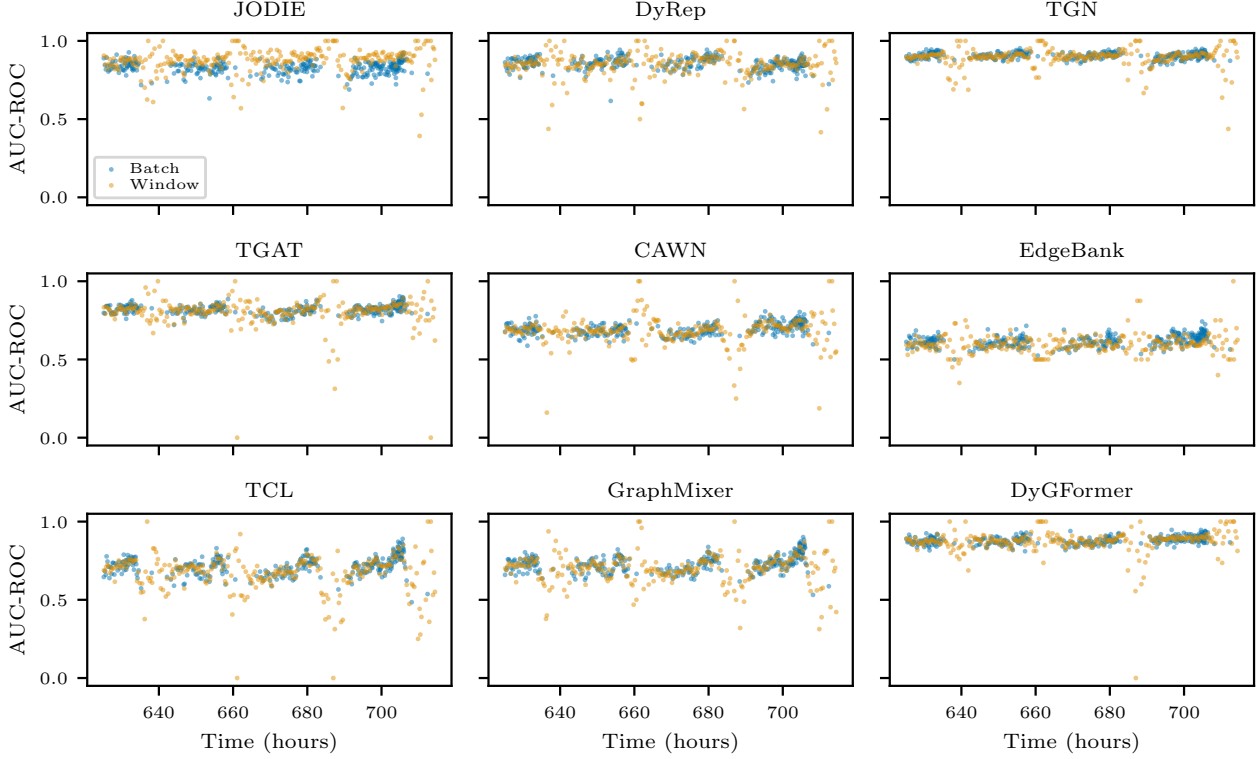

Figure 17: Model performance of MOOC over time.

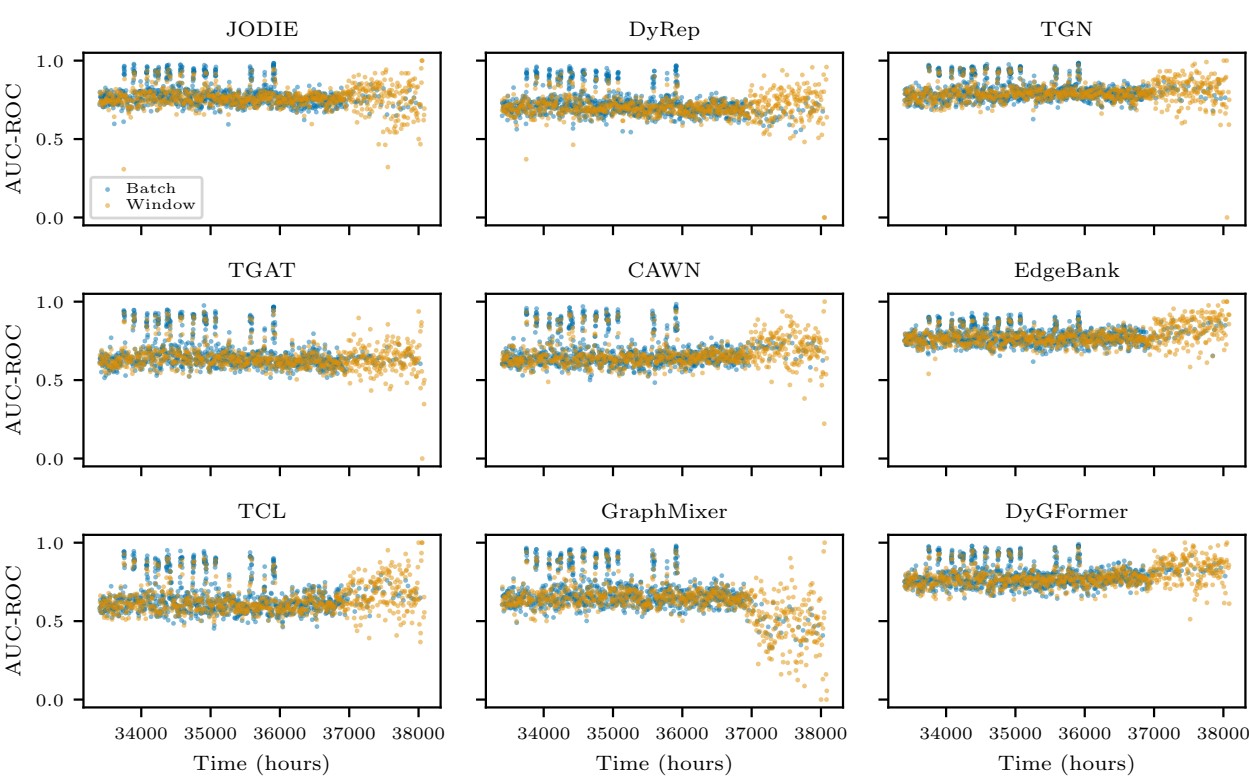

Figure 18: Model performance of LastFM over time.

### H.4  Investigating Information Leakage

The problem of information leakage can give memory-based models an unfair advantage during evaluation because these models obtain more information than non-memory-based models due to this leakage. However, whether this leaked information is useful for making better predictions depends on the data. We assume that the order of the edges within each snapshot plays an important role in many cases where the leaked information is useful. The data could, for example, retain an implicit temporal ordering if the data was obtained by coarse-graining a continuous-time graph into snapshots. The edges inside the snapshots could also be sorted by some ID, like the node ID of the source node. This would mean that all occurrences of each node as a source node would be consecutive, making it very unlikely that each node appears again as a source node after it was already seen in a previous batch of the same snapshot.

In Figure 19, we show the AUC-ROC scores of each batch individually in the evaluated order and highlight the borders of each snapshot with grey dashed lines. We observe upwards trends in the red regression lines for the US Legislators (DyRep, TGN), the UN Trade (all models), and the Flights (JODIE, DyRep) datasets. This suggests that these datasets contain edges in an order that simplifies the prediction for the indicated models, given knowledge from previous batches. Thus, shuffling the data within each snapshot should destroy the useful leaked information and lead to a drop in performance. We report the changes in performance between (i) the models that are evaluated based on edges in the order found in the dataset and (ii) the models evaluated on the datasets with the edges inside each snapshot shuffled in Table 21.

The tables contain the relative change in performance for both evaluation approaches: the batch-based link prediction and our window-based link forecasting. We observe that the performance of the batch-based approach drops for the above-identified dataset and model combinations, suggesting that the edges are indeed ordered in such a way that the earlier batches in a snapshot provide information about later batches, making the prediction easier. Our time-window-based approach prevents this information leakage. Thus, the performance after shuffling is similar to the performance without shuffling because no information, whether useful or not, is leaked.

Table 21: Relative change of the performance between test sets with the order of edges within snapshots, as provided by the discrete-time datasets and test sets with edges shuffled within each snapshot. We compare the evaluation approaches of our window-based link forecasting with batch-based link prediction on all memory-based models.

(a) AUC-ROC scores

| Model Eval | JODIE Forec. | JODIE Pred. | DyRep Forec. | DyRep Pred. | TGN Forec. | TGN Pred. |
|---|---|---|---|---|---|---|
| UN V. | ↓1.6% | ↓20.0% | ↓7.1% | ↓21.4% | ↓1.7% | ↓15.3% |
| US L. | ↓0.1% | ↓3.5% | ↓0.1% | ↓10.9% | ↓0.6% | ↓10.4% |
| UN Tr. | ↓0.2% | ↓13.8% | ↑0.1% | ↓15.5% | ↓3.5% | ↓14.2% |
| Can. P. | ↓0.3% | ↑4.3% | ↓4.4% | ↓0.1% | ↓14.0% | ↓10.1% |
| Flights | ↑1.8% | ↓1.4% | ↑1.3% | ↓2.0% | ↑0.3% | ↑0.1% |
| Cont. | ↓0.9% | ↓1.2% | ↑0.1% | ↓1.1% | ↓0.1% | ↓0.2% |
| $\mu$ | 0.8% | 7.4% | 2.2% | 8.5% | 3.4% | 8.4% |

(b) Average precision scores

| Model Eval | JODIE Forec. | JODIE Pred. | DyRep Forec. | DyRep Pred. | TGN Forec. | TGN Pred. |
|---|---|---|---|---|---|---|
| UN V. | ↓4.2% | ↓18.4% | ↓6.1% | ↓20.5% | ↓1.5% | ↓13.1% |
| US L. | ↓0.0% | ↓2.2% | ↓0.1% | ↓11.0% | ↓1.5% | ↓10.5% |
| UN Tr. | ↓0.2% | ↓10.0% | ↑0.9% | ↓11.4% | ↓3.3% | ↓9.9% |
| Can. P. | ↓0.2% | ↑3.1% | ↓5.3% | ↓0.1% | ↓17.6% | ↓9.1% |
| Flights | ↑1.0% | ↓2.0% | ↑0.9% | ↓3.7% | ↑0.4% | ↑0.7% |
| Cont. | ↓2.3% | ↓3.1% | ↓0.2% | ↓2.4% | ↓0.2% | ↓0.6% |
| $\mu$ | 1.3% | 6.5% | 2.2% | 8.2% | 4.1% | 7.3% |

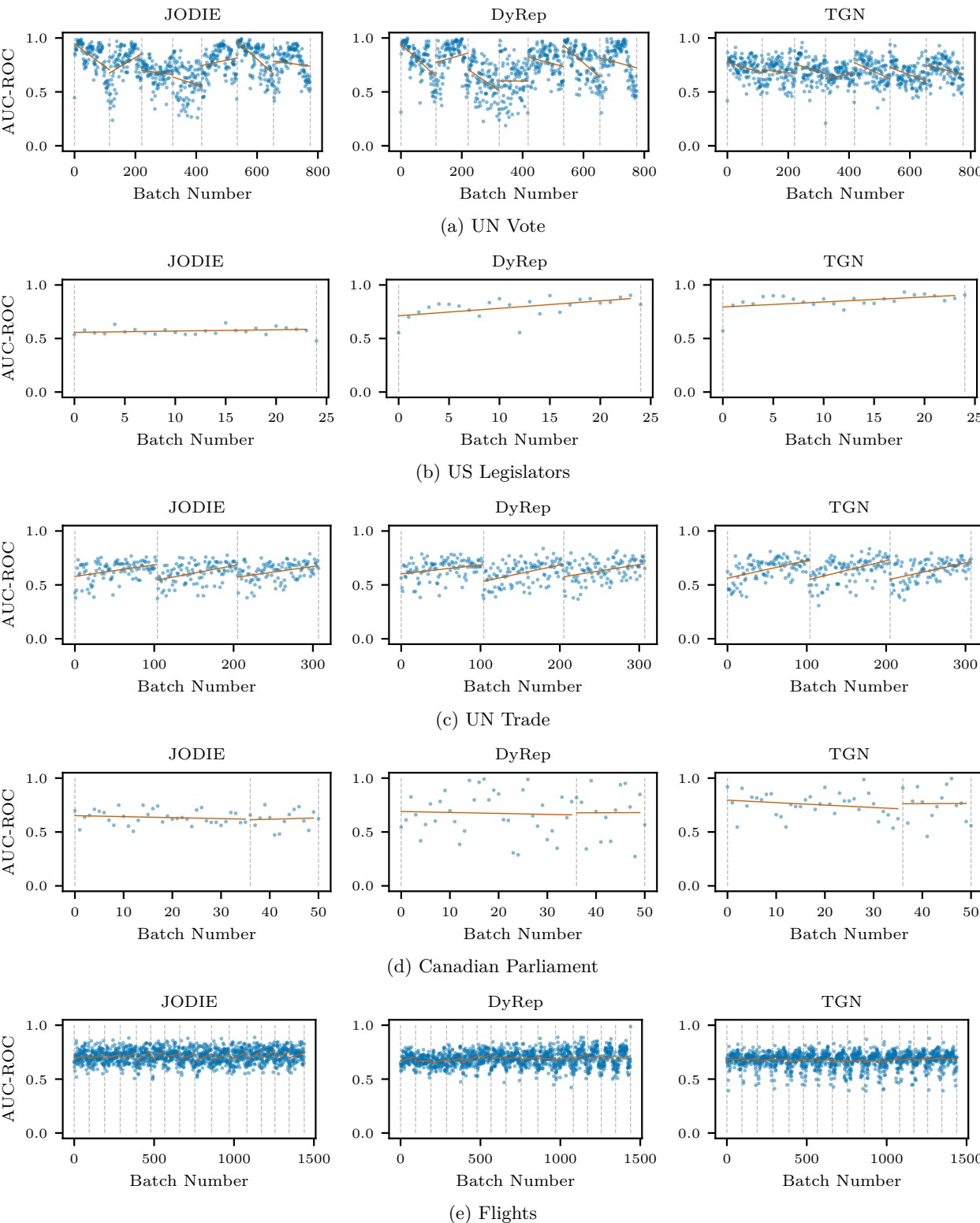

Figure 19: AUC-ROC score for each batch individually in the order the edges are found in each discrete-time dataset (except Contacts, since it has too many snapshots for visualisation). Grey dashed vertical lines separate snapshots, i.e. all points within the same dashed lines correspond to the performances of batches of the same snapshot. The red line represents the best-fit line obtained by performing linear regression on the points within each snapshot.

