# OpenReview forum: "From Link Prediction to Forecasting: Addressing Challenges in Batch-based Temporal Graph Learning"
_TMLR — Accepted by TMLR_

### Review · Reviewer_VJZb · 2025-11-29

**Summary Of Contributions:**

This paper identifies four fundamental issues in batch-based dynamic link prediction—information loss, information leakage, inconsistent prediction windows, and the tunability of task difficulty through batch size. The authors propose dynamic link forecasting, which evaluates models using fixed-duration time windows instead of fixed-size batches. They quantify the issues on 14 datasets, evaluate nine TGNNs, and show that batch-based evaluation can substantially distort model performance, especially for memory-based models on discrete-time datasets. Implementations for DyGLib and PyG are provided.

Strengths:

1. Clear identification of overlooked evaluation flaws.

2. Strong empirical study covering many datasets and models.

3. Practical and intuitive alternative evaluation method.

4. Good reproducibility and clear presentation.

Weaknesses:

1. Guidance on selecting forecasting horizons is limited.

2. Training–evaluation mismatch (batch-based training, window-based evaluation) is not fully discussed.

3. Scalability under extreme temporal densities needs more explicit treatment.

4. Interaction with negative sampling strategies is underexplored.

5. Limited discussion on applicability to ranking-based metrics (MRR).

**Audience:**

Yes

**Audience Explanation:**

The results directly influence how TGNNs should be evaluated and interpreted, which is highly relevant to temporal graph learning researchers and practitioners.

**Broader Impact Concerns:**

I do not identify significant ethical risks beyond those typical for foundational methodological work. As a small enhancement, the authors may briefly discuss the implications of more reliable temporal prediction evaluation in sensitive domains (e.g., mobility, communication networks, socio-economic interactions), though no major concerns are present.

**Claims And Evidence:**

Yes

**Claims Explanation:**

The empirical analyses are thorough, well-designed, and clearly support the core claims. Some secondary aspects (e.g., effect of horizon choice) could be elaborated further.

**Requested Changes:**

Although the paper remains strong overall, the following points would benefit from clearer discussion or refinement.

1. Provide more explicit high-level guidelines for selecting h in practice, including: 1) how h relates to timestamp resolution, 2) how to adapt h for highly bursty vs. uniform datasets, 3) typical ranges for common application domains.

2. Add a dedicated discussion on the implications of evaluating models trained with batch-based methods using window-based evaluation.
Clarify what behavior may change if models are also trained with time-window-based updates.

3. Include a short subsection or appendix elaborating on the runtime and memory implications of window-based evaluation in extreme-density scenarios.

4. Comment on or briefly examine how negative sampling interacts with time-window-based evaluation, especially historic sampling.

5. Expand the limitations section to more clearly articulate what aspects do not transfer to ranking-based evaluation and what would be needed to extend the framework.

---

> ### Author Response · Authors · 2025-12-19
>
> We thank the reviewer for their positive and constructive feedback.
> We integrated all of the suggestions for improvement into the revision of our manuscript and provide more information regarding how we included each suggestion in the following:
>
> 1. We agree that high-level guidelines for selecting the horizon $h$ would be very beneficial. We added detailed guidelines for choosing the horizon $h$ to Appendix E. However, note that selecting $h$ is highly dataset- and application-dependent, and requires the expertise of domain experts in practice.
> 2. We dedicated the second paragraph of our revised "Limitations and Future Work" section to the implications of evaluating models trained with a batch-based approach using a window-based evaluation, including a discussion regarding expected changes in model performance if the models are trained with a window-based approach.
> 3. We elaborate on the runtime and memory implications of window-based evaluation and possible problems in extreme-density scenarios in Appendix G. We provide mitigation strategies using Python-like pseudocode.
> 4. Regarding the influence of our window-based approach on different negative sampling techniques, we added a dedicated section to Appendix A, investigating this relationship in detail.
> 5. We dedicated the first paragraph of the revised "Limitations and Future Work" section to the relationship between our window- and ranking-based evaluation methods. In this section, we compare the pros and cons of both approaches in detail and outline how to extend our framework to such a setting.
>
> Lastly, we followed the suggestion to extend our "Broader Impact Statement".

---

### Review · Reviewer_wmFe · 2025-12-03

**Summary Of Contributions:**

Summary:
This paper demonstrates that the widely used batch-based evaluation protocol for dynamic link prediction in temporal graphs systematically distorts temporal information and can lead to misleading model comparisons. By grouping a fixed number of edges into each batch, traditional evaluation introduces information loss in continuous-time graphs, information leakage in discrete-time graphs, and inconsistent prediction horizons due to varying batch durations. Moreover, the batch size effectively becomes a hidden hyperparameter that influences task difficulty. To address these issues, the authors reformulate the task as dynamic link forecasting, where predictions are made over fixed-duration time windows rather than fixed-size batches. This window-based approach preserves true temporal structure, eliminates leakage within snapshots, and ensures consistent evaluation across models. Extensive experiments on 14 real-world datasets and nine state-of-the-art TGNNs show that forecasting yields more faithful and reliable performance estimates, revealing substantial biases in conventional batch-based evaluation.


Strengths:
1. The paper offers a clear and insightful identification of fundamental weaknesses in the traditional batch-based evaluation protocol for dynamic link prediction.

2. The proposed fixed-horizon, window-based evaluation strategy is simple, intuitive, and empirically well-validated.

3. The paper is highly readable and accessible.

Weaknesses:
1. The level of technical novelty is somewhat limited. While the problem addressed is important and the empirical analysis is thorough, the core methodological contribution--shifting from fixed-size batches to fixed-duration windows--is conceptually straightforward. The work is stronger in its diagnostic and empirical contributions than in proposing fundamentally new modeling or theoretical techniques.

2. The proposed solution focuses on improving evaluation, but does not address information loss or leakage that occurs during training. Since the models are still trained with batch-based temporal ordering, the learned parameters may remain biased, and the resulting models may not fully generalize to temporally coherent test settings. It would be valuable to discuss whether window-based ideas could also extend to training or how training-time biases might impact downstream conclusions.

3. The window-based protocol may face practical limitations in datasets with highly bursty temporal activity. Because the time horizon is globally fixed, certain windows may contain extremely dense interaction bursts, which may result in large memory footprints or potential GPU out-of-memory issues. The paper mentions possible sub-window chunking, but such strategies may still introduce information loss and leakage.

**Audience:**

Yes

**Audience Explanation:**

The paper addresses a fundamental and often overlooked issue in the evaluation of temporal graph learning. The discovery that standard batch-based evaluation can introduce systematic information loss and leakage, and thereby misrepresent model performance, is highly relevant to researchers developing or benchmarking temporal GNNs. The proposed window-based forecasting protocol provides a practical and easily adoptable alternative that improves fairness and comparability across models.

**Broader Impact Concerns:**

I do not see ethical issues.

**Claims And Evidence:**

Yes

**Claims Explanation:**

The paper’s arguments are grounded in systematic empirical analysis across 14 diverse temporal graph datasets and 9 representative TGNN models, covering both continuous-time and discrete-time settings. The authors quantify information loss and leakage using NMI, controlled perturbation experiments, and temporal-density visualizations, and they provide direct comparisons between batch-based prediction and window-based forecasting, which strongly support their conclusions. The experimental results consistently demonstrate that batch-based evaluation distorts temporal ordering and can inflate or deflate model performance, particularly for memory-based architectures. The paper also includes well-designed ablation studies, implementation details, and sanity checks that increase credibility. Overall, the evidence is comprehensive, coherent, and sufficient to substantiate the paper’s central claims.

**Requested Changes:**

1. Discuss the limitations arising from training-time information loss and leakage. The proposed solution addresses evaluation-time inconsistencies but does not mitigate the known issues of temporal discontinuity during training. Since models are still trained using batch-based processing, their parameters may remain biased, and this could limit the generalizability of conclusions drawn from the updated evaluation. A dedicated discussion would help quantify the impact of training-time biases and clarify how they should be interpreted in practice.

2. Address the potential memory challenges of window-based evaluation on highly bursty datasets. Because time windows have a fixed temporal span, windows covering periods of extreme edge density may contain very large numbers of events, potentially resulting in out-of-memory errors or substantially higher compute costs.

---

> ### Author Response · Authors · 2025-12-19
>
> We are grateful for the reviewer's insightful comments and have incorporated all suggestions for improvement in the revised version of our manuscript.
>
> - We discuss the limitations arising from training-time information loss and leakage in the second paragraph of our revised "Limitations and Future Work" section. In particular, we discuss expected changes to model performance when using our window-based approach for training and evaluation.
> - We dedicated Appendix G.1 to presenting an approach for handling possible memory overflows, and included Python-like pseudocode for clarity.

---

### Review · Reviewer_3kuv · 2025-12-05

**Summary Of Contributions:**

- This paper clearly and concisely introduces the concepts of Temporal Graph Neural Networks (TGNN) and dynamic link prediction (the task of predicting future graph links events from past graph link events).
- The paper identifies significant methodological flaws that arise from evaluating dynamic link prediction methods using the same fixed-batch-size structure as is typically used to train TGNNs. Specifically, the paper gives a clear description of four major issues which arise from batching events into groups of fixed size rather than fixed duration:
    - (a) information loss (from batches encompassing very long durations),
    - (b) information leakage (from splitting the same snapshot time into multiple batches),
    - (c) inconsistent forecast horizons (which depart from the real-world motivations underlying TGNN evaluations),
    - And (d) tunable batch size (i.e. allowing the forecasting method to change the evaluation itself to get a better score).
- The authors introduce empirical measurements that quantify the issues they have identified with the dynamic link prediction approach to evaluation across a variety of evaluation datasets commonly applied to TGNNs.
- The authors introduce a new evaluation methodology, termed dynamic link *forecasting*, which solves the identified problems of dynamic link prediction by using fixed-duration evaluation batches rather than fixed-event-count batches. They empirically demonstrate the varying degrees of impact that remedying the four identified issues has across a variety of evaluation datasets commonly applied to TGNNs.


Strengths:
* This work is straightforward to follow thanks to the authors’ efforts towards clear exposition through diagrams, clean prose, and well-organized tables and figures.
* The authors make a thoroughly argued and highly intuitive case for evaluating using fixed-time slices rather than fixed-event-count slices.
* The authors implement their novel evaluation methodology and evaluate a wide body of TGNN works in the new paradigm, presenting a substantial body of empirical evidence to back up their claims.


Weaknesses:
* The current structure of the manuscript entails substantial repetition of core ideas which, while often instructive, may to some readers feel redundant. In some ways the later sections of the paper are a victim of the success of the introduction in so clearly laying out the issues with dynamic link prediction that when these problems are recapitulated later, the repeated explanations can feel unnecessarily verbose.
* Though the introduction explains the present paradigm of dynamic link prediction as arising from a need to batch computation, the authors do not provide measurements of the runtime impact incurred by switching to their method of evaluation.

**Audience:**

Yes

**Audience Explanation:**

The paper addresses a fundamental issue in the evaluation of Temporal Graph Neural Networks. Researchers developing and applying TGNN methodologies should certainly find this work interesting.

**Broader Impact Concerns:**

I have no concerns.

**Claims And Evidence:**

Yes

**Claims Explanation:**

- The authors introduce empirical measurements that quantify the issues they have identified with the dynamic link prediction approach to evaluation across a variety of evaluation datasets commonly applied to TGNNs.
- They empirically demonstrate the varying degrees of impact that remedying the four identified issues has across a variety of evaluation datasets commonly applied to TGNNs.

**Requested Changes:**

Requests
* Clarification of computational cost associated with unbalanced batch sizes during evaluation
* More consistency in the number of significant figures presented in tables, e.g. tables 2 and 4.
* Some discussion of the asymmetry/skew beyond the mean and standard deviation statistics of batch size show in Table 3, plus some discussion of whether this is important. From Appendix C, it appears that some distributions have quite extreme spreads of event counts.
  - If alternative statistics (e.g. the interquartile range) show interesting asymmetrical qualities of event count distributions, it may be nice to include these in the paper. Expanding appendix C with histograms or other distribution-summarizing plots of window sizes for the specific horizon lengths studied in the main body of the paper would also be of interest. However measured, some discussion of the significance of the often-dramatic differences in number of events per time window in the real-world datasets studied could help readers get a deeper appreciation of fixed-time-window evaluation.

Food-for-thought suggestions:
* As evidenced by the lovingly padded-out appendix, there is a lot of great content that the authors would like to share with the community. To make room for more content, the authors may consider removing some repetitions of the issues identified with temporal link prediction. In particular, Figure 1, the intro text, and section 3.1 all discuss the same problems in somewhat the same way.
* Offering a little more discussion as to the implications of temporal link forecasting to the design and training of TGNNs (beyond the single sentence given in the last subsection of the conclusion at present) may be of great interest to readers.
* TMLR does not require novelty for publication, so the authors are free to modify discussion in the Open Challenges In Link Prediction section that emphasizes how novel this work’s approach is and how much more thorough it solves the identified problems as compared to prior works. For example, the authors may consider instead highlighting the complementary strengths of prior works rather than their limitations.

---

> ### Author Response · Authors · 2025-12-19
>
> We appreciate the reviewer's positive assessment of our work. We included their constructive feedback as follows:
>
> - We clarify the runtime and memory implications of our approach on unbalanced window sizes in Appendix G of our revision. We provide a detailed explanation on how to mitigate memory overflow issues with pseudocode. Another subsection investigates the empirical runtime of our approach in our experiments and compares it with the runtime of batch-based evaluation.
> - We corrected the inconsistent display of decimal places of NMI values shown in Tables 2 and 3. In the revised version of our manuscript, all NMI values are reported with four decimal places.
> - We investigated the asymmetry/skew of the window sizes in continuous-time graphs with histograms in Figure 13 of our revised manuscript and discuss the figure in Appendix C.5.
>
> We thank the reviewer for the additional suggestions for improvement. We followed the suggestion to include a more detailed discussion on the implications of temporal link forecasting to the design and training of TGNNs in our "Limitations and Future Work" section.

---

### Author Response · Authors · 2025-12-19
**Aggregate Response**

We thank the reviewers for their consistently positive feedback and helpful suggestions.
We uploaded a revised manuscript that addresses all of the reviewers' concerns.
In this aggregate response, we briefly list all changes made to our manuscript to address the points raised by the reviewers. We provide a more detailed description of the changes in the individual responses to the reviewers.

- As suggested by all reviewers, we have discussed the implications of using a window-based evaluation on models trained on batches in the "Limitations and Future Work" section.
- We address the questions by reviewers VJZb and wmFe regarding empirical runtime and handling possible memory overflows in Appendix G.
- We corrected the inconsistent display of decimal places of NMI values shown in Tables 2 and 3 as suggested by reviewer 3kuv.
- We investigated the asymmetry/skew of the window sizes in continuous-time graphs with histograms in Figure 13 of our revised manuscript and discuss the figure in Appendix C.5 as suggested by reviewer 3kuv.
- We added detailed guidelines on how to choose the horizon $h$ to Appendix E as suggested by reviewer VJZb.
- We added a dedicated section to Appendix A that investigates the influence of our window-based approach on different negative sampling techniques as suggested by reviewer VJZb.
- We extended the "Limitations and Future Work" section by a discussion on the relationship between our window-based and ranking-based evaluation methods as suggested by reviewer VJZb.
- Lastly, we followed the reviewer VJZb's suggestion to extend our "Broader Impact Statement".

We believe that these additions have further strengthened our manuscript and hope that it is now suitable for publication.

---

### Decision · Action_Editor_KSTf · 2026-01-20

**Recommendation:** Accept as is

**Additional Comments:**

The paper exposes a flaw in traditional batch-based evaluation for temporal link prediction and suggests a window-based alternative, highlighting how standard methods can bias performance comparisons. The ideas and results resonate with community members. All three reviewers agree to accept this manuscript. I am glad to recommend final acceptance.

**Audience:**

Yes

**Audience Explanation:**

Yes, at least some individuals in TMLR's audience would likely be interested in the findings of this paper. The insights into flaws in traditional evaluation methods for temporal link prediction and the proposed window-based alternative address relevant issues in the field.

**Claims And Evidence:**

Yes

**Claims Explanation:**

The authors provide robust empirical analysis and well-documented data that effectively substantiate their arguments.